# Broad misappropriation of developmental splicing profile by cancer in multiple organs

**Arashdeep Singh** [1]✉, **Arati Rajeevan**[1,3], **Vishaka Gopalan**[1,3], **Piyush Agrawal**[1], **Chi-Ping Day** [2] **& Sridhar Hannenhalli** [1]✉

Oncogenesis mimics key aspects of embryonic development. However, the underlying mechanisms are incompletely understood. Here, we demonstrate that the splicing events specifically active during human organogenesis, are broadly reactivated in the organ-specific tumor. Such events are associated with key oncogenic processes and predict proliferation rates in cancer cell lines as well as patient survival. Such events preferentially target nitrosylation and transmembrane-region domains, whose coordinated splicing in multiple genes respectively affect intracellular transport and N-linked glycosylation. We infer critical splicing factors potentially regulating embryonic splicing events and show that such factors are potential oncogenic drivers and are upregulated specifically in malignant cells. Multiple complementary analyses point to *MYC* and *FOXM1* as potential transcriptional regulators of critical splicing factors in brain and liver. Our study provides a comprehensive demonstration of a splicing-mediated link between development and cancer, and suggest anti-cancer targets including splicing events, and their upstream splicing and transcriptional regulators.

Cancer onset and progression results in the dedifferentiation and gradual loss of lineage-specific phenotypes and echoes multiple facets of early embryonic development including rapid proliferation, epithelial-mesenchymal transition (EMT), cellular migration, and angiogenesis. The mechanistic details of these cancer-associated changes in cellular function and physiology, termed as 'hallmarks of cancer'[1], are not completely understood. Past studies have shown that a core set of transcription factors (TFs) and signaling pathways, which maintain pluripotency in embryonic stem cells (ESCs) and orchestrate normal embryonic development, are reactivated in cancer and thus underlie physiological reversal in cancer progression[2–5]. For instance, the core pluripotency markers *OCT3/4* and *SOX2*, are important biomarkers of several cancers[6–8]. Likewise, the Myc module of ESCs gets reactivated in mouse models of mixed-lineage leukemias and is a predictor of patient outcome in many human cancers[5]. Consistent with these anecdotes, a universal signature of stemness accurately predicts the tumor infiltration by leukocytes and response to immunotherapy[9].

In addition to TFs, various signaling pathways involved in embryonic development, such as Wnt, Notch, and Hippo, also get reactivated in cancer and their associated genes accumulate oncogenic mutations[3,10,11].

In addition to gene expression, alternative splicing (AS), wherein multiple isoforms of the same gene are expressed, affects >95% of the multi-exonic genes in humans[12,13] and underlies diverse biological processes such as stemness, differentiation, development, and ageing[14–17]. A plethora of gene-centric studies have demonstrated the critical role that AS plays in cancer[18]. For instance, long and short isoforms of Bcl-x protein have anti-apoptotic and pro-apoptotic roles respectively[19,20]. Several members of the receptor tyrosine kinase family express multiple isoforms enhancing the proliferative or metastatic ability of cancer cells. For example, the *FGFR2* isoform, FGFR2III-b, is mainly expressed in epithelial cells while FGFR2III-c is expressed in mesenchymal cells[21]. This isoform switching is involved in epithelial-mesenchymal transition (EMT)[22] and is linked to invasiveness

[1]Cancer Data Science Laboratory, National Cancer Institute, National Institutes of Health, Bethesda, MD, USA. [2]Laboratory of Cancer Biology and Genetics National Cancer Institute, National Institutes of Health, Bethesda, MD, USA. [3]These authors contributed equally: Arati Rajeevan, Vishaka Gopalan. ✉e-mail: arashdeep.singh@nih.gov; sridhar.hannenhalli@nih.gov

and metastasis of colorectal[23,24] and breast cancers[25]. Likewise, alternatively spliced isoforms of genes such as *P63*, Cyclin D1, *CD44*, *HRAS*, *RAC1*, and *PKM* can modulate proliferative, apoptotic, metabolic, and invasive properties of cancer cells[18,26,27]. Recent comparative transcriptomic analyses across multiple organs showed the prevalence and cross-species conservation of alternative splicing events during development[28]. Despite the established importance of AS in development and cancer, as well as broad phenomenological links between development and cancer, an unbiased and comprehensive investigation of the links between development and cancer AS events in a tissue-specific fashion is still lacking and can have major implications on our broader mechanistic understanding of oncogenesis and cancer therapies.

In this work, leveraging the human developmental transcriptome across multiple time points in three organs[29] as well as the transcriptomic data of the corresponding cancer from The Cancer Genome Atlas (TCGA) (https://www.cancer.gov/tcga), we chart the landscape of embryonic splicing events that are reactivated in the organ-specific cancer, and investigate their upstream regulators and downstream functional implications. Focusing on the most common type of AS event type, namely, exon skip events, we show that embryonic AS events associate with key oncogenic processes such as rapid proliferation, migration, and angiogenesis, and are significantly reactivated in tumors. The reactivation of embryonic AS events predicts the patient's survival and is associated with the proliferation rate in cancer cell lines. Among 'embryonic positive' (EP) and 'embryonic negative' (EN) exons, the nitrosylation domain (ND), transmembrane-region domain (TRD), and WD40 domain are significantly enriched in all three tissues. Detailed molecular and functional analysis reveals that NDs and TRDs respectively affect retrograde cellular transport by coordinately regulating the activity of Arf and Ras family GTPases and N-linked glycosylation by regulating the transmembrane localization of oligosaccharyl transferase subunits. We further train a splicing regulatory model based on the developmental gene expression data of splicing factors which accurately predicts the inclusion of embryonic AS events in cancer patients and identifies critical splicing factors (CSFs) potentially regulating embryonic AS events. The identified CSFs are upregulated in cancer, often accompanied by copy number amplifications. Leveraging tumor single cell RNA-seq data, we show that the CSFs are specifically activated in the malignant epithelial cells, further supporting their role in malignancy. Based on multiple complementary approaches, we identify key transcription factors (TFs) predicted to regulate the identified CSFs, including *MYC* and *FOXM1* in the brain and liver, respectively, and can be targeted using known FDA-approved drugs. Overall, our work establishes, through multi-modal data integration, reversal to developmental AS in cancer, and suggests therapeutic avenues directly targeting the regulators of such a reversal.

## Results

### Identification of exons associated with human fetal development

To identify the AS events associated with fetal development, we implemented a two-step approach where we first identified fetal development associated pathways, and then obtained the AS events correlated with those pathways (Fig. 1a); the rationale and advantages of this approach are discussed in the Methods section and Supplementary Note 1. Based on organ-specific transcriptomic data across multiple stages (Supplementary Data 1) of pre- and post-natal development[29], we first estimated the activity for each of the 332 KEGG pathways[30], quantified as the median expression of the pathway genes, in each sample, independently in brain, liver and kidney tissues. Principal component analysis (PCA) of the pathway activity clearly separates the pre- and post-natal stages along the first principal component (Supplementary Fig. 1a). Clustering of pathways in the PCA space

("Methods") revealed two mutually exclusive sets of pre- or post-natal pathways which were correspondingly assigned as 'embryonic positive' or 'embryonic negative' (Fig. 1b).

As expected, genes constituting embryonic positive pathways are enriched in several gene ontology (GO) terms related to the processes which are crucial for embryonic development such as EMT, extracellular matrix (ECM) remodeling, cellular proliferation, and angiogenesis, providing additional validation of our approach used to detect embryonic pathways (Fig. 1c, Supplementary Data 2). Next, we used PEGASAS[31] to identify alternative exons whose sample-specific inclusion is significantly correlated with the activity of embryonic positive pathways across developmental timepoints ("Methods"). We defined an exon as embryonic positive (EP) or embryonic negative (EN) based on the fraction of embryonic positive pathways whose activities are respectively significantly positively or negatively correlated with the exon's inclusion level (Fig. 1d, e, "Methods"). We thus identified on average ~2000 EP as well as EN exon skip events in each tissue (Supplementary Data 3); as expected, EP and EN exons exhibit opposite inclusion patterns in the pre- and postnatal stages (Fig. 1e, and Supplementary Fig. 1b).

We found that the EP and the EN exon inclusion levels are broadly uncorrelated with the expression of their host genes, suggesting that these AS events vary independent of their host gene's expression (Fig. 1d, Supplementary Fig. 1c). This independence is further supported by our observation that ~20-30% of the host genes of EP/EN exons in fact contain both EP and EN events (Supplementary Fig. 1d). Moreover, in almost all cases (>99%) when an exon's inclusion correlates with an embryonic positive pathway, the exon's host gene is not member of that pathway. Collectively, these data suggest that AS provides an additional regulatory layer to gene expression programs for controlling developmental pathways.

The host genes of EP and EN exons are significantly enriched in tissue specific processes in the case of brain and liver (Supplementary Fig. 1e, Supplementary Data 4). For example, GO terms for neuronal activities, such as synapse organization, dendrite development, neuron death, cell polarity, regulation of neurotransmitters, are enriched in the host genes of EP/EN exons specifically in brain. Likewise, liver EP/EN exons are involved in the regulation of many key metabolic processes as well as regulation of cell junctions and cytokinesis. EP/EN exons in all three tissues are enriched for autophagy, consistent with the emerging role of AS in the regulation of autophagy[32]. Overall, we identify numerous exons that, independent of the expression of their host gene, are preferentially utilized during fetal development and repressed postnatally, and strongly associate with key developmental and oncogenic processes.

### Embryonic AS events are recapitulated in cancer and are associated with cancer stage and patient survival

We next assessed the extent to which the organ-specific EP events are recapitulated in the corresponding cancer types. First, we found that in all three organs the genome-wide profile of the AS events clearly distinguishes tumor samples from their non-malignant counterparts in TCGA (Supplementary Fig. 2a), as observed previously[31,33]. Next, we identified the cancer-associated AS events in each organ by comparing the splicing profiles in tumors with healthy GTEx counterparts ("Methods", Fig. 1a) and assessed their overlap with organ-specific EP and EN events. In all three organs we found that the EP events are significantly enriched among the AS events frequently increased in cancer, while the EN events are enriched among the AS events frequently decreased in cancer (Fig. 2a). These enrichment values correspond to the reactivation of almost 50% of the embryonic events in brain, 20% in kidney and 15% in liver, implying that several hundred (in liver and kidney) to thousands (in brain) of alternative splicing events in cancers revert back to their embryonic counterparts (Supplementary Fig. 2b, c). The observed enrichment may simply be because EP

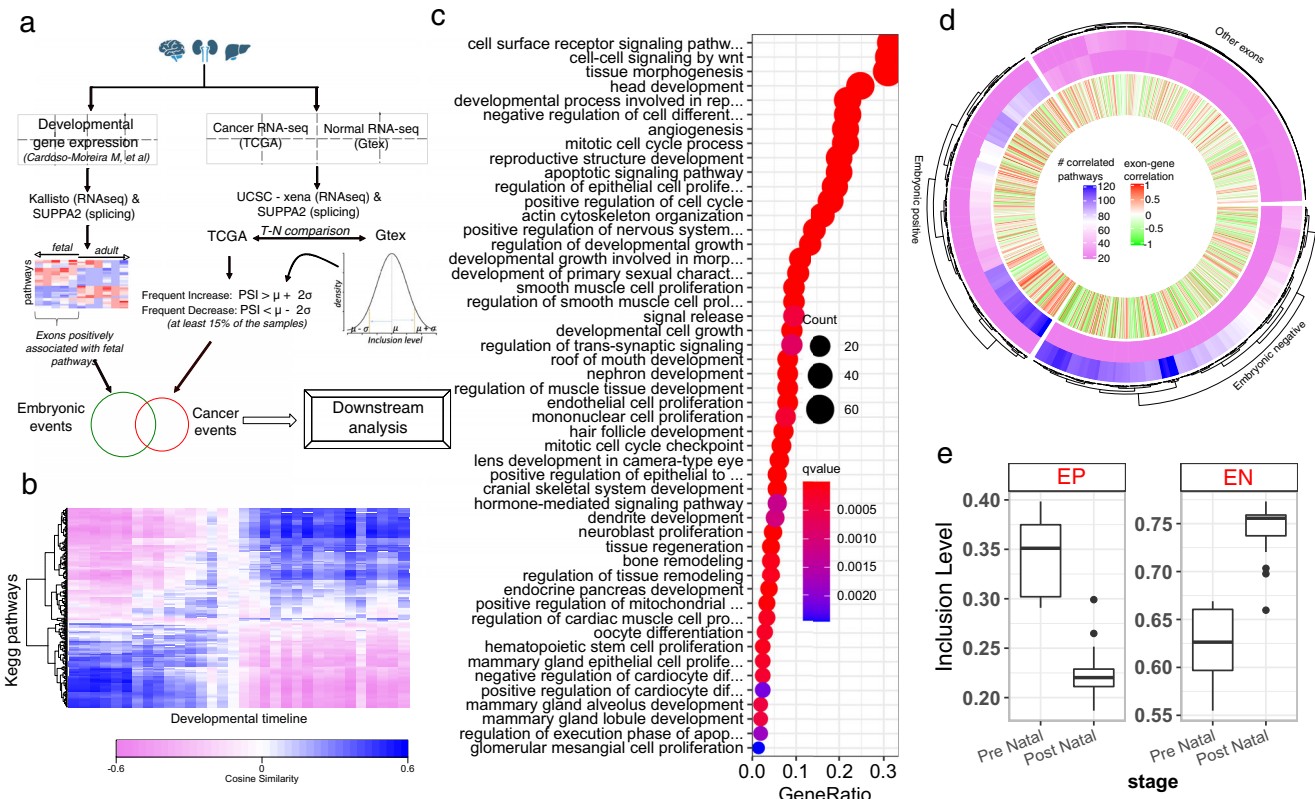

**Fig. 1 | Detection of AS events relevant to development of organs. a** Overview of the pipeline for the identification and comparison of developmental and cancer-associated splicing events. **b** Hierarchical clustering of KEGG pathways in brain cerebellum. Each colored cell in the heatmap corresponding to a pathway p and a developmental time point t represents the cosine similarity between p's contribution (loading) to the first 5 PCs and t's PC score for the first 5 PCs, thus indicating the activity of pathway p at timepoint t ("Methods"). **c** Dot plot for the GO term enrichment of the genes comprising embryonic pathways inferred in (**b**). Dots are colored based on FDR-corrected one-sided p-value from Fisher's test (labelled as q-value) as implemented in clusterProfiler package in R and sized based on the number of genes in each functional category. **d** Circular heatmap showing the number of positively and negatively correlated embryonic pathways with each exon. Each leaf in the dendrogram is an exon. Outer two rows represent respectively the number (per legend colors) of positively and negatively correlated embryonic pathways with the exon. The innermost layer shows the Pearson's correlation coefficient of the PSI value of each exon with the expression of its host gene. For visual clarity, only 1000 randomly chosen exons are included in the plot. **e** Boxplots showing the differential inclusion of embryonic positive (EP) and embryonic negative (EN) events during pre-natal (n = 11) and post-natal (n = 21) stages of development. Each data point in the boxplots is the median inclusion level of the EP and EN exons at each developmental time point sampled by Cardoso-Moreira et al.[29]. The horizontal line in the middle of boxplots is the median value and the lower and upper edges of the boxes correspond to the 25th and 75th percentiles of the inclusion level (y axis). Extending vertically upwards/downwards of the boxes are the lines showing 1.5 times the interquartile range (i.e., distance between 25th and 75th percentile). Dots are the outliers. Source data for these figures are provided as a Source Data file.

events have lower inclusion level in healthy postnatal tissues as shown in Fig. 1e and are therefore more likely to increase in cancer (analogously EN events might be more likely to decrease). We ruled out this potential confounder by randomly sampling alternatively spliced exons with low (psi < 0.3) and high (psi > 0.7) inclusion level in healthy GTEx samples of liver and testing their enrichment among the events frequently increased and decreased in liver cancer, respectively (nominal false positive rate <0.01; Supplementary Fig. 2d; Methods). Additionally, removing the exons with ~0 inclusion in healthy GTEx tissues did not affect the enrichment of EP events among the cancer-specific events (Supplementary Fig. 2e). Further, the ΔPSI values for between pre- and post-natal stages were strongly correlated with the ΔPSI values between TCGA and GTEx, in brain and liver, hinting at the broad and global similarity in the pattens of alternative splicing during embryonic development and cancer (Supplementary Fig. 2f). Using an alternative approach to quantify cancer-specific events or filtering EP events based on stringent ΔPSI criteria (prenatal – postnatal > 0.2) did not affect the significance of embryonic splicing in cancer (Supplementary Fig. 2g and Supplementary note 1). We observe an even greater enrichment of EP and EN events in advanced tumors compared with early-stage tumors ("Methods"; Supplementary Fig. 2h, I), linking

embryonic splicing to not only oncogenesis but also to cancer progression. Furthermore, in all three organs, the EP (respectively EN) inclusion levels across samples are positively (respectively, negatively) correlated with cancer hallmark signature gene set scores (Fig. 2b), indicating a possible direct link between oncogenic processes and embryonic splicing. Unlike other signatures, apoptosis and DNA damage gene sets, whose activity is known to inversely correlate with tumor aggressiveness[1], are negatively correlated with EP events.

Next, we directly assessed whether the EP and EN inclusion level is associated with patient survival using Cox regression ("Methods"). In all three tissues, EP inclusion had significantly higher (and positive) hazard ratios and EN inclusion had respectively lower (and negative) hazard ratios compared to the rest of the exons (Fig. 2c); we ensured that the observed trends were not confounded by the expression level of the host genes (Supplementary Fig. 2j).

Since early embryonic development shares several molecular programs across organs[29], we derived a set of EP events (197 events) common to all three organs and assessed its association with survival across 20 cancer types (Fig. 2D). We further hypothesized that the shared set of EP events were more likely to result in worse prognosis across multiple cancer types, and found that indeed, a greater fraction

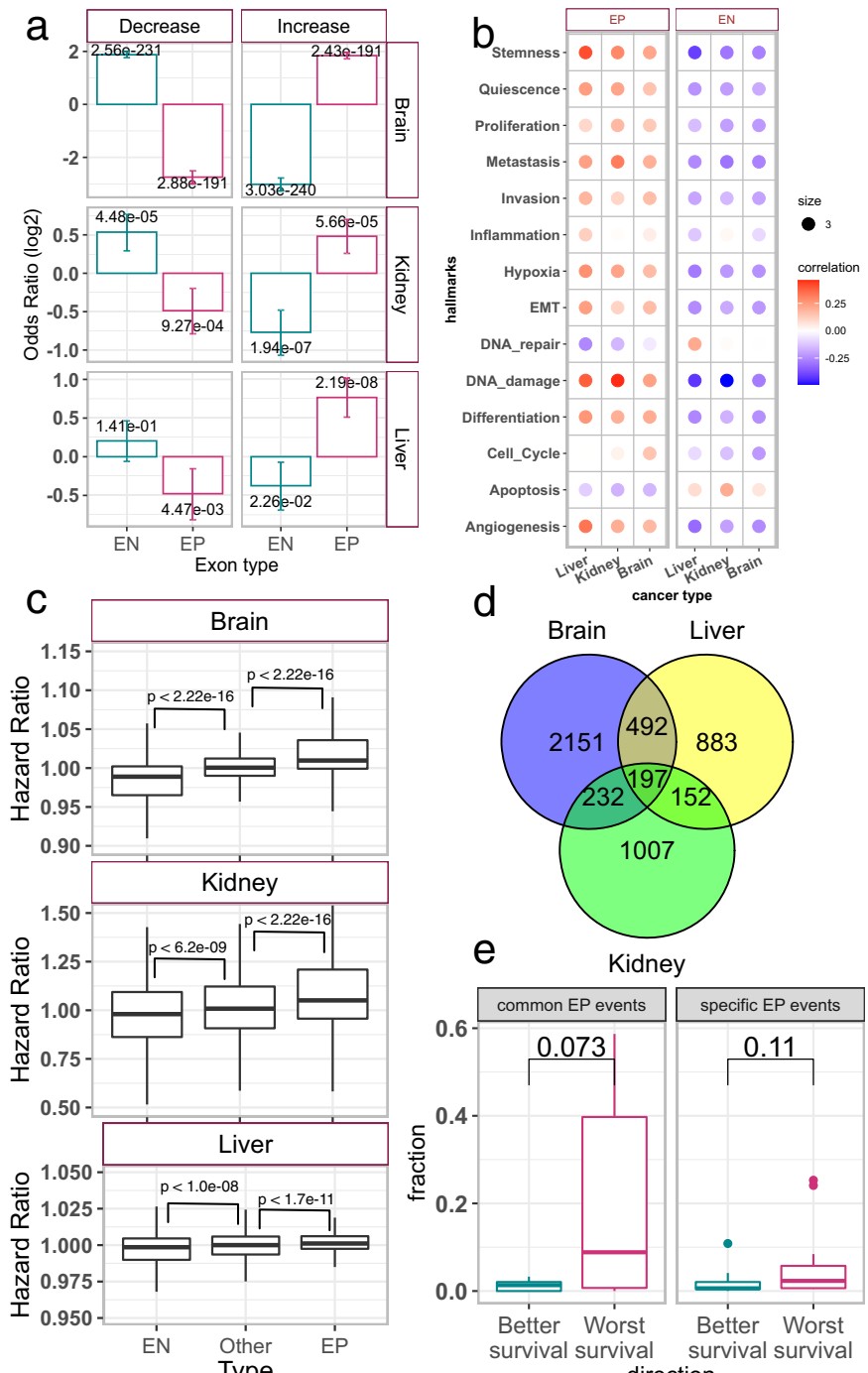

**Fig. 2 | Embryonic splicing events in cancer. a** Bar plots showing the odds ratio and 95% confidence intervals (whiskers) calculated using Fisher's test to assess the statistical significance of overlap between embryonic splicing events and frequently increased/decreased events in cancer for brain, kidney, and liver. The numbers at the top of each bar are FDR-corrected two-sided *p*-values from Fisher's test. **b** Dot plots for Pearson's correlation between the median inclusion level of EP and EN events and mean expression (log (tpm + 1)) of the cancerSEA hallmark gene sets in cancer samples. **c** Boxplots distribution of hazard ratios of the EP (*n* = 3051) and EN (*n* = 3457) detected in brain, kidney and liver in their corresponding cancers. The 'Other' set (*n* = 29,349) of exons are the remining exon and serve as genome-wide control. The cancer types used in this analysis are LGG for brain, LIHC for liver, and KIRP for kidney. Two-sided *p*-values from Wilcoxon's test are shown. **d** Venn

diagram shows the overlap between the EP events detected in three tissues.
**e** Boxplots showing the proportion of specific EP events (detected in only 1 tissue) and common EPs (detected in all three tissues), with better (HR < 1, FDR < 0.1, *n* = 10 for common EP events and *n* = 12 for specific EP events) or worse survival (HR > 1, FDR < 0.1, *n* = 10 for common EP events and *n* = 12 for specific EP events) across 20 different cancer types from TCGA. Each data point is a cancer type. Two-sided *p*-values from Wilcoxon's test are shown. In boxplots (**c**, **e**), the horizonal line in the middle is the median value and the lower and upper edges of the boxes correspond to the 25th and 75th percentiles. Extending vertically upwards/downwards of the boxes are the lines showing 1.5 times the interquartile range (i.e., distance between 25th and 75th percentile). Dots are the outliers. Source data for these figures are provided as a Source Data file.

of EP events resulted in poor prognosis of in multiple cancer types (Fig. 2e, single-tailed *p* value <0.05), further underscoring the embryonic roots of splicing changes in cancer.

### Alternatively spliced transmembrane-region and nitrosylation domain may regulate N-linked glycosylation and retrograde cellular transport during development and cancer

To get insights into the functions potentially affected by dynamic inclusion of EP and EN exons, we performed molecular functional enrichment analysis of the genes containing the EP and EN events. In all organs, we observed a significant enrichment of Ras GTPase binding, cell adhesion, and cytoskeleton binding classes such as cadherin, actin, and microtubules (Supplementary Fig. 3a). Brain and Liver EP/EN genes were additionally enriched for dynactin and clathrin binding (Supplementary Fig. 3a). These processes promote tumorigenesis by modulating the cytoskeleton and cellular transport during the proliferation and migration of cancer cells[34,35]. A more detailed discussion is provided below in the "Discussion" section.

To gain further insights into the molecular role of EP and EN exons and investigate their link with oncogenesis, we identified protein domains from PFAM database[36] enriched among the EP/EN exons ("Methods"). Three domains− transmembrane-region domain (TRD), nitrosylation domain (ND), and WD40−are enriched among EP and EN exons in all three organs (Fig. 3a), leading us to speculate their potential role in some of the functions performed by the host genes of EP and EN exons. To explore this potential link, we identified the gene subsets whose EP/EN exons contained these domains (total 6 gene subsets per tissue: 3 domains × 2 EP/EN gene sets) and performed molecular function enrichment analysis for each subset (Fig. 3b). As expected, enriched molecular functions in a gene set could be unambiguously attributed to the corresponding domain. For instance, gene subsets of WD40 domains were enriched for ubiquitin binding, consistent with the established role of WD40 as binding interfaces for ubiquitin proteins[37]. Likewise, the genes containing the transmembrane region domain were indeed enriched for various kinds of transmembrane transporters (Fig. 3b). Further, the assessment of overlap among the host genes of EP and EN exons harboring these domains across tissues indicates that the observed enrichment of protein domains is not driven by the same set of genes but instead, multiple host genes of EP and EN exons coordinately splice in and out these domains across tissues (Fig. 3b). To probe the interplay between these enriched molecular functions and biological processes affected by dynamic inclusion of these domains, we performed biological processes enrichment analysis on the same gene sets and assessed the overlap of genes having a specific enriched molecular function with those having a specific enriched biological process.

The observed correspondence between molecular function and biological processes among the host genes of EP and EN exons is well supported. For instance, in brain, host genes of EN exons with a transmembrane domain and encoding various types of transporters (molecular function) are predominantly involved in cross-membrane transport (biological process) (Fig. 3c).

Moreover, in brain and liver EP exons, the molecular function oligosaccharyl transferase activity significantly overlapped with biological processes related to N-glycosylation of proteins (Fig. 3c, and Supplementary Fig. 3c), a modification which typically takes place in the phospholipid bilayer of ER and Golgi bodies through the multi-subunit oligosaccharyl transferase complex (OST). We observed that four subunits of OST showed a coordinated reduction in the inclusion of TRD from pre- to post-natal stages, which increased again in cancer patients in brain (Fig. 3e), with *TUSC3* and *RPN2* undergoing greatest change. This suggests that modulation of transmembrane localization of OST through alternative splicing of TRD during embryogenesis might directly impact the process of N-glycosylation. Notably, N-glycosylation of several proteins have been implicated in cellular

proliferation and migration by modulating the cell-matrix interactions[38]. Therefore, increased inclusion of TRD among the subunits of OST might help the cancers (Fig. 3e) to upregulate the increased demand for N-glycosylation. To the best of our knowledge, the role of alternatively spliced TRDs among the subunits of OST complex in regulation of N-glycosylation has not been reported so far. To support this conclusion that removal of TRD can affect the function of OST by affecting its localization, we highlight the example of an integrin gene, *ITGA2B,* which contains an EN exon encoding TRD (Supplementary Data 7) in developing liver. Past research has shown that *ITGA2B* is alternatively spliced in melanoma, prostate cancer, and leukemia producing a truncated isoform lacking the transmembrane and cytoplasmic domain[39,40]. This truncated isoform, instead of integrating into the plasma membrane, is secreted into the extracellular matrix, unscrewing the adhesion, and promoting the migration of cells. Our analysis suggests that a similar mechanism is used in the case of OST complex, where the removal of TRDs would result in its dissociation from ER membrane, impeding the process of N-glycosylation of proteins.

Similar analysis for ND revealed that host genes of EN exons containing this domain in the brain were significantly enriched for the molecular functions related to GTPase activity and its regulators (Fig. 3b). Previous studies have implicated the role of nitrosylation modification in the upregulation of GTPase activity[41,42]. Our result thus suggests the role for alternatively spliced ND in the modulation GTPase activity during embryogenesis and cancer. In fact, few of the genes containing nitrosylation domain among brain EN exons, such as *RAB6A* and *RAB6B*, are GTPases belonging to RAS oncogene family, hinting at autoregulation of their GTPase activity through dynamic inclusion and exclusion of nitrosylation domain. Interestingly, one of the small GTPases, the *RHOA*, was previously shown to be inactivated through alternative splicing in diffuse-type gastric carcinoma cells[43]. We found that the exon involved in this splicing event (3rd exon) indeed encoded a ND and was embryonic negative (EN) in liver and kidney. This supports the broader role of the alternatively spliced ND in regulating the activity of the various small GTPases and cellular transport during development and cancer (Fig. 3d).

As for transmembrane domain, we obtained the genes having a ND among the EN exons in brain and identified the correspondence between the enriched molecular functions and biological processes (Fig. 3d). We observed that genes having GTPase activity were involved in Rab protein signal transduction and retrograde vesicle transport from endosomes to Golgi bodies to endoplasmic reticulum (Fig. 3d), the processes where GTPases are known to play a critical role[44,45]. Among the GTPases having a ND in their EN exons, *ARL1* gene had the greatest change in the inclusion of ND from pre-natal to post-natal stages and then in cancer (Fig. 3f). Our analysis thus suggests the underappreciated role of alternatively spliced ND in the regulation of the cytoplasmic transport by modulating the activity of GTPases. Additionally, some of the genes containing a ND among brain EN exons were enriched for the molecular functions related to BH-domain binding, death-domain binding and MAP-kinase signaling, which corresponded to the processes related to intrinsic apoptotic signaling pathways (Fig. 3d), potentially implicating exclusion of ND in modulating apoptosis[46].

Overall, our results implicate recapitulation of embryonic alternative splicing patterns of transmembrane and nitrosylation domains in several key oncogenic processes.

### Splicing regulatory model of EP events reveals key splicing factors dysregulated in cancer

Splicing factors (SF) control the choice and inclusion level of alternatively spliced exons[47]. To identify potential SFs regulating embryonic splicing, we trained a partial least squares regression (PLSR) model to predict the median inclusion level of EP events based

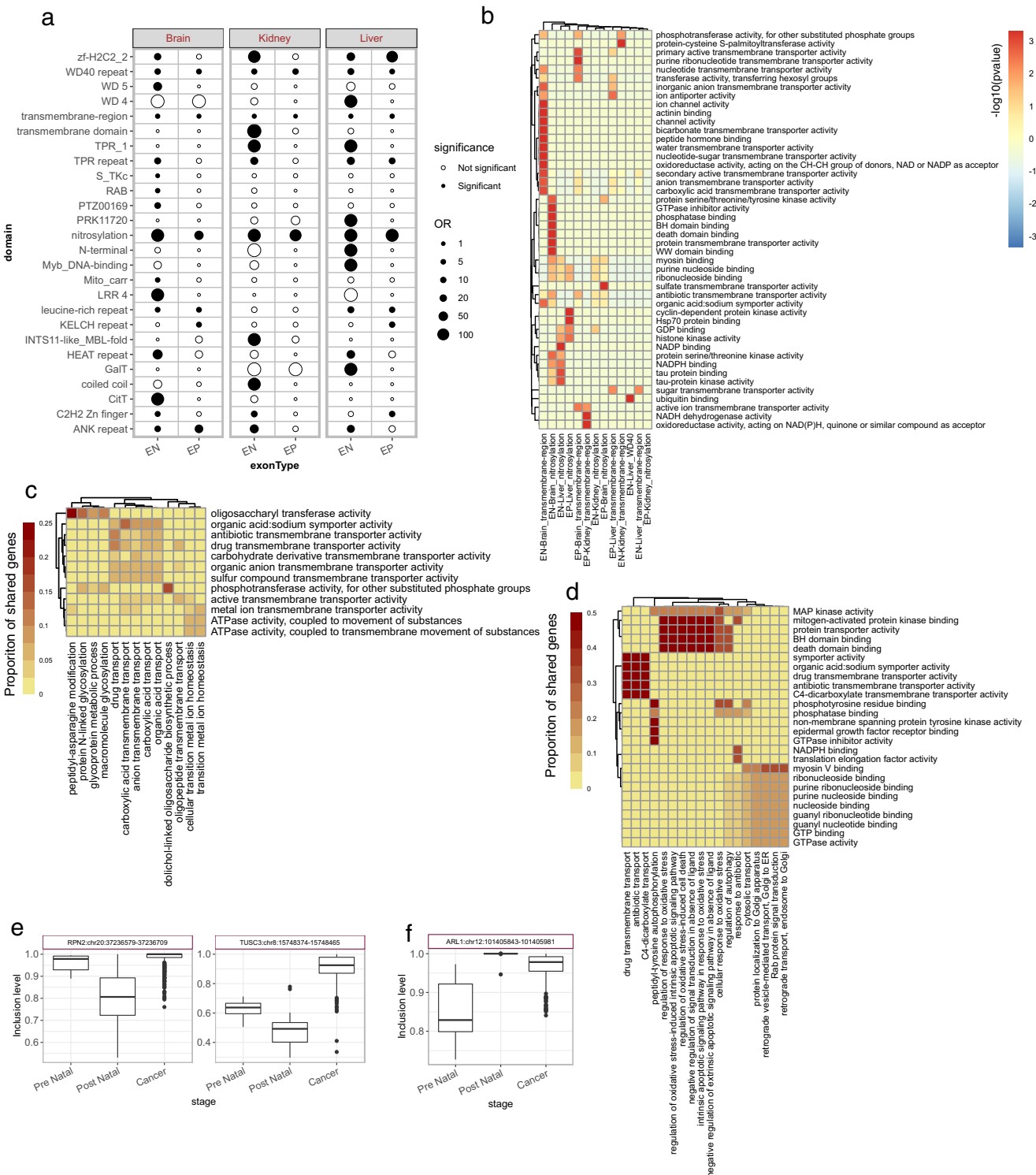

**Fig. 3 | Functional assessment of EP and EN exons. a** Dot plot showing the enrichment of domains in EP and EN events across three tissues. Size of the dots is scaled according to the magnitude of odds ratio calculated using Fisher's exact test; solid and hollow dots respectively indicate significant and non-significant domains based on FDR adjusted two-sided *p*-value threshold of 0.1. **b** Molecular functional enrichment across three organs for the host genes of EP and EN events containing nitrosylation, transmembrane-region and WD40 domains (indicated along the columns). The heat colors indicate −log10 of FDR adjusted one-sided *p*-value of enrichment from Fisher's test as implemented in clusterProfiler library in R. **c** Heatmap showing the cooccurrence of enriched biological process (columns) and molecular functions (rows) among the genes containing transmembrane-region

domain in EP exons in brain. **d** same as (**c**) but for nitrosylation domain in EN exons in brain. **e, f** The inclusion of EP exons encoding TRD among the subunits of OST complex (**e**) and EN exons with ND among the GTPases potentially involved in the regulation of vesicle transport (**f**). In **e, f**, *n* = 11 for pre-natal, *n* = 21 for post-natal, and *n* = 501 for cancer samples. In boxplots (**e, f**), the horizontal line in the middle is the median value and the lower and upper edges of the boxes correspond to the 25th and 75th percentiles. Extending vertically upwards/downwards of the boxes are the lines showing 1.5 times the interquartile range (i.e., distance between 25th and 75th percentile). Dots are the outliers. Source data for these figures are provided as a Source Data file.

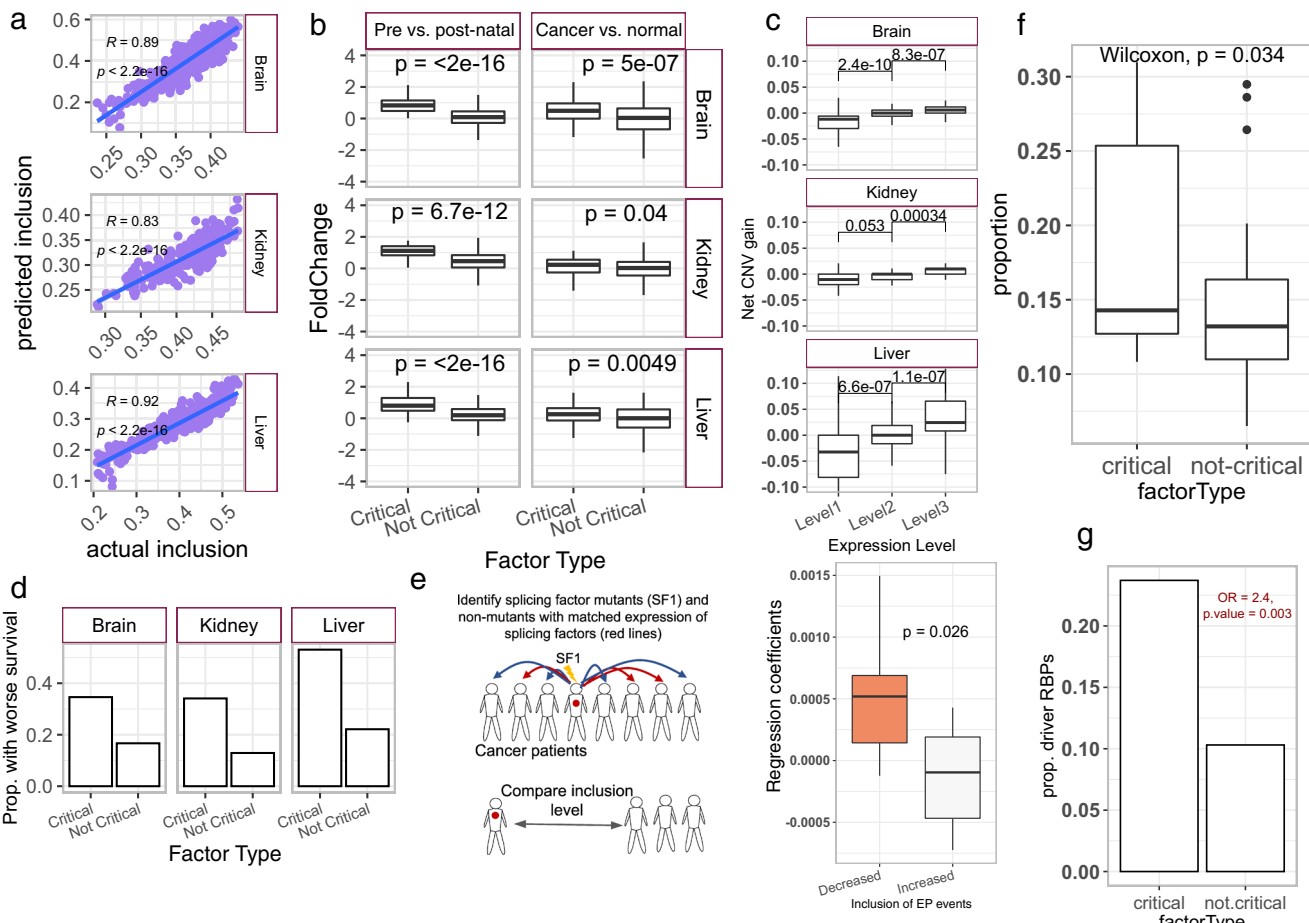

**Fig. 4 | Splicing regulatory model of EP events reveals key splicing factors dysregulated in cancer. a** Scatter plot of the actual and predicted median inclusion level of EP events across TCGA samples in a tissue-specific cohort. Blue lines depict the best fitting lines based on linear regression between actual and predicted median inclusion of EP events. Pearson's correlation coefficients and two-sided p-values are shown in the plots. **b** Boxplot distribution of fold-change of critical (n = 119 for brain, n = 167 for liver, and n = 45 for kidney) and non-critical (n = 322 for brain, n = 274 for liver, and n = 396 for kidney) splicing regulators of EP splicing events during development (left) and in cancer (right). The cancer types used in this analysis are LGG for brain, LIHC for liver, and KIRC for kidney. Two-sided p-values from Wilcoxon's test are shown. **c** Boxplots showing distributions of net CNVs gain ("Methods") in critical splicing factors in patients stratified based on the expression of the splicing factors (n = 100 for brain and liver and n = 45 for kidney). Two-sided p-values from Wilcoxon's test are shown. **d** Bar plots showing the proportion of critical and non-critical splicing factors which result in the poor prognosis of cancer patients in three cancer types. The odds ratio (OR) and FDR-adjusted two-sided p-values (pval) shown next to each plot are calculated using Fisher's exact test by comparing the proportion of critical and non-critical splicing factors having worse prognosis in cancer patients. Worse prognosis was defined based on >1 hazard ratio

in cox-regression at the FDR level of 0.3. **e** Schematic illustration of mutation analysis (left) and the distribution of the regression coefficients (y-axis) of splicing factors resulting in decrease (n = 14) or increase (n = 6) in the median inclusion level of EP events in the mutated samples compared to expression matched unmutated samples. Two-sided p-values from Wilcoxon's test are shown. **f** Boxplots showing the proportions of liver EP events which decrease in their inclusion (ΔPSI < −0.1) upon the shRNA knockdown of RNA binding proteins (n = 17 critical and n = 36 not-critical) in HepG2 cell line from ENCODE database. Single-sided p-value is derived from the Wilcoxon's test with the alternative hypothesis that deletion of CSFs affects a greater proportion of EP splicing events as compared to the deletion of non-critical splicing factors. **g** Proportion of critical and not-critical regulators of EP splicing among the RNA binding proteins taken from Seiler et al. 54 and known to harbor driver mutations in single or multiple cancer types. The odds ratio and two-sided p-value derived from this Fisher's test are shown. In boxplots (**b, c, e, f**), the horizonal line in the middle is the median value and the lower and upper edges of the boxes correspond to the 25th and 75th percentiles. Extending vertically upwards/downwards of the boxes are the lines showing 1.5 times the interquartile range (i.e., distance between 25th and 75th percentile). Dots are the outliers. Source data for these figures are provided as a Source Data file.

on the expression levels of 442 annotated SFs ("Methods", Supplementary Data 5). In each organ, trained solely on the developmental data, our model predicted the median inclusion level of EP events in independent tumor samples (TCGA) as well as normal samples (GTEx) with a high accuracy (average correlation between predicted and observed EP levels -0.88 for TCGA and 0.84 for GTEx; Fig. 4a and Supplementary Fig. 4a). Further, the predicted EP inclusion values can distinguish GTEx normal samples from their corresponding TCGA cancer samples with a high accuracy in brain and medium accuracy in liver and kidney (Supplementary Fig. 4b), underscoring that the model can predict the cancer-associated changes in the EP splicing.

Next, we obtained the list of splicing factors that were significant positive predictors of median EP splicing during embryonic development based on their regression coefficients in the PLSR model ("Methods") and termed those as critical splicing factors (CSFs, Supplementary Data 5). As expected, CSFs in each organ had higher expression during the prenatal stage of development and underwent significant upregulation in their corresponding cancer (Fig. 4b). Though our focus is only the positive regulators of EP splicing as those are upregulated in cancers relative to normal tissues, we confirmed that splicing factors with negative regression coefficients in the PLSR model undergo downregulation in cancers relative to the normal tissues and are potential negative regulators of the EP events

(Supplementary Fig. 4c). Further, the deletion of orthologous genes of brain CSFs results in defective nervous system development in mice, and CSFs from all three tissues are much more likely to result in pre-weaning lethality as compared to the other splicing factors (Supplementary Fig. 4d, Supplementary Data 6), further supporting the developmental role of CSFs.

We observe that cancer patients with higher expression of CSFs and correspondingly higher inclusion level of EP events have a significantly greater number of copy number amplifications in CSFs (Fig. 4c). In addition, a gain in CSF expression is significantly associated with worse patient survival in cancer (Fig. 4d).

To assess whether CSFs play a causal role in regulating EP events, we tested if the EP inclusion level is decreased in tumor samples bearing nonsense (inactivating) mutations in CSFs. We first identified all SFs whose mutant samples have lower and higher EP inclusion than the wildtype samples and found that potentially causal SFs (i.e., SFs whose mutant samples have lower EP inclusion relative to WT samples, Methods) have significantly higher (and positive) regression coefficients as compared to the other SFs in the PLSR model of EP splicing (Fig. 4e), establishing a potentially causal role of CSFs in regulation of EP events. We ensured that our results are not confounded by SFs expression differences between the mutant and wildtype samples ("Methods").

We further ascertained that PLSR can identify the causal factors underlying the inclusion of EP events by using shRNA knock-down followed by RNA-seq data for RNA binding proteins in HepG2 (liver cancer) cell line from ENCODE database[48]. Following an identical procedure as above, we learned the CSFs critical for the inclusion of liver-specific EP events in HepG2 cell line. We observed that knocking out these CSFs is much more likely than other splicing factors to decrease the inclusion of EP events ("Methods"; Fig. 4f), providing a strong support for the causal role of CSFs in EP splicing.

Some of the CSFs identified in developing human tissues are known drivers of various solid and hematological malignancies. For instance, *CDC5L* and *PCBP2* (CSFs in brain) are reported to promote the growth of gliomas[49,50] and bladder cancers[51]. Additionally, *SF3B1* (a CSF in kidney and liver) and *U2AF2* (CSF in liver) are frequent drivers of lung and pancreatic adenocarcinomas[52,53].

Besides the aforementioned examples, the pooled set of CSFs from all three tissues identified in our work was significantly enriched for 119 RNA binding proteins which were previously identified as the driver genes in one or more cancer types[54] (Fig. 4g, Methods). Further, a greater fraction of CSFs in brain, liver, and kidney had mutational hotspots in their corresponding cancers as compared to non-critical splicing factors (Supplementary Fig. 4f), further underscoring the role of CSFs in promoting malignancy.

Overall, these results reveal potentially causal SFs underlying the EP events and link the induction of such SFs, potentially via copy number amplification, to cancer. In the TCGA cancer samples, a median of 47%, 32%, and 16% of the CSFs were respectively upregulated (fold-change > 1.5) in brain, liver, and kidney cancers as compared to normal samples (Supplementary Fig. 4g). Considering this along with the mutational and shRNA analysis presented above, it appears that, although the deletion of a single CSF could have a small (albeit significant) effect on the inclusion level of a subset of EP exons, the broad reprogramming of splicing observed in cancers is achieved by activation of several CSFs, possibly driven by upstream transcription factors as we investigate in the next sections.

## Embryonic splicing events are associated with proliferation rates in cancer cell lines

Our results above (Figs. 1b, 2a) suggest that increased inclusion of EP events in tumors might be involved in mediating oncogenic processes such as rapid proliferation, EMT, and angiogenesis. Leveraging the DepMap database (https://depmap.org/portal/) that includes RNA-seq data and proliferation rates in multiple cancer cell lines, we find that in liver and brain, there is a negative (respectively positive) association between the doubling time and the median EP (respectively EN) inclusion levels across cell lines derived from the organ-specific cancer type (Supplementary Fig. 5a), hinting at a possible link between EP/EN usage and proliferation rate of cancer cell lines.

To further consolidate this link, we calculated the proportion of EP and EN exons among all the splicing events that were strongly correlated with the doubling time of cancer cell lines ("Methods"). We observed that the brain and liver EP and EN events were strongly enriched among the exons which were respectively negatively (PCC < −0.5) and positively (PCC > 0.5) correlated with the doubling time of their corresponding cell lines in CCLE data (Fig. 5a, Supplementary Fig. 5a). This enrichment implies that exons linked with proliferation rates of cancer cell lines were more likely to be embryonic in nature. The lack of association between embryonic events and doubling times of cancer cell lines for kidney could be the result of heterogeneity as discussed below.

Further, splicing regulatory models learned from the developmental data could accurately predict the EP event inclusion in the corresponding cell lines (Fig. 5b). Collectively, these observations further validate the links between CSFs and proliferation, mediated by EP events. Given the links between CSF activity and proliferation, we expect that inactivation (by CRISPR or RNAi) of the CSFs will have an adverse effect on the proliferation rates of the cell lines. Indeed, we found that in liver cancer-derived cell lines, the more critical a SF (based on PLSR coefficient), the greater was the dependency of the cell line on that SF (negative dependency scores, Supplementary Fig. 5b), supporting a functional role for CSFs; however, we did not see this trend in brain and kidney, as discussed below. Further supporting the role of CSFs in malignant transformation, we found that in the single-cell transcriptome of liver and brain ("Methods") tumor micro-environment, CSFs were specifically expressed in the malignant but not in non-malignant cells (Fig. 5c). Collectively, these observations link the role of CSFs in tumor cells with cellular proliferation rates through regulation of specific AS events, which might serve as potential therapeutic targets.

## CSFs are potentially regulated by MYC, FOX, and BRD family transcription factors

Next, we investigated potential upstream transcriptional regulators of CSFs, as targeting them may have a broader effect on CSFs, with the resulting changes in EP inclusion potentially improving patient prognosis. We applied four criteria to identify high-confidence upstream transcriptional regulators of CSFs (Fig. 6a). First, as an initial filtering step, we utilized a large collection of ChIP-seq datasets across multiple cell lines curated in the TFEA.ChIP database[55] and shortlisted TFs whose binding was significantly enriched within the promoter regions of CSF as compared to non-critical splicing factors (nCSFs) of EP events (first column in Fig. 6b; "Methods"). Next, we used the KnockTF database[56], which details transcriptome changes upon TF deletion, to calculate the enrichment of CSFs relative to nCSFs among the down-regulated targets following TF deletion and retained significant hits (second column in Fig. 6b, Methods). A major limitation of KnockTF is low coverage of TFs. We therefore applied two additional computational approaches to filter the TFs shortlisted based on TFEA.ChIP.

First, for each factor shortlisted based on TFEA.ChIP, we inferred its in-silico targets using the ARACNe software tool[57] and selected TFs whose in-silico targets were more significantly enriched for CSFs relative to nCSFs (third column in Fig. 6b, Methods). Secondly, among the list of ChIP-seq filtered regulators, we identified TFs whose expression was more strongly correlated with CSFs as compared to the nCSFs in cancer transcriptomic data (fourth column in Fig. 6b, Methods). Overall, we retain in each organ, the TFs that (after the ChIP-seq-based filtering) either qualified the experimental KnockTF-based

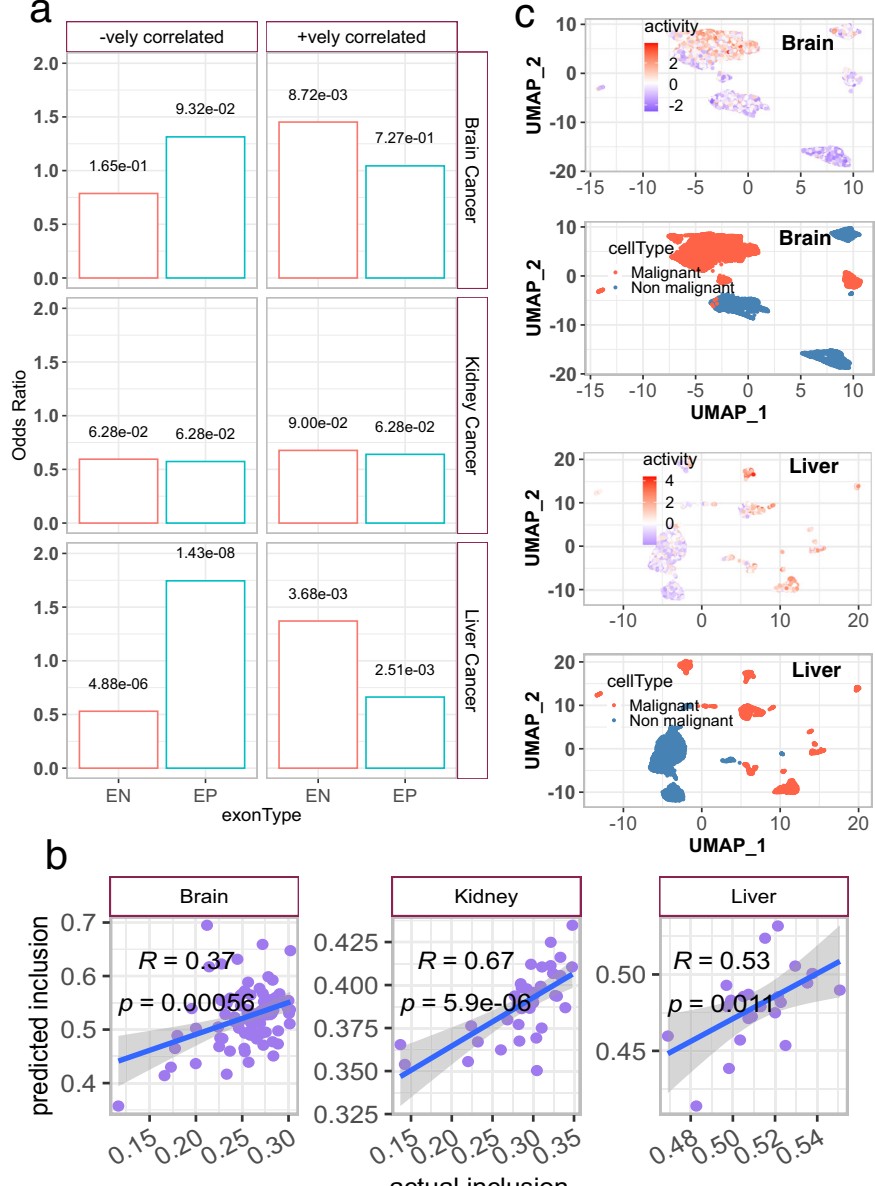

**Fig. 5 | Embryonic splicing events and their regulators in cancer cell lines. a** Bar plots showing the odds ratio for enrichment/depletion of embryonic positive (EP) and embryonic negative (EN) events among the exons having strong positive (+vely) and negative (−vely) correlation with the doubling time of cancer cell lines (obtained from DepMap portal) corresponding to brain, kidney, and liver. FDR-adjusted two-sided *p*-values obtained from the Fishers' test are shown next to each bar. **b** Scatter plot of the observed and predicted median inclusion level of EP events in brain, kidney, and liver cancer cell lines from CCLE. Blue lines and shaded grey areas depict the best-fitting lines and 95% confidence intervals based on a linear regression between actual and predicted median inclusion of EP events. Pearson's correlation coefficients and two-sided *p*-values are shown in the plots. **c** UMAP showing the activity of critical splicing factors in the malignant and non-malignant cells of the tumor microenvironment; top row: activity, bottom row: cell types. Source data for these figures are provided as a Source Data file.

criterion or both of the computational filters. Collectively, these results implicate MYC, FOX (specifically *FOXM1*), and BRD family of TFs in the regulation of EP events through upregulation of CSFs and may represent master regulators of broad splicing changes associated with development and cancer. Such master regulators and key CSFs of EP splicing could be plausible druggable targets (Supplementary Methods & Supplementary Table 1) to halt cancer progression by impeding the processes mediated by EP splicing events.

## Discussion

The availability of transcriptomic datasets of tumors from TCGA and PCAWG consortia have facilitated the genome-wide analysis of alternative splicing changes in cancer elucidating their prognostic value[58],

genetic basis[59,60], and the discovery of tumor neoantigens generated by alternative splicing[61]. However, none of these studies analyzed the broader developmental context of splicing changes in cancer. Leveraging recently available temporal developmental transcriptomic data in three human organs, in this work, we have shown that the genome-wide splicing landscape of cancers significantly reverts to the early embryonic developmental stage of their tissue of origin, strongly implicating developmental splicing events in oncogenesis and tumor progression.

Similar to gene co-expression modules, inclusion of multiple exons across genes is coordinated to affect specific cellular functions during differentiation[62,63], cell state transition[64], apoptosis[65], and hormonal induction[66]. Our results suggest that coordinated programs of

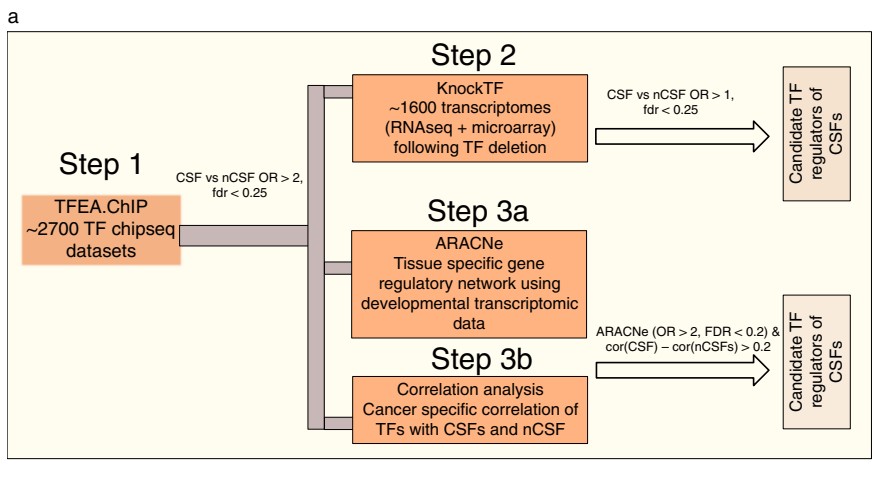

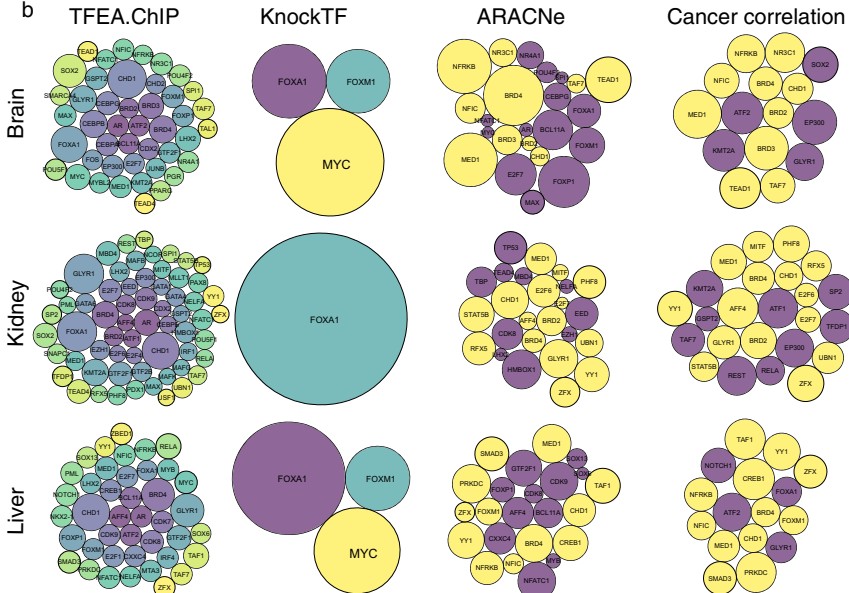

**Fig. 6 | Potential TFs regulating CSFs in three organs. a** Schematic representation of steps involved in the detection of TF regulators of CSFs. **b** Three rows correspond to three different tissues and four columns are different strategies that were used to infer TF regulators of CSFs, as labeled on the top of figures. In first three columns, bubble sizes correspond to −log10 of FDR-adjusted two-sided *p*-values. In the fourth column, bubble size corresponds to the difference between correlation coefficient of TF with median expression of CSFs and nCSFs in relevant cancer types. In the first two columns, the bubbles are colored by TF names. In the last two columns, yellow bubbles indicate the evidence in support of TF by both ARACNe and Cancer correlation analysis and magenta color indicates support by either one. Bubble sizes are not comparable across columns. Source data for these figures are provided as a Source Data file.

EP and EN splicing events are involved in embryonic processes, such as cellular proliferation, apoptosis, EMT, migration, and that cancers seem to misappropriate these coordinated exon inclusion events to revert to an embryonic-like state. Evolutionary comparisons in past have shown that alternative splicing results in neo-functionalization and increases proteome complexity of genes[67,68] which is often driven by the divergence of exonic structure of genes. Changes in exonic structure of genes have been observed in cancers as well *via* mutations that create splice sites[69]. Therefore, we speculate that alternative splicing can promote carcinogenesis through two distinct routes, either through the re-activation of multiple aspects of the embryonic physiology, or by fueling the functional novelties and proteome complexity driven by creation of new splicing events, or a combination thereof.

Further, EP and EN events ascertained based on developmental context alone are significantly prognostic in the corresponding cancers in TCGA. For instance, the inclusion level of EP and EN events respectively predicted worse and better survival of cancer patients, underscoring the value in studying fetal development to better understand cancer mechanisms. Moreover, the enrichment of the EP exons among the splicing events which had a negative correlation with doubling time (equivalently, positive correlation with proliferation rate) of brain and liver cancer cell lines provides an independent functional validation for the role of these splicing events in mediating cellular proliferation, which is relevant to both development and cancer. However, these associations do not hold true for the case of kidney cancer cell lines. While cell lines are standard choice to model several diseases, they do not entirely capture the in vivo complexity. In our analysis, although we derived the EP and EN exons from the developing human embryos and yet, rapidly proliferating CCLE cancer cell lines indeed have higher usage of EP exons and lower usage of EN exons in brain and liver, suggesting a conserved cell-intrinsic links between splicing and proliferation.

Previous research has shown that AS can affect cytoskeleton, enzymatic properties, and membrane localization of proteins[70]. Here we observe that the molecular functions related to cytoskeleton binding and regulation of GTPase activity and cellular transport were highly enriched among EP and EN exons across all three organs we

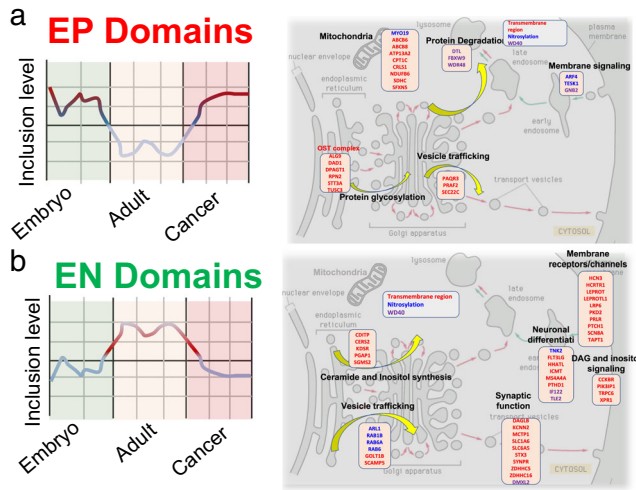

**Fig. 7 | Coordinated splicing in brain development and cancer.** This schematic shows the proposed role of coordinately spliced TRD, ND, and WD40 domain among EP and EN events in mediating the N-glycosylation and retrograde transport functions. **a** *Left panel*, inclusion level of EP domains in different developmental and pathological stages. *Right panel*, the host genes of EP events are enriched for oligosaccharyltransferase (OST) complex, vesicle trafficking, and mitochondria (transmembrane-region domain), protein degradation (WD40 domain), and membrane signaling (nitrosylation domain). These processes suggest active protein synthesis, processing including N-glycosylation, and energy metabolism during neural cell development. **b** *Left panel*, inclusion level of EN domains in different developmental and pathological stages. *Right panel*, the host genes of EN events are enriched for vesicle trafficking (nitrosylation domain) and neuronal function, including ceramide/inositol synthesis, synaptic function, and membrane receptors/channels/signaling (transmembrane-region domains). Most of the genes have function required for mature neural cells (e.g., neural transmission and synaptic signaling).

studied (Supplementary Fig. 3a). These molecular functions are central to cellular proliferation through regulation of cell cycle[71–75] and cellular migration[76–78] and, consequently, have emerged as important players in cancer progression and metastasis[35,79].

The analysis of protein domains enriched among EP and EN events further suggested their functional coordination in regulating diverse cellular processes such as proliferation, migration, neuronal physiology, and stress resistance. For instance, proliferation and migration of cells relies on alterations in the cytoskeleton, extracellular matrix, and cell adhesion, which are modulated by N-glycosylation of proteins like actin, cadherins and integrins[38]. Our observation that subunits of OST (including *TUSC3* and *RPN2*) undergo coordinated splicing of their TRDs among EP events suggests the role of this splicing in the regulation of N-glycosylation during organogenesis (Fig. 3c). Further TRDs in vesicle trafficking (*PAQR3*, *PRAF2*, *SEC22*) and mitochondria (*ABCB6*, *ABCB8*, and *SDHC*), WD40 in E3 ligase involved in protein degradation (*DTL*, *FBXW9*, *WDR48*), and NDs in GTPases for membrane signaling (*ARF4*, *TESK1*) were coordinately spliced among the brain EP exons. This suggests the functional coordination in energy metabolism and protein synthesis/processing during neuronal development is, in part, mediated via alternative splicing. In accordance, knocking down CSFs that regulate brain EP events result in the defects in the nervous system development in mice (Supplementary Fig. 4d), supporting the essential role of coordinately spliced EP events in organ development.

The EP and EN-mediated functional coordination is further illustrated in the case of coordinately spliced protein domains among the EN events in brain (Fig. 7b). Coordinately spliced ND in the GTPases involved in vesicle trafficking (*ARL1* and *RAB* family genes), TRD in the endoplasmic reticulum-associated proteins involved in the ceramide/inositol synthesis (*CDITP*, *CERS2*, *KDSR*, etc.) and synaptic proteins

involved in neuronal signaling (*DAGLB*, *KCNN2*, *MCTP1*, etc.) suggests the coordination in post-natal neuronal function such as setting up and firing rapid action potentials (Fig. 7b, right panel) and loss thereof in cancer (Fig. 7b and Supplementary Fig. 6b).

Moreover, for a vast majority of the EP domains, their inclusion level, which is higher in pre-natal stages, switches back to pre-natal stages in cancer (Fig. 7a and Supplementary Fig. 6a). This suggests that host genes containing these domains drive cancer progression and EP exons of these genes can be potential therapeutic anti-cancer targets.

We note that most of the protein domains are enriched among EN exons (Fig. 3a), implying that a relatively larger fraction of annotated domains is involved in processes that are active postnatally. A general bias in functional roles of alternatively spliced domains to be involved in development-related functions has been noted previously[80] but the differential functional underpinnings of this observation relative to EP and EN are currently unclear and will require further investigation.

Many of the EP/EN events are previously reported and experimentally validated to be alternatively spliced in various diseases including cancer (Supplementary Data 8), For instance, *APAF1* gene encodes an apoptotic protein and hosts an EN exon encoding WD40 domain in developing brain. Interestingly, a previous report has shown that *APAF1* is alternatively spliced in prostate cancer cell lines, producing a shorter isoform called APAF1-ALT lacking WD40 domain[81]. Moreover, this shorter isoform impeded the induction of DNA-damage-induced apoptosis in cells, thereby allowing cells to acquire DNA-damage-induced resistance against treatment. Thus, the change in the apoptotic roles of *APAF1* via alternatively spliced WD40 domain appears to be general mechanisms employed during embryogenesis as well as cancer. Additionally, the gene *FLVCR1* encodes a heme transporter and hosts an EN exon encoding TRD domain in brain. Previous work has shown that various alternatively spliced isoforms of this gene lacking the TRD are expressed in the case of Diamond Blackfan anemia (DBA). Importantly, the patients with DBA have an elevated risk of neoplastic growth[82]. This example implies that the regulation of iron metabolism by controlling its transport by alternatively spliced *FLVCR1* gene could be a crucial mechanisms to regulate iron levels in developing human brains[83] as well as cancers. The truncated isoform of integrin *ITGA2B* lacking the transmembrane domain is another example, which was previously reported to be secreted into the ECM in various cancers, breaking adhesion and facilitating cell migration[40].

Further, many of the tetraspanins, which are scaffolding proteins present at the membrane of the cell, and mediate various cellular functions such as proliferation, adhesion and signaling[84] contained an EP or EN exon encoding TRDs across the tissues (Supplementary Data 7). Alternatively spliced TRD in these proteins are reported to generate isoforms having alterations in the tetraspanin-enriched microdomain functions, which includes cell signaling and cell adhesion[84,85]. These examples (and Supplementary Data 8) support that the EP and EN events can indeed change the function of proteins and contribute to the broad functional convergence observed between embryogenesis and cancer.

Several single-cell RNA-seq studies in the recent past have noted a general similarity in development and cancer[86–88]. Therefore, our results indicate that these similarities are hinged upon a much broader and coordinated reprogramming of splicing in cancer cells back to their embryonic counterparts.

Critical splicing factors, which were inferred to regulate the inclusion of EP events based solely on the embryonic developmental data, are upregulated in cancer, and confer poor prognosis to the patients. Furthermore, inactivating mutations of critical splicing factors in cancer patients and shRNA knock-down in HepG2 cell line result in the decreased level of embryonic splicing, strongly supporting their causal role in regulating embryonic splicing events. The causal role of CSFs also supported by their significantly higher, and experimentally quantified, dependency scores in the DepMap dataset in

liver cancer-derived cell lines. However, we did not see this trend in the brain and kidney, which may be attributed to divergent physiology and regulatory networks in cell lines as compared to tumors in the context of the tumor microenvironment. Although we observed that significant numbers of CSFs were drivers in multiple cancer types (Fig. 4g and Supplementary Fig. 4e), we did not observe a progressive increase in the mutation load (defined as total no. of mis-sense mutations per sample) among CSFs in the late-stage cancers as compared to the early stages (Supplementary Fig. 4h). This suggests that the mutational perturbation of CSFs and the corresponding change in splicing profile is involved in tumor initiation; however, it is not clear if different sets of CSFs are involved in initiation and progression of cancer and will require longitudinal data. Notably, consistent with the previous reports implicating the role of alternative splicing in the regulation of splicing factors[89], we too observed that a significant fraction of splicing factors hosted an EP and EN events, with CSFs having relatively higher proportion of EP and EN events (Supplementary Fig. 4i). Therefore, we speculate that the action of CSFs in promoting malignancy is mediated through their specific isoforms requiring in depth investigation in future.

Together, these observations highlight the inferred critical splicing factors as potential therapeutic targets against cancer progression.

Further we found that CSFs in each developing organ, as well as the corresponding cancer, were likely regulated by FOX (*FOXM1*), MYC, and BRD family of transcriptional regulators. Regulation of splicing factors and splicing events by *MYC* has been previously noted[31,90]. A recent report shows that *MYC*-driven splicing factors regulate ~4000 splicing events across cancers[91]. Consistent with our findings, FOX and the MYC family of regulators control growth, proliferation, and survival of cells in multiple contexts during embryogenesis as well as cancer[92,93]. Our work extends the previous studies by showing the regulation of splicing factors and functionally coordinated embryonic splicing events by MYC, BRD, FOX family of TFs in the developmental context, thus providing further mechanistic links between development and cancer. These observations hints that the embryonic reversal of cancer splicing drives cancer in conjunction with much broader transcription and epigenetic reprogramming mediated via perturbations in various master regulators (such as *MYC* and *FOXM1*) as well as critical splicing factors.

Although gene regulation is best studied experimentally using gene knockouts followed by RNA-seq experiments to reconstruct transcriptome-wide gene regulatory networks[94], such datasets do not always exist for desired transcription factors in every cell line/model system in humans. In our analysis presented in Fig. 6, we have used KnockTF, which is one such database, along with three other computational filters to identify the key master regulators of CSFs (Fig. 6b, 2nd column). Our results suggest that the broad changes in the expressed isoforms of key genes driven by the upregulation of CSFs is likely a major mechanism by which these TFs exert their physiological effects. Therefore, targeting the upstream regulators of CSFs might result in broader changes in genome-wide splicing and improve the survival rates of patients. But such an approach is likely to suffer from unintended side effects owing to the lack of specificity and pleiotropic nature of transcription factors. Therefore, direct targeting of EP exons, through recently developed CRISPR-based techniques[95,96], as opposed to their upstream regulators, might result in specific lethality in the tumor cells. In the future, transcriptomic experiments following the deletion of CSFs or their upstream regulators would be necessary to establish the proposed mechanistic links and explore their therapeutic potential.

Collectively, our multi-pronged investigation not just conceptually enhances the understanding of broad functional roles and regulation of alternative splicing in the context of development and cancer, but also suggests putative cancer therapeutic targets. Our work also provides a framework to study the cellular mechanisms implicated in development and cancer using other molecular modalities such as miRNA and lncRNA activities, DNA methylation and histone modification profiles, alternative promoter, and poly-A usage.

## Methods

### Datasets and quantification of exon inclusion

For brain, liver, and kidney, uniformly processed RNA-seq data for tumors from TCGA (https://www.cancer.gov/tcga) and normal samples from GTEx[97] were downloaded from the UCSC-Xena browser (data version V7). We used UCSC-Xena browser[98,99] as it hosts the datasets from UCSC toil RNA-seq recompute compendium[100] which were normalized for multiple computational as well as within cohort batch effects. The UCSCXenaTools library in R[98] was used to download transcript-level TPM values computed using Kallisto[101]; the details of data integration and processing can be obtained from UCSC-Xena browser (https://xenabrowser.net/). In total, we obtained the expression levels of 197,046 transcripts across all samples. The number of samples obtained are brain cancer – Lower grade glioma (LGG): 523; Glioblastoma GBM: 172, normal brain – brain cerebellum: 118; brain cortex: 107, liver cancer – Liver hepatocellular carcinoma (LIHC): 369, normal liver –110, kidney cancer – Kidney renal papillary cell carcinoma (KIRP): 321; Kidney renal cell carcinoma (KIRC) 595, normal kidney – 27. For developmental data[29], we obtained the raw reads from the array express using the accession number E-MTAB-6814 and computed the transcript level TPM values using Kallisto[101] and the transcriptome index based on Gencode version v23 (https://www.gencodegenes.org/human/release_23.html) annotations, the same version which was used by UCSC-Xena. We used pseudoalignments based approach using Kallisto software to process RNA-seq datasets as it is much faster than classical alignment[101,102], and estimated TPMs showed very high concordance with RT-PCR-based measurements[103,104]. The data includes multiple pre-natal and post-natal time points in each organ (Supplementary Data 1). To quantify the inclusion level of exons in each sample, we calculated the 'percent-spliced-in' (PSI) value for each exon, which ranges from 0-1 (i.e. from fully excluded to fully included), using SUPPA-2[105]. We choose SUPPA2 as it enabled us to directly use the elegant datasets from UCSC Toil RNA-seq recompute compendium[100] hosted at toilhub of UCSC-Xena browser[98], ensuring uniform processing and normalization of batch effects, Additionally SUPPA2 is much faster than most other tools and requires lesser storage space as it can use pre-computed TPM values[105]. Further, we validated our main conclusion, namely, reversal of splicing events in cancer to pre-natal state of the corresponding tissue, using an entirely different pipeline – STAR 2 pass alignment[106] followed by rMATs[107] (Supplementary note 2). Transcript-level TPMs were converted to gene-level TPMs and subsequently quantile normalized as needed for the follow-up analyses. All the scripts used for downloading and processing the RNA-seq datasets are available in at https://github.com/hannenhalli-lab/AltSplDevCancer.

### Developmental splicing events

To identify splicing events deemed to be involved in embryonic development, we adapted a previously published strategy called PEGASAS[31]. PEGASAS identifies the alternative splicing events that correlate with the activity of a specific biological pathway. In this study, we identified developmental exons via a three-step process as follows.

**Step 1:** We scored the activity of each of the 332 KEGG pathways[30] at each time point during development using the median of log-transformed expression of its constituent genes, resulting in a 332 × N activity matrix, where N is the number of developmental time points that were sampled for each tissue and are given in Supplementary Data 1. Clustering this activity matrix reveals two broad clusters

(Supplementary Fig. 1a)—one active pre-natally and the other active post-natally.

**Step 2:** We applied an additional smoothing procedure in PCA space where our goal was to quantify each pathway's tendency to be preferentially oriented towards a specific developmental timepoint. In 5-dimensional PC space (first 5 PCs explain ~65% of variance), each timepoint occupies a unique coordinate based on the PC scores. In this space, similarly, each pathway corresponds to 5-dimensional vector of the pathway's loading in each of the 5 PCs. We quantify the preferential orientation of a pathway toward a specific timepoint as cosine similarity between the loading vector and the location of the time point in the 5-dimensional space. This procedure yields a smoothed 332 × N matrix clearly segregating 332 pathways into two broad groups based on their preferential activity during pre- or post-natal stages of development (Fig. 1a). The grouped pathways were correspondingly called embryonic positive and embryonic negative pathways.

**Step 3:** Next, we used an approach similar to PEGASAS[31] and computed the cross-sample Pearson's correlation coefficient (PCC) between the PSI value of each exon and pathway activity score in Step 1. For each exon, we selected the significantly positively or negatively correlated KEGG pathways correcting for 332 tests performed for each exon based on the Benjamini-Hochberg FDR threshold of 0.05. We call an exon embryonic positive (EP) if it is significantly correlated with at least 10% of the embryonic positive pathways vs. at most 5% of the embryonic negative pathways. Analogous criteria were applied to define embryonic negative (EN) exons.

The PEGASAS-based approach is superior in detecting the splicing events relevant to embryonic development of tissues compared to simply performing differential splicing between pre- and post-natal stages of development because (i) the sample size of the developmental dataset is insufficient for a robust differential inclusion analysis, (ii) an individual exon's inclusion can be highly variable within pre- and post-natal stages, which can confound the identification of embryonic splicing events using differential analysis, (iii) since the PEGASAS approach is anchored on robustly identified embryonic positive and negative pathways, instead of relying only on an individual event's temporal dynamics, it is likely more robust to noise. In Supplementary note 1, we provide a detailed discussion of relative advantages of PEGASAS approach compared with the conventional differential inclusion analysis.

## Cancer-specific splicing events

For each exon skipping event identified by SUPPA2, we performed a tumor-normal comparison of its PSI value to identify the splicing events which were differentially included in tumors. Owing to the transcriptomic heterogeneity across tumors, a standard differential splicing analysis, which assesses the significance of difference in the median PSI values of cancer and normal samples, will not detect exons mis-spliced in a small number of tumors, which can nevertheless be biologically significant[61]. Therefore, we selected the events which were at least 2 standard deviations away from the mean of their distribution in the corresponding GTEx normal samples in a consistent direction (i.e., increased or decreased) in at least 15% of the cancer patients. (Fig. 1a). Correspondingly, such events were termed as frequently increased or decreased in cancer. We focused only on exon skipping events as those are better annotated in transcriptional databases and are easier to interpret functionally.

## Comparison of cancer and developmental splicing and functional enrichment analysis

To assess if cancer recapitulates embryonic splicing events, we assessed the significance of overlap between cancer and developmental splicing events using Fisher's exact test and adjusted the *P*-value using Benjamini-Hochberg's FDR method. Functional enrichment analysis was performed using the clusterProfiler library in R and the p-values of

the resulting significant terms were adjusted with Benjamini-Hochberg's method. For plotting, the resulting GO terms were simplified based on their semantic similarity using the 'simplify' function from clusterProfiler in R (similarity threshold of 0.7).

## Protein domain enrichment in EP and EN exons

We downloaded the transcriptomic coordinates of all the PFAM domains that were mapped to the reference genome (hg38) from the prot2hg database (http://www.prot2hg.com)[108]. Since any given domain can be incorporated either fully or partially in multiple transcripts, the downloaded file was preprocessed to remove redundancy of genomic coordinates resulting from the same domain mapping to multiple transcripts by using bedtools merge[109]. We then intersected the preprocessed genomic coordinates of protein domains to the unique and non-overlapping set of EP and EN exons as well as the rest of the alternatively spliced skip exons (called background exons) using bedtools intersect in each tissue. To identify the domains enriched in EP and EN events, we computed the frequency of occurrence of each domain in EP, EN, and background exons and performed a Fishers' test of enrichment in each tissue. The resulting p-values from the Fishers' test were corrected for multiple testing by using Benjamini-Hochberg's method and the domains with an odds ratio > 1 and corrected *p*-value <0.1 were considered enriched among EP or EN exons in each tissue.

## Survival analysis

We used clinical data from TCGA to model the overall survival of cancer patients using the inclusion level (PSI value) of each exon as a predictor variable and age as a covariate in the cox regression. We used the R library "survival" for this analysis and the resulting p-values were adjusted for multiple testing using Benjamini-Hochberg's method. The distribution of the resulting hazard ratios was compared between embryonic positive, negative and the rest of the splicing events.

## Model for regulation of embryonic splicing

To dissect the potential regulators of embryonic splicing events, we built upon a commonly used notion that differential expression of splicing factors could lead to the differential splicing of the exons[110]. For this, we identified 442 proteins which have the term 'splicing' in their GO definition from the Amigo database[111]. We then used a partial least square regression (PLSR) analysis to model the inclusion of EP events using the gene expression of splicing factors in the developmental data. PLSR outperforms multiple linear regression when dealing with multicollinearity among the predictor variables or when the predictor matrix is non-singular[112].

For gene expression matrix X of 442 features (SFs) across N developmental timepoints ($n \times 442$) and response matrix Y of median EP splicing across n timepoints ($n \times 1$), the PLSR transforms X and Y as per the following relations:

$$\mathbf{X} = \mathbf{T}\mathbf{P}^{T} + \mathbf{E} \tag{1}$$

$$\mathbf{Y} = \mathbf{U}\mathbf{Q}^{T} + \mathbf{F} \tag{2}$$

where **T** and **U** are the N × r matrices of the extracted latent vectors and **P** (p × r) and **Q** (1 × r) are the loadings of **X** and **Y**. **E** (n × p) and **F** (1 × p) are the residuals. In the PLSR algorithm, **T** and **U** are constrained to have a maximum covariance as per following relation:

$$\mathbf{U} = \mathbf{T}\mathbf{B} + \mathbf{H} \tag{3}$$

where **B** (r × r) is a diagonal matrix of regression coefficients and **H** is a matrix of residuals.

Splicing factors with positive regression coefficients and a significant *p*-value ($p < 0.05$ after FDR correction) were considered critical

regulators of EP events (CSF) PLSR was implemented using 'pls' package in R[112].

## Mutation analysis of splicing factors

To assess the causal role of CSFs in the regulation of EP events, we obtained level 2 mutation data from TCGA cohorts of brain, liver, and kidney cancers (https://portal.gdc.cancer.gov/) using 'maftools' in R[113] and identified the tumors which had nonsense or truncating mutations for these factors. For each mutated factor in each cancer type, we compared the median inclusion level of EP events in the mutant samples against the background set of samples that were not mutated for any of the splicing factors. Thus, the factors were classified into 'increased' or 'decreased' categories depending upon at least 5% increase or decrease in the median EP inclusion level. To account for the potential confounding effect of the differential expression of splicing factors between samples, we identified, for each mutant sample, a set of 10 non-mutant samples with similar splicing factor expression. Specifically, for each mutant sample, we identified 10 non-mutant samples with the shortest Euclidean distance to the mutant sample in terms of the gene expression of all splicing factors. For robustness, we discarded the splicing factors for which the background set of patients had a high variability (standard deviation > 0.1) in the median EP splicing across the 10 samples (Supplementary Fig. 4e).

## Transcriptional regulators of splicing factors

To identify the potential transcriptional regulators of critical splicing factors (Fig. 6a), in each organ independently, we divided the splicing factors into two classes: namely, a foreground set comprising of the top 100 critical splicing factors, and a background set comprising the remaining splicing factors (nCSFs). To assess whether a TF was more likely to regulate CSFs compared to nCSFs, we used four complementary approaches (Fig. 6a). In the first step, we used the TFEA.-ChIP library in R, which uses publicly available genome-wide binding datasets from ChIP-seq experiments[55]. TFEA.ChIP used a Fisher's test to assess if a specific TF's binding is significantly enriched in the promoter regions (i.e., within 1 kb upstream of the transcription start site) of the CSFs relative to nCSFs (step 1 in Fig. 6a). TFs with an odds ratio > 2 and an FDR of 0.05 were considered putative regulators of CSFs. This first step was used as a strict filter for a TF to be further considered. To validate the ChIP-seq-based findings with the gene knockout/knockdown studies, we used the KnockTF database[56], which is a compendium of publicly available genome-wide transcriptional profiling following the deletion of TFs across multiple cell lines (step 2 in Fig. 6a). In this step, we obtained all the genes which were marked as downregulated based on a robust statistical analysis in the KnockTF database[56] following the deletion of a transcription factor and again assessed if CSFs were enriched as compared to nCSFs among the downregulated targets using a Fisher's test. TFs with an FDR of <0.25 and a positive odds ratio in any of the cell lines were considered putative experimentally derived regulators of CSFs. Furthermore, because KnockTF has a poor coverage of TFs, we did not use this as a strict filter and instead used two additional computational approaches to infer the potential TFs: (i) We built a gene regulatory network for TFs shortlisted by ChIP-seq using the developmental time course data for relevant tissues and the ARACNe software[57] and assessed if the CSFs were enriched relative to nCSFs among the in silico derived targets of each TF using a Fisher's test (step 3a in Fig. 6a). TFs with an odds ratio > 2 and an FDR < 0.2 were considered potential in silico derived regulators of CSFs. (ii) In parallel, we assessed the correlation of ChIP-seq shortlisted factors with CSFs and nCSFs in relevant cancer types (step 3b in Fig. 6a). The factors, with a correlation difference > 0.2 between CSFs and nCSFs were considered putative regulators. The ChIP-seq shortlisted factors, which either passed the KnockTF test OR passed both computational tests, were proposed as regulators of CSFs. In all the applicable cases, p-values were adjusted for multiple comparisons using the Benjamini-Hochberg procedure in R.

## Analysis of shRNA data for HepG2 cell line

To investigate the effect of knocking down of CSF on the inclusion level of EP events, we used an shRNA knockdown data for RNA binding proteins in HepG2 (a liver cancer) cell line from ENCODE database[48]. The dataset consisted of RNA-seq experiments following the knockdown of 223 RNA binding proteins, each with two biological replicates, and controls which were shared between different targets. The raw sequencing reads for the knockdown as well control experiments (26 controls with two replicates each) were downloaded and processed to quantify transcript/gene expression using Kallisto and exon inclusion using SUPPA2. Following a similar procedure as before (i.e., EP events in human tissues), the gene expression and splicing quantification in the control set of cell lines were used to train a PLSR model and learn the critical splicing factors of liver EP events in HepG2 cell line. We considered only those splicing factors in which shRNA knockdown resulted in at-least 50% reduction in their expression. For each RNA binding protein considered in this analysis, we calculated the proportion of EP events whose inclusion was consistently decreased across two biological replicates ($\Delta PSI < -0.1$ after shRNA knockdown relative to the controls) and plotted the distribution of this proportions in critical and remaining splicing factors in HepG2 cell line.

## CNV analysis

For each cancer type we obtained the level 4 CNV data from TCGA, which contained sample-specific information about the CNV profile of each gene (1 being CNV amplification, 0 being no CNVs, −1 being CNV deletion). To assess the CNVs of CSFs in each cancer type, we divided all samples into three quartiles based on the gene expression of each CSF. For each group of samples obtained in this way, we calculated the average CNV value for each CSF and compared these values for all CSFs between the quartiles using a Wilcoxon test.

## Single cell validation

For single cell validation of prioritized transcription and splicing factors, we obtained GBM single-cell SMART-seq datasets from 20 adult GBM tumors[114] from the Broad Institute Single Cell Portal (https://singlecell.broadinstitute.org/single_cell; Accession: SCP393). We also obtained normal brain single-cell SMART-seq and RNA-seq data and the annotations of cells from multiple cortical areas of the human brain from the Allen Brain atlas (2019 SMART-seq release, https://portal.brain-map.org/atlases-and-data/rnaseq)[115]. Oligodendrocytes, astrocytes, and oligodendrocyte progenitor cells were used as a normal reference to compute log-fold changes between malignant and normal cells. For liver cancer, LIHC single-cell RNA-seq data is 10X data sourced from a previous study[116] and the read count matrices and annotations were downloaded from the GEO database (GSE125449). For healthy liver, read count matrices were obtained from the HumanLiver package[117] (https://github.com/BaderLab/HumanLiver). Hepatocyte clusters (Hep 1–6) and cholangiocytes were used as a normal reference to compute log-fold changes between malignant and normal cells.

The activity of CSFs at the single-cell level was scored as a gene set using AUCell[118], and the resulting activity scores were z-scored across all cells separately in each tissue. We used the batch ID of the samples as a covariate in this analysis to account for sequencing differences due to differing batches[119]. In each case, the cell type annotations and their uniform manifold approximation and projection (UMAP) coordinates were also downloaded from the respective source indicate above.

## Reporting summary

Further information on research design is available in the Nature Portfolio Reporting Summary linked to this article.

## Data availability

The public RNA-seq datasets for human cancers were generated by TCGA consortium (https://www.cancer.gov/tcga) and are publicly available from the 'toilhub' of UCSC-Xena browser[100] (UCSC-Xena-TCGA). The public RNA-seq datasets for healthy human tissues were generated by GTEx consortium and publicly available from the 'toil-hub' of the UCSC-Xena browser[100] (UCSC-Xena-GTEx). The public mutation calls and copy number amplifications from whole exome sequencing data of human cancers are publicly available from TCGA genomics data commons portal (https://portal.gdc.cancer.gov/)[120]. The public RNA-seq datasets spanning multiple stages during human organogenesis are publicly available and downloaded from array express (E-MTAB-6814)[29]. The public clinical and survival data of cancer patients is publicly available and downloaded from Pan-Cancer Atlas initiative (TCGA-clinical)[121]. The public mapping of PFAM domains to hg38 assembly was performed by a previous study and the mapping coordinates are publicly available to download from the prot2hg database (http://www.prot2hg.com)[108]. The public data for frequently mutated splicing factors with a significant evidence for their cancer driver gene activity is publicly available and downloaded from Table S1 of Seiler et al.[54]. The public RNA-seq datasets for shRNA knockdown of splicing factors and corresponding controls for HepG2 cell line were downloaded from ENCODE database (ENCODE-shRNA-HepG2)[48]. The public data for doubling time, RNA-seq, and genome-wide dependency score for cancer cell lines are publicly available and downloaded from the DepMap portal release 22Q2 (DepMaP)[122]. The public single-cell RNA-seq data for glioblastoma patients is publicly available and downloaded from the Single Cell Portal of the Broad Institute under the accession code SCP393 (sc-GBM)[114]. The public single-cell RNA-seq data for healthy brain samples is publicly available and downloaded from Allen Brain Atlas (sc-Brain)[115]. The public single-cell RNA-seq data for liver cancer is publicly available and downloaded from GEO database under the accession code GSE125449 (sc-LIHC)[116]. Single-cell RNA-seq data for healthy liver is publicly available and imported with HumanLiver package in R (sc-Liver)[117]. The public data for differentially expressed genes following the deletion of TFs across multiple cell lines is publicly available and downloaded from KnockTF database (KnockTF)[56]. The public ChIP-seq datasets for the genome-wide binding of TFs across multiple model systems is publicly available and downloaded from the GitHub repository of the TFEA.ChIP library in R[55] (TFEA.ChIP). The public data for human phenotype ontology terms is publicly available and downloaded from The Jackson laboratory (HPO)[123]. Gene and transcript coordinates for hg38 assembly were downloaded from Gencode (Gencode V23)[124]. The remaining data generated in this study are provided with this paper as supplementary files and source data file. Source data are provided with this paper.

## Code availability

All the codes used in collection, processing and analysis of datasets are available are deposited at GitHub (https://github.com/hannenhalli-lab/AltSplDevCancer/) and the corresponding DOI is as follows: https://doi.org/10.5281/zenodo.7325464[125].

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

## Acknowledgements
This work is supported by the Intramural Research Program of the National Cancer Institute, Center for Cancer Research, NIH, and utilized the computational resources of the NIH HPC Biowulf cluster. We would like to thank Stephan Muljo and Thomas Gonatopoulos-Pournatzis for feedback on the manuscript.

## Author contributions
A.S.: Data curation, study design, software, formal analysis, investigation, visualization, methodology, writing–original draft, writing–review and editing. A.R., V.G., P.A.: Data curation, formal analysis, software, writing–review and editing. C.P.D.: Data curation, formal analysis, writing–review and editing. S.H.: Conceptualization, study design, supervision, funding acquisition, investigation, visualization, methodology, writing–original draft, project administration, writing–review and editing.

## Funding

## Competing interests
The authors declare no competing interests.
