## [Peer Review File · Nature Communications]

Broad misappropriation of developmental splicing profile by cancer in multiple organs.Reviewers' Comments:

Reviewer #1:

Remarks to the Author:

The paper describes analysis of alternative splicing changes during human development and cancer based on public RNA-seq data. The main claim of the paper is that developmental alternative splicing changes are reverted under cancer progression. Authors suggest possible regulatory network of splicing and transcriptional factors responsible in observed changes. While in general idea looks interesting and promising, I'm not convinced by provided results and not always satisfied by methods used.

Major issues

- 1) Authors use pseudoalignments and transcript level TPMs to calculate exon inclusion ratios (PSI). This method suffers from problems with assignment of reads mapped to exons shared across multiple transcripts to the particular transcript; it depends on particular transcript annotation and cannot take into account aberrant splicing that are known to be abundant in cancer. Low precision of pseudoalignment causes additional problems. It is widely accepted that methods used on reads mapped to exon-exon junction are preferred for quantitative analysis of alternative splicing, some of these methods might be found here: <https://github.com/OlgaVT/Alternative-Splicing-Tools>
- 2) Authors used very indirect way to define main object of their analysis: exons that are used in embryo but not in adults (EP) and exons that are not used in embryo but included in adults (EN). To do so authors use correlation between PSIs on individual exons and mean activity of gene pathways. Authors motivate this analysis by noisiness of individual events, but their analysis is still based on PSI values of individual exons that are much more noisy than gene expression where agglomeration of individual events was performed (to pathways), so the motivation makes no sense. Our paper (Mazin et al, 2021, cited by authors) clearly show that such analysis can be performed on level of individual exons without use of any additional information and allows to identify thousands of events. Inclusion ratio changes between pre- and post- natal stages for identified EP and EN are very moderate (median < 0.15, fig 1E). According to fig 1E EPs do never have inclusion ratio above 0.5 thus, transcripts including such exons are always minor even in pre-natal stage. So EP cannot be considered as "embryo-specific" (and vice versa for EN). Taking into account that there are many known switch like events (that change inclusion from almost 1 to almost 0 or vice versa, see Mazin et al, 2021 or https://vastdb.crg.eu/wiki/Main_Page for example) during development, proposed EPs and ENs look not so specific.
- 3) In cancer-related analysis as well as in single cell analysis authors use data from multiple datasets (GTEx vs TCGA) that makes their results confounded by batch effect. Authors should consider to look for data that do not struggle from such problems or clearly state and justify usage of this data. It is important to prove that major results are not affected by batch effect.
- 4) From results shown it is unclear what proportion of developmental changes are reversed under cancer and what proportion of cancer-related changes corresponds to reversal of developmental changes. Odds ratio of figure 2A are very moderate, especially for liver and kidney. So, I'm not sure that reversal of developmental changes are so "broad". I think it would be very useful to show scatterplots of that would show exon PSI changes during development (post-natal minus pre-natal) vs cancer induced changes.
- 5) Probably partially out of the scope of this work, but I believe that paper would benefit from contrasting of changes observed on level of alternative splicing with changes observed on level of gene expression. For instance, scatterplot for log fold changes similar to ones suggested above might be used.
- 6) Correlation is not sufficient to implicate splicing factors in regulation of alternatively spliced exons. It should be complemented by motif analysis (using motifs from CIS-BP-RNA for instance) and/or knockout experiments and/or by clip-seq (from ENCODE for example).
- 7) It seems like authors assume that splicing factors always enhance exon inclusion and thus have positive correlation between exon PSI and SF expression. But this assumption is not true, SF might suppress exon inclusion and effect of some of factors depends on position of its binding to mRNA

(upstream or downstream of exon).

Minor issues

L88 I believe that "EN" is "embryonic negative" but it is first used in line 88 and not formally defined

L87 "transmembrane-region domain (ND)" – should it be TRD instead of ND?

L117 - "Supplementary Fig. S1A" I would suggest to show different stages in different colors, now it is hard to notice stage clustering

Fig 1B colors denote cosine similarity, but it is not described between what and what the similarity is calculated

L129 "exon with its cognate gene" – I'm not sure that "cognate gene" is a widely used term. I would suggest to use "exon host gene".

L144 "EP and EN exons exhibit mutually exclusive inclusion patterns in the pre- and postnatal stages (Fig. 1E, and Supplementary Fig. S1B)" – "mutually exclusive exons" is specific and relatively rare type of AS where only one of set of consequently placed cassette exons is included in mature mRNA. Authors clearly mean something else, so I would suggest to not use "mutually exclusive" to evade confusion. Additionally, plots provided clearly show that for both exon types (EP and EN), in most cases both isoforms are expressed in both stages (pre- and post-natal). In case of mutually exclusive expression these distributions should collapse to 0 and 1.

fig 2B, does colour shows correlation coefficients? Authors should specify type of correlation coefficient used (Pearson, Spearman, etc) and what values were used for "cancerSEA hallmark gene set" (mean log(CPM)?). What dataset was used to calculate this correlation coefficients? Is it correlation along development?

What is shown on figure 2D bottom? If it is "proportion of EP events associated with..." should it be just a single value (for each tissue)? In this case it should be shown as barplot. How EP events associated with better (or worst) survival were defined? Shouldn't their proportions sum up to 1?

L214 "We found that the shared EP events were a significantly stronger predictor of pan-cancer patient survival as compared to tissue-specific EP events (Fig. 2D)" – all comparisons on figure 2D have p-value above 0.05 and, thus, not significant

there are no panel label for 2E

L317 "As expected, the top positive predictors of EP events identified based on their coefficients in the PLSR model, which we refer to as critical splicing factors (CSFs, Supplementary Table S5), had higher expression during the prenatal stage of development, and underwent significant upregulation in their corresponding cancer (Fig. 4B)" – why is it expected? Different SFs might either activate or suppress exon inclusion. In latter case CSFs should exhibit high expression in healthy samples and low in embryonic or cancer.

I would suggest to show odds ratios (fig 2A for instance) in log2 scale

if EP are not included in health GTEx then their dysregulation (lost of differentiation) can result in increase only. How dPSI(cancer – healthy) depends on PSI(health)? If I understand S2B correctly it only shows that exons with moderately low inclusion ($PSI < 0.3$) do not have strong bias toward increased inclusion in cancer. But it do not exclude the possibility, that part of EP are not included in healthy adults ($PSI \sim 0$) and then any dysregulation for them should results in increase of their

inclusion.

What do figure S2B show? What is the universe (all exons considered in analysis)? What are two factors compared by fisher test? What exactly was sampled randomly? What is the purpose of random sampling and may it be replaced by confidence intervals calculated on whole sample? What conclusion was drawn from this analysis (I186)?

I327 "In addition, a gain in CSF expression is significantly associated with worse patient survival" – for brain median for CSF is close to 1, that mean no effect, right? in this case rather higher expression of non-CSF seems to be associated with better prognosis. So the statement is not correct, there are no tests for difference from 1. Fig 4D just shows that increase in CSF expression is worse than increase in non-CSF expression.

Fig 4A shows that model predicts well exon inclusion in cancer, but significant questions are: a) whether the model prediction is better than a trivial one (for instance mean (across all developmental stages) PSI) b) can model predict cancer-related changes, that is whether difference between prediction for cancer and healthy samples correlates with observed changes. If answer to a) is "no" then model is trivial if answer to b) is "no" than model makes no sense for studied processes (development and cancer

L329 "we tested if the EP inclusion level is lower in tumor samples bearing nonsense (inactivating) mutations in CSFs. We first identified all SFs whose mutant samples have lower EP inclusion than the wildtype samples" – this looks like authors selected only cases that confirms their hypothesis... need to be clarified

L332 "significantly higher (and positive) regression coefficients as compared to the other SFs (Fig. 4E)" – regression coefficients between what and what?

L359 "there is a remarkable association between the doubling time and the EP/EN inclusion levels across cell lines derived from the organ-specific cancer type (fig 5A)" – all p-value of fig 5A are above 0.05, then there are no significant association. BTW, kidney EP have positive slope but negative Pearson correlation coefficient, that is weird.

What is the difference between 4A and 5B?

L376 "the greater was the dependency of the cell line on that SF" – how the "dependence of the cell line" on SF was measured?

What data was used for fig 5c? Are they described in "Single cell validation." section of methods? If so the data were obtained from multiple datasets and should be confounded by batch effects than makes comparison unreliable. Additionally, authors use very large symbols on 5c, so only few cells from the top layer are visible, rest is hidden behind them. Visualization should be changed in such way to make most of cells visible.

Methods:

I593 "In 5-dimensional PC space (first 5 PCs explain ~65 % of variance), each timepoint occupies a unique coordinate based on the PC scores" what do authors mean under "timepoints" here?

L604 "For each exon, we selected the significantly positively or negatively correlated KEGG pathways based on the Benjamini-Hochberg FDR threshold of 0.05." - how exactly BH correction was applied? Was it done for each exon or each pathway independently? Or was it performed for all exon*pathways pairs simultaneously? The former approaches could cause underestimation of FDR.

l625 - "Therefore, we selected the events which were at least 2 standard deviations away from the mean of their distribution in the corresponding GTEx normal samples in a consistent direction (i.e. increased or decreased) in at least 15% of the cancer patients" – same exon can be both increased or decreased by this definition. It is better to use dPSI threshold, since changes below 5 (10%) are usually considered as noise. Some estimate of statistical significance is necessary.

Reviewer #2:

Remarks to the Author:

In their manuscript titled "Broad misappropriation of developmental splicing profile by cancer in multiple organs," Singh et al. present a comprehensive, systems-level study that identifies developmentally-regulated splicing events in liver, kidney and brain and explores the misregulation of these exons in the corresponding, tissue-specific cancer. The study integrates information from a wide-variety of databases and public data collections to conclude that tumors show a progressive reversion to embryonic splicing patterns, identify potential splicing factors regulating these events and transcription factors controlling their expression, and suggest possible existing drugs that may target these factors and thus have putative therapeutic potential.

As someone who studies developmental splicing, I was intrigued by the large number and wide variety of predictions that can be explored in future studies in terms of splicing regulators, transcriptional regulators, spliced exons and drugs that might influence these splicing outcomes and associated cellular processes. I think this is an interesting paper that puts forth multiple, important hypothesis that should be experimentally tested, and will be of interest to the oncogenesis community, but also developmental biologists and the splicing community. That said, the central tenet of the manuscript is not new, and the authors themselves cite multiple studies suggesting important roles for splicing and reversion to the embryonic state in various cancers throughout their manuscript. The manuscript also relies completely on systematics and correlation, which although it indeed provides strong motivation to pursue novel hypotheses, doesn't itself provide new data verifying those hypotheses. Below I provide several suggestions to improve the study and manuscript.

1. Please provide a full list of all abbreviations and define them when they are first introduced in the text. Please also check that abbreviations in figures are defined in the legend (ie LIHC in Fig. S2 or vely in Fig. 1D).

2. The authors should somewhere address the relationship between "cause" and "effect" in terms of the embryonic splicing profile and misregulation of splicing factors in the cancers they examine. Based on Fig. S2 C & D, there is a progressive misregulation of splicing in early and late stage cancers. Does this reflect that splicing is therefore not a cause of the oncogenesis, but rather an effect of the changes that allow oncogenesis in the first place? For example, how many early stage cancers already contain inactivating mutations in key CSFs? Is there also a corresponding increase in CSF mutations in late stage cancers? In muscle disease, for example, mutations in splicing factors are causal for and precede the disease, but it is still poorly understood how the wide variety of splicing changes and the reversion to embryonic splicing patterns on whole actually contributes to the disease. Can the authors provide such insight in the cancer context? Another aspect of this is how many CSF genes are actually activated in malignant cells? Is activation of 1 CSF sufficient, or do multiple CSFs have to be targeted to promote malignancy?

3. Figure 3 – The underlying finding that EP/EN exons contribute to alternative isoforms of proteins containing TRD, ND and WD40 domains is very interesting. Can the authors provide examples or do a transcript-level analysis to better characterize or to confirm that the exon switches do indeed cause protein-level coding differences? Have any of the identified exons already been reported to be regulated in cancer or development, that the authors can use as a "marker" to support that the ES/EN exons they identify actually contribute to isoform switches that are physiologically relevant? The

correlation matrices in Fig 3C & D to me are somewhat redundant, because at least in my experience, we focus on either BP or MF terms because of the large overlap in genes in related terms between the two annotations. The enrichment is performed on the same list of terms, so I wouldn't expect anything different. I do agree that practically, such a plot (ie Fig3D) would help you focus on 3 processes instead of dealing with 18+ GO terms.

4. Is it possible to provide experimental data for the OST complex, or GTPase activity or ARL1 to confirm that the correlations and predictions from the systematic analysis actually have coalesced on valid targets? The analysis looks compelling, but from experience, when you start working with individual genes from such an analysis you do not always get the expected result. The impact of the manuscript would be stronger if the authors could show new experimental data. More generally, are there possibly reported proteins or events that can confirm some of the systematic findings and can be used as a "case study" to support the conclusions, in lieu of new wet-lab experiments?

5. Figure 5 – the authors state the association between EP/EN inclusion is "remarkable," but especially in kidney the R is not remarkable, and only in brain is the regression/relationship statistically significant. The results in Fig. 5C are also not so clear-cut as the authors state in the text. I do not doubt that some ES events are linked to changes in proliferation, but the reality seems to be more nuanced and heterogeneous. Can the authors define a narrower set of exons or even CSFs that are tightly linked with proliferation? This is also somewhat relevant to a point they raise in the discussion in lines 540-545. It is unclear that targeting a single EP exon would be effective, but is there a way to predict the smallest set of exons that would have to be targeted to produce an effect? How heterogeneous are the different cancer cell lines in this respect?

6. Can the authors comment on EP/EN use in the CSFs they have identified? Splicing factors are reported to themselves be heavily spliced, in particular in both the developmental and oncogenic context. Are there any relationships in the data supporting that specific isoforms of CSFs are associated with oncogenesis?

7. In the methods, the authors mention that the cancer-specific splicing events are limited to exon skipping events. Is this also true for the developmental events? Can the authors comment if their findings only are valid for exon skipping events, or for other types of AS events?

Minor points:

-Line 87: typo: transmembrane-region domain (ND) should be (TRD)

-Line 100: Is this MYC and FOXM1, or does the MYC, FOXM1 mean another TF is missing from the list?

-Lines 133-136: This sentence isn't logical. Enrichment of EP pathways in terms related to oncogenesis doesn't validate a role for EP pathways in embryogenesis and organ development. At best, it just reflects that both cancers and developing tissues are actively proliferating, and the approach the authors used accurately identifies these KEGG pathways.

-Lines 245-248: This is an interesting point, but is not clearly stated. If I understand correctly, the analysis identifies EP and EN exons as contributing to coding TRD, ND and WD40 domains across all of the tissues, BUT the genes that are identified are tissue specific. I do not follow how this is then related to "multiple genes coordinate the splicing of these domains across tissues", because identification of the CSF's comes at a later point in the manuscript.

-Legend Fig. S1 – Plot C is missing in the legend, and D and E are mislabeled in the legend as C and D.

-Legend Fig. S2 – Panel A is a dotplot, not a biplot. Is this a typo in the legend? Or is Panel A incorrect?

Validity: To my knowledge, the approach the authors take is valid. They apply methods generally accepted in the field, and elegantly combine data from multiple sources to evaluate their questions. There are several cut-offs used in the analysis that clearly do not make this a "comprehensive" list of developmental, oncogenic exons and regulatory factors; however, the thresholds are reasonable and

justified.

Reviewer expertise: I am an experimental geneticist who studies alternative splicing and RNA-regulation in the context of tissue development and differentiation.

Reviewer #3:

Remarks to the Author:

The manuscript by Dr. Singh and colleagues is a very well written paper on the timely subject of cancer splicing providing valuable new data with novel and thought-provoking findings. The authors did a comprehensive bioinformatic survey of publicly available transcriptomic data to identify organ-specific alternative splicing profiles and compare them to cancer transcriptomic data of the same organs. They find that organ-specific developmental splicing is enriched in the respective cancer at a large-scale. They also identify PFAM domains affected by the alternative exons and splicing factors predicted to regulate these splicing events. This validation on a more global level is much needed in the field as there has only been anecdotal evidence of individual developmental splices being reactivated in cancer. In particular, their findings on splicing of EP transmembrane (TRD) and nitrosylation (ND) domains and their implications for secretory/Rab pathways and cell communication and signaling during neuronal development and cancer are quite novel and very insightful. Overall the paper is very strong in bioinformatics analysis and the findings are well presented and explained. Certain of the claims could be further strengthened by experimental validation. Nevertheless, these findings provide interesting hypothesis for experimental labs to further validate.

I have just a few points/comments that need clarification:

1. The authors mention they used SUPPA2 software for their splicing analysis. Did the authors try other commonly used splicing software like rMATS? If yes, where the results comparable? Some justification on their choice of software would be desirable.
2. In the PFAM domain analysis in brain and kidney it looks like the TRD, ND and most of the other PFAM domains are enriched only in EN exons. Does this mean that the TRD- and ND-encoding exons are skipped in these cancers? If so, then this is a major point that deserves more detailed discussion.
3. Did the authors investigate the potential effects of the EN and EP exons encoding for TRD and/or glycosylation domains on subcellular localization of the encoded proteins? For example, if a transmembrane domain (TRD) is skipped in cancer then that would be predicted to affect the subcellular localization and the solubility of the encoded proteins. Are there any examples of TRD EPs/ENs resulting in a transmembrane protein becoming secreted? Such EPs/ENs would be expected to play a major role in intercellular signaling for example.
4. Lines 318-324 on Critical Splicing factors: how did the authors assess that they were critical? CRISPR essentiality database? This should be stated in the text.
5. Lines 330-339, causal CSFs: the effect of some of these mutations might be more subtle than lead to total CSF inactivation. Another complementary way of validating this would be to look in DepMap or other essentiality databases for CRISPR or RNAi data for these CSFs and correlate with EP levels. Or even better, do the actual experiment in the lab for the top 10 factors: that is silence them by RNAi or CRISPR and assess if the EP inclusion levels are affected.
6. CSFs: There are no names of the CSFs mentioned. The authors should provide the names of at least their top ranking CSFs to enable comparisons and integration with previous literature. Do these include the already known CSFs that are frequently mutated in cancer? If this is the case the manuscript could be substantially strengthened.
7. TFs regulating CSFs: a lab experiment would be appropriate to validate a subset of these predictions. Alternatively, the authors could mine the DepMap or other gene essentiality databases and/or publicly available data for further validation.
8. Lines 424-425: The drug repositioning for targeting TFs with FDA drugs as an approach with therapeutic potential is rather speculative without any experimental data. Unless validation data are presented such claims are rather weak and should be removed.

Minor points:

Line 247: 'the' is missing before 'observed enrichment...'

Line 288: 'for' instead 'of' before 'alternatively spliced ND...'

There is a few other sentences where some articles like 'the' are missing before the nouns (minor grammatical errors).

The Tsai et al. 2015 reference is missing.

REVIEWER COMMENTS

Reviewer #1, expert in multi-omics, bioinformatics and splicing (Remarks to the Author):

The paper describes analysis of alternative splicing changes during human development and cancer based on public RNA-seq data. The main claim of the paper is that developmental alternative splicing changes are reverted under cancer progression. Authors suggest possible regulatory network of splicing and transcriptional factors responsible in observed changes. While in general idea looks interesting and promising, I'm not convinced by provided results and not always satisfied by methods used.

We appreciate and thank the reviewer for providing us with feedback on this work. In the response document and the updated version of manuscript, we explain multiple additional analyses (including the ones suggested by the reviewer) and find that the observed reversal of cancer splicing towards their embryonic counterparts of the tissue of is robust even when doing additional controlled analyses. Further we have added the experimental evidence from HepG2 cell line to support the causal role of positive regulators of EP splicing events detected from PLSR model. Below we provide a detailed point-by-point response to the reviewers' comments.

Major issues

1) Authors use pseudoalignments and transcript level TPMs to calculate exon inclusion ratios (PSI). This method suffers from problems with assignment of reads mapped to exons shared across multiple transcripts to the particular transcript; it depends on particular transcript annotation and cannot take into account aberrant splicing that are known to be abundant in cancer. Low precision of pseudoalignment causes additional problems. It is widely accepted that methods used on reads mapped to exon-exon junction are preferred for quantitative analysis of alternative splicing, some of these methods might be found here: <https://github.com/OlgaVT/Alternative-Splicing-Tools>

This is a great comment for technical validation of our analyses approach. We would like to kindly note that there are always multiple software options to choose from for RNA-seq analysis including splicing. Every method has its own strengths and weaknesses, depending upon their strategy for alignment, dealing with multi-mappers, and calculating effective transcript lengths (C. Zhang et al., 2017). In this work, our goal was not to detect novel splicing junctions in cancer (something which has been done previously, Kahles et al., 2018), but rather investigate functional parallels between embryogenesis and cancer based on transcriptome-wide alternative splicing. It is worth mentioning that human transcriptome is most well annotated and the process of annotation, in addition to automated tools, involved a slow but accurate manual efforts from HAVANA team, which took almost 13 years to complete (Zerbino et al., 2020). Therefore, we chose to utilize pre-existing comprehensive transcriptome annotations along with accurately quantified RNA-seq datasets from recompute2 project (Vivian et al., 2017) to do this analysis.

The reviewers concern has two parts and below we respond to each one by one.

a. First concern is regarding potentially low precision of the pseudoalignments and problem of assigning multi-mapping reads to their correct transcript of origin. We would like to note that Kallisto has been benchmarked in several publications, showing that pseudoalignment is as good as or even better than classical alignment and the resulting quantification is highly correlated with the ground truth. For eg.

<https://bmcgenomics.biomedcentral.com/articles/10.1186/s12864-017-4002-1>
<https://www.ncbi.nlm.nih.gov/pmc/articles/PMC7084517/>
<https://www.nature.com/articles/s41598-020-76881-x> (see 'quantification by pseudoalignment').
<https://www.nature.com/articles/s41598-017-01617-3>
<https://www.biorxiv.org/content/10.1101/444620v1.full.pdf>

Also worth mentioning the following independent validations:

<https://cgatoxford.wordpress.com/2016/08/17/why-you-should-stop-using-featurecounts-htseq-or-cufflinks2-and-start-using-kallisto-salmon-or-sailfish>
<http://genomespot.blogspot.com/2015/08/how-accurate-is-kallisto.html>

b. The second concern is that splicing quantification using reads mapped to exon-exon junctions is more accurate as compared to using the reads mapped to junctions as well as the exon body. This is a valid point, but as the reviewer would appreciate, the choice between only junction reads vs. all reads entails a tradeoff between stringency and sensitivity. For the genes with low expression, there could be too few reads which span a splicing junction, thus reducing the sensitivity of the detection. However, to directly address the reviewer's concern, as a test case, we recomputed the splicing values in all brain samples by aligning the .fastq files with STAR two pass alignment followed by splicing quantification using rMATs (which does not rely on transcript level expression) and calculated the correlation between rMATs splicing values using exclusively the junction reads vs. SUPPA2 splicing values (Y-axis in the following plot). We observed that median correlation across all the brain samples was 0.84 with a narrow distribution, suggesting high concordance between the two approaches.

Additionally, SUPPA2 enabled us to utilize elegantly quantified RNA-seq datasets from UCSC-toil recompute compendium (Vivian et al., 2017) for TCGA and GTEx, which were normalized for within cohort and potential computational batch effects. Therefore, SUPPA2 is naturally faster as it can use precomputed TPM values (such as Vivian et al., 2017)

Therefore, the choice of our pipeline, i.e. Kallisto (Bray et al., 2016) + SUPPA2 (Trincado et al., 2018) is motivated by careful consideration of its speed, accuracy, need for lesser memory, and very common use within the community. Thank you for bringing up this important point. We have now added this additional commentary at lines 739 and 746.

2) Authors used very indirect way to define main object of their analysis: exons that are used in embryo but not in adults (EP) and exons that are not used in embryo but included in adults (EN). To do so authors use correlation between PSIs on individual exons and mean activity of gene pathways. Authors motivate this analysis by noisiness of individual events, but their analysis is still based on PSI values of individual exons that are noisier than gene expression where agglomeration of individual events was performed (to pathways), so the motivation makes no sense. Our paper (Mazin et al., 2021, cited by authors) clearly shows that such analysis can be performed on level of individual exons without use of any additional information and allows to identify thousands of events. Inclusion ratio changes between pre- and post-natal stages for identified EP and EN are very moderate (median < 0.15, fig 1E). According to fig 1E EPs do never have inclusion ratio above 0.5 thus, transcripts including such exons are always minor even in pre-natal stage. So EP cannot be considered as “embryo-specific” (and vice versa for EN). Taking into account that there are many known switch-like events (that change inclusion from almost 1 to almost 0 or vice versa, see Mazin et al., 2021 or https://vastdb.crg.eu/wiki/Main_Page for example) during development, proposed EPs and ENs look not so specific.

We appreciate this comment and something we considered before adopting the methodology used in the manuscript. The first objection of the reviewer is regarding our choice of pathway-guided analysis of alternative splicing. With regards to ‘noise’ what we meant is that even though PSI values are (presumably) noisier, we only retain the events which are highly correlated with agglomeration of more than 100 embryonic pathways, and therefore, more robust than relying on differential inclusion alone. We would like to highlight that this is motivated by the previous approach – PEGASAS, by Y Xing’s group (Phillips et al., 2020). There were multiple considerations that went into our specific choice, as detailed below.

The rationale underlying PEGASAS is that the critical molecular entities underlying the phenotypic variations across disease, development, and ageing, co-vary with the biological processes and pathways linked with the specific phenotypic states. Therefore, the task of identifying the molecular changes associated with a phenotype is reduced to first identifying the processes and pathways linked to the phenotype and then in a subsequent step identifying molecular changes co-varying with those processes and pathways. This concept has been successfully applied to investigate alternative splicing in prostate, breast, and lung cancer datasets by Yi Xing’s group (Phillips et al., 2020).

While the use of differential analysis, which relies on assessing the significance of difference between two biological conditions/states (for instance embryonic vs. adult) is appealing in its directness, the functional interpretation of the resulting gene list (exons in our case) is challenging and requires additional steps of gene set analysis, which can be unstable (Gaudet & Dessimoz, 2017; Jacobson et al., 2018) and heavily depends on the choice of software (Xie et al., 2021).

Contrastingly, in our approach, we first identify hundreds of KEGG pathways which show preferentially high activity during the pre-natal stage of the development (note that in this step, we do not calculate functional enrichment, but rather score the preferential activity of each pathway, Fig 1B) and select the splicing events which exhibit a significant co-variation with those embryonic pathways. Such splicing events, by virtue of their preferential correlation with embryonic pathways, have a more straightforward interpretation which does not suffer from aforementioned pitfalls for their functional interpretation. To sum up, this approach enables us to identify the exons which are:

- a. Preferentially embryonic in nature to begin with, and importantly,

- b. are correlated amongst each other, i.e., they change in coordinated fashion across developmental timepoints, which is consistent with previous publications showing the coordinated change of several hundred to thousands of exons in response to diverse biological signals and contexts (Bland et al., 2010; Moore et al., 2010; Warzecha et al., 2010). Also, the multivariate structure of exon inclusions has been successfully used recently to discover the sQTLs across GTEx tissues (Garrido-Martín et al., 2021), further supporting the need for correlation based approach.

Here, we show that the PEGASAS based approach is superior in delivering these goals as compared to the conventional approach of differential splicing. For differential splicing, we used Wilcoxon test followed by FDR correction as has been performed previously (Y. Zhang et al., 2019) and called an event to be embryonic positive if Δ PSI (prenatal – postnatal PSI) values were > 0.2 with an FDR ≤ 0.05 in developing brain. We intersected this new set of embryonic events with our pathway-based EP events and obtained unique EP events in our approach (referred here to as pathway-only), pathway-based events that additionally qualified the differential splicing criteria above (referred to as pathway+wilcox), and the unique Wilcoxon test based events (referred here to as wilcox-only).

First, we assessed the coordination in splicing within each of the three sets of exons by calculating the within-group pair-wise Pearson correlation coefficients among their inclusion level across the developmental data. As shown in the boxplots below, we observed that pathway-based events (even the unique ones, i.e. pathway-only) are significantly more correlated with each other than the wilcox-only group.

The coordinated nature of splicing events is an important aspect of our study which has been relatively neglected so far in the field of cancer biology and which helped us elucidate the important associations of the coordinated embryonic splicing events with cellular functions such as N-linked protein glycosylation and retrograde transport (Fig 3, 7 and S6).

Moreover, while the events identified by the pathway approach (as shown in Fig. 1C) and even the events unique to pathway-based approach (as shown below) were enriched for several GO terms related to the embryonic development of brain, notably, the events unique to differential events were not enriched for any functional category as shown below. This shows that standard differential splicing analysis, which does not directly take into account the multivariate structure of exon inclusions (as explained in Garrido-Martín et al., 2021), fail to capture the biologically relevant splicing events.

Finally, we show that the recapitulation of embryonic splicing in cancer holds up even in the pathway-only events but exhibits a much weaker trend for wilcox-only events that was statistically significant in only one of the three tissues.

Taken together, these analyses suggest that pathway-based approach is more effective in detecting the coordinated set of context-specific events that are better recapitulated in cancer as well as provides a more direct functional interpretation of splicing events correlated. Further, the coordinated nature of pathway-based events enabled us to detect the coordinated mis-regulation in functionally related exons

in multiple genes. For example, as we show in Fig 3E,F (and more broadly in Figure S6), all the exons encoding the transmembrane domain in the four subunits of oligosaccharide transferase complex undergo coordinated change in their splicing during embryonic development and in cancer, thus, regulating the N-linked protein glycosylation in context of the overall physiological context of the cell. The successful identification of such examples (Figure S6) provides a strong support in favor of the pathway-based approach.

We acknowledge that there are alternative statistical approaches (for instance Mazin et al. use a cubic spline fit to the inclusion levels and use Max-Min threshold of 0.2 to define developmentally dynamic events), PEGASAS is another established approach that suits our purpose as it provides direct functional interpretation, as well as strictly picks only those exons which consistently and significantly co-varied with hundreds of embryonic pathways, thereby filtering out the potentially noisy inclusion events.

We have now added this additional analysis in Supplementary note 1 mentioned at line 797.

The reviewer's second objection in this comment is the magnitude of change in the splicing of EP events. We would like to clarify that Fig 1E shows the median of all the EP events during pre-natal and post-natal timepoints. There are several events for which the delta PSI is greater than 0.5. Also, as shown previously, in some instances, even 10 percent change (i.e., $\Delta\text{PSI} \sim .10$) in the composition of transcriptional isoforms can have substantive effects on the physiology of the cells (Ma et al., 2022) Further, in Slaff et al., 2021, ΔPSI of 0.1 was used to identify the exons which are differentially spliced upon the knockdown of U2AF2. Some of the commonly used splicing analysis tools, for instance, rMATS, used 0.05 in their benchmarking studies to identify differentially spliced events between two prostate-cancer cell lines (Muller et al., 2021). In the same work, when the authors validated 30 exons with delta PSI ranging between 0.1 and 0.9 using fluorescent quantitative RT-PCR, they found a high concordance (correlation = 0.96), suggesting that delta PSI values as small as 0.1 can be experimentally recapitulated. Therefore, to be experimentally reproducible and physiologically relevant, exon inclusion does not necessarily have to switch between 0 and 1. Consistent with these examples, a threshold of delta PSI 0.1 or 0.2 is commonly used in many of the publications related to cancer transcriptomics, as exemplified by the following studies:

<https://bmcmmedgenomics.biomedcentral.com/articles/10.1186/s12920-020-00836-4>

<https://journals.plos.org/plosbiology/article?id=10.1371/journal.pbio.3001138> (threshold of 0.05)

<https://academic.oup.com/narcancer/article/3/2/zcab024/6299998> (threshold of 0.1)

<https://www.ncbi.nlm.nih.gov/pmc/articles/PMC4989457/> (threshold of 0.1)

<https://www.ncbi.nlm.nih.gov/pmc/articles/PMC5002109/> (threshold of 0.1)

<https://academic.oup.com/nar/article/50/D1/D1340/6374477> (database for clinically relevant splicing)

As a final point, for coordinated events affecting a single multi-subunit complex or process (such as our Oligosaccharide example, where the all four subunits of OST complex downregulate the inclusion of their TRD domain, albeit to the varying extents), it is very much likely that small change in an individual event is physiologically relevant effect in coordination with more stringent changes.

3) In cancer-related analysis as well as in single cell analysis authors use data from multiple datasets (GTEx vs TCGA) that makes their results confounded by batch effect. Authors should consider to look for data that do not struggle from such problems or clearly state and justify usage of this data. It is important to prove that major results are not affected by batch effect.

Again, this is very important comment to ensure the sanity of our work. There could be two potential sources of batch effects:

- a) Effects due to computational processing: Different computational pipelines sometimes provide disparate quantifications for certain genes. But for our current comparison of TCGA and GTEx, we used the transcriptomic datasets from the toil-hub of the UCSC Xena browser which processed the two datasets using uniform computational pipelines, thus minimizing potential batch effects due to differences in computational pipelines.
- b) Effects due to experimental handling and sequencers: Within each cohort (i.e. within a cancer type and a tissue type), the confounding effects originating from different batches of sequencing were removed for the datasets processed on toil-hub ((Vivian et al., 2017). However, the batch effects between TCGA and GTEx is a trickier issue as those are two entirely different cohorts as well as biologically different samples, and unfortunately, if there are systematic batch differences between the two, there is no way to learn and remove such effects. Only if some of the TCGA samples were sequenced along with GTEx cohorts (and vice versa), one could learn and remove the batch effects. Nevertheless, the transcriptomic differences originating from the biological differences between TCGA tumor and GTEx normal for a given tissue should be much more pronounced as compared to the batch effects. Indeed, this concern would apply to numerous published works comparing GTEx with TCGA.

We have now discussed this issue at line 723.

As far as batch effects in the single cell analysis are concerned, we used the batch ID as a covariate in the differential expression analysis as is done previously conventionally (Ntranos et al., 2019), now mentioned at line 961.

4) From results shown it is unclear what proportion of developmental changes are reversed under cancer and what proportion of cancer-related changes corresponds to reversal of developmental changes. Odds ratio of figure 2A are very moderate, especially for liver and kidney. So, I'm not sure that reversal of developmental changes are so "broad". I think it would be very useful to show scatterplots of that would show exon PSI changes during development (post-natal minus pre-natal) vs cancer induced changes.

It is indeed very important to have an idea about the proportion of developmental events which change in cancer. As suggested, we have plotted the proportion of such events. Remarkably as much as 50% of the EP/EN events in brain, 20% in kidney and 10% percent in liver switched back to their embryonic state in the cancer samples. Likewise, almost 30% of the mis-spliced events in brain cancer, 7% in kidney cancer, and 9% in liver cancer belonged to EP or EN category. These are modest but substantial fractions to be functionally relevant. We have explained these results at line 183 and Supplementary Fig. S2B,C.

Moreover, functionally related events switch back to the embryonic counterparts in a coordinated fashion (Fig 3 and S6), which is highly supportive of the coordinated mis-appropriation of developmental splicing in cancer.

Further, we made a scatter plot of Δ PSI during development (i.e. prenatal-postnatal) against of Δ PSI during in cancer (TCGA-GTEX) as suggested by the reviewers. As shown below, there is an overall significant correlation between developmental splicing and cancer-induced changes, especially in brain and liver. This suggests that cancer splicing broadly reverts to their embryonic counterparts of their tissue of origin. Even though we do not observe equally strong correlation in kidney, we do observe significant odds ratio (Fig 2A). While cancer cells clearly do not recapitulate all aspects of embryonic cells, they do mimic several key aspects of embryonic development, such as rapid proliferation, migration, immune suppression etc. Cancer is driven by several other factors which are non-embryonic in nature. But despite these differences, we observed a significant enrichment of embryonic splicing in cancer in all three tissues (Fig 2A) as well as a direct correlation for liver and brain, supporting the general premise that cancer reactivates embryonic patterns of alternative splicing.

We have now added these additional analyses at Line 196 and Supplementary Fig S2F.

5) Probably partially out of the scope of this work, but I believe that paper would benefit from contrasting of changes observed on level of alternative splicing with changes observed on level of gene expression. For instance, scatterplot for log fold changes similar to the ones suggested above might be used.

This is again a good comment, and we had indeed done this. As shown in Fig 1D and S1C, the EP and EN exons had both positive and negative correlation with their host genes, although the former had slightly biased toward positive and latter slightly toward negative correlation. Additionally, we made it very explicit that nearly 20-30% of the host genes had an EP as well as EN event (line 155), suggesting that splicing variation can be orthogonal and complementary to gene expression. This is consistent with previous reports, where alternative splicing can better distinguish between cancer types and subtypes as compared to the gene expression. (Fig 2C and S10,P in Kahles et al., 2018).

6) Correlation is not sufficient to implicate splicing factors in regulation of alternatively spliced exons. It should be complemented by motif analysis (using motifs from CIS-BP-RNA for instance) and/or knockout experiments and/or by clip-seq (from ENCODE for example).

We thank the reviewer for this comment. We do agree that correlation is not sufficient to imply causal role of splicing factors in the regulation of the embryonic splicing. Precisely for this reason, we had analyzed the somatic mutation data of splicing factors in the TCGA cohort (Fig 4E), where we show that splicing factors whose inactivation (through non-sense mutations) associated with decreased embryonic splicing have positive regression coefficients in the PLSR model. However, to further address these concerns, we have now added the following new analysis in the manuscript at Line 388, and methods section at line 920, and Fig 4F.

We downloaded the raw RNA-seq datasets for shRNA knockdown of ~231 RNA binding proteins in HepG2 (a liver cancer cell line), quantified transcripts expression, gene expression and splicing using our pipeline, and trained a PLSR model for the regulation of liver specific EP events in HepG2 using a set of 52 samples which were used as controls in the shRNA experiments. We identified critical splicing factors (i.e. the positive regulators of EP, see our response to comment no. 7 below) based on their regression coefficients in PLSR model as described in the manuscript. Very encouragingly, we observed that upon shRNA knockdown a greater fraction of critical splicing factors resulted in the decreased EP splicing as compared to non-critical splicing factors. Taken together, the analysis of EP events in the developmental transcriptomes, cancer transcriptomes, mutation analysis, and shRNA-seq provides very strong evidence for the role of critical positive regulators in the EP events.

As far as enrichment of eCLIP peaks is concerned, we did not find the enrichment of CSFs eCLIP peaks, at 5' or 3' splice site of the EP exons. However, this does not necessarily negate our findings as the direct binding of a specific RBP is not a requirement for it to regulate a splicing event. An elegant publication from ENCODE consortia explicitly showed that a large fraction of splicing changes associated from the knockdown of RBPs resulted from the indirect effects (Van Nostrand et al., 2020). The PLSR model identifies CSFs of EP splicing which can potentially act through direct binding as well as indirect regulation.

7) It seems like authors assume that splicing factors always enhance exon inclusion and thus have positive correlation between exon PSI and SF expression. But this assumption is not true, SF might suppress exon inclusion and effect of some of factors depends on position of its binding to mRNA (upstream or downstream of exon).

We regret giving the misleading impression. We however make no assumption regarding the directional role of splicing factors in the regulation of splicing. We have focused on the positive regulators of EP splicing, since we deemed those to be potentially over-expressed in cancer (consistent with increased EP splicing in cancer), and therefore could be potentially targeted therapeutically. The reviewer is correct in that, the CSFs, as we define them, could be either positive regulators of EP events or equivalently, negative regulators of EN events. Likewise, SFs with significant negative regression coefficients in our PLSR model could be negative regulators of EP events. Accordingly, we expect SFs with negative regression coefficient to decrease in cancer. We confirmed this trend and have added a Supplementary Fig. S4C, as shown below. We have now clarified this point more explicitly at line 361 and line 366.

Minor issues

8. L88 I believe that “EN” is “embryonic negative” but it is first used in line 88 and not formally defined

We apologize and appreciate the reviewer for pointing this out. We have made the change at line 90.

9. L87 “transmembrane-region domain (ND)” – should it be TRD instead of ND?

We apologize and appreciate the reviewer for pointing this out. We have made the change at line 89.

10. L117 - “Supplementary Fig. S1A” I would suggest to show different stages in different colors, now it is hard to notice stage clustering

We thank the reviewer for this suggestion and have now used the suggested color schema in Fig S1A as shown below:

11. Fig 1B colors denote cosine similarity, but it is not described between what and what the similarity is calculated

We are sorry for the confusion. The calculation of cosine similarity was explained in the methods section in the methods section. To avoid confusion, we have added an additional line in the legend of Fig. 1B at line 126 to explain the cosine similarity.

12. L129 “exon with its cognate gene” – I’m not sure that “cognate gene” is a widely used term. I would suggest to use “exon host gene”.

We appreciate this feedback and have changed the word cognate gene to host gene throughout the manuscript.

13. L144 “EP and EN exons exhibit mutually exclusive inclusion patterns in the pre- and postnatal stages (Fig. 1E, and Supplementary Fig. S1B)” – “mutually exclusive exons” is specific and relatively rare type of AS where only one of set of consequently placed cassette exons is included in mature mRNA. Authors clearly mean something else, so I would suggest to not use “mutually exclusive” to evade confusion. Additionally, plots provided clearly show that for both exon types (EP and EN), in most cases both isoforms are expressed in both stages (pre- and post-natal). In case of mutually exclusive expression these distributions should collapse to 0 and 1.

We do agree that the word mutually exclusive has a specific meaning in the literature of the alternative splicing. We have changed the word to ‘opposite’ at line no. 150. We thank the reviewer for suggesting this change.

14. fig 2B, does colour shows correlation coefficients? Authors should specify type of correlation coefficient used (Pearson, Spearman, ets) and what values were used for “cancerSEA hallmark gene set” (mean log(CPM)?). What dataset was used to calculate this correlation coefficients? Is it correlation along development?

We apologize for this lack of clarity. Colors indeed represent the correlation coefficients, and this correlation is Pearson’s correlation coefficient computed using TCGA cancer datasets. For each hallmark gene set from cancerSEA, we used the median expression of $\log_2(\text{tpm} + 1)$ transformed gene expression values. We have added the necessary details at line 216 in the legend of Fig. 2

15. What is shown on figure 2D bottom? If it is “proportion of EP events associated with...” should it be just a single value (for each tissue)? In this case it should be shown as barplot. How EP events associated with better (or worst) survival were defined? Shouldn't their proportions sum up to 1?

We are sorry for the confusion. In Fig 2D, each data point is a cancer type from our pan-cancer cohort comprising of 20 different cancer types, and the y-axis is the proportion of EP events (either specific or common) having better or worst survival prognosis in each cancer type. The worst and better survival was defined based on the hazard ratio obtained from the cox regression as follows; worst survival: $HR > 1$ and $FDR < 0.3$, better survival: $HR < 1$ and $FDR < 0.3$. We have edited the legend of Fig 2D to make these clear at line 221.

16. I214 “We found that the shared EP events were a significantly stronger predictor of pan-cancer patient survival as compared to tissue-specific EP events (Fig. 2D)” – all comparisons on figure 2D have p-value above 0.05 and, thus, not significant

We thank the reviewer and appreciate this concern. Our hypothesis, in this analysis, was that a greater fraction (y-axis) of common set of EP events would result in worse prognosis as compared to the fraction resulting in better prognosis. Accordingly, the one-tailed test yields a p-value of ~ 0.035 , suggesting that common set of EP events are better predictor of pan cancer survival. We have clarified the directionality of the test p-value at line 234. Further, we note that in the present analysis, a less stringent FDR threshold ($FDR < 0.30$) was used to ascertain if an EP event resulted in better or worst prognosis. If we use a more stringent threshold ($FDR < 0.20$), even the two tailed p-value become significant, as shown in the following plot.

17. there are no panel label for 2E

We apologize and have added a label for panel 2E.

18. I317 “As expected, the top positive predictors of EP events identified based on their coefficients in the PLSR model, which we refer to as critical splicing factors (CSFs, Supplementary Table S5), had higher expression during the prenatal stage of development, and underwent significant upregulation in their corresponding cancer (Fig. 4B)” – why is it expected? Different SFs might either activate or suppress exon inclusion. In latter case CSFs should exhibit high expression in healthy samples and low in

embryonic or cancer.

This concern is related to comment no. 7. As mentioned earlier, we have focused on the positive regulators of EP splicing, as inferred from their positive regression coefficient in PLSR model, because positive regulators are activated in cancer and therefore represent plausible drug targets. Positive regulators of EP splicing, by design, co-vary with inclusion of EP exons during embryonic development and given the higher inclusion of EP events during pre-natal stages and in cancer, we expect the positive regulators of EP events to follow a similar trend. As we have shown in response to comment no. 7, the splicing factors with significant negative regression coefficients indeed follow a reverse trend as compared to CSFs, as shown in newly added Supplementary Fig S4C.

19. I would suggest to show odds ratios (fig 2A for instance) in log2 scale

We have changed the representation of odds ratio to log2 scale.

20. if EP are not included in health GTEx then their dysregulation (lost of differentiation) can result in increase only. How $dPSI(\text{cancer} - \text{healthy})$ depends on $PSI(\text{healthy})$?. If I understand S2B correctly it only shows that exons with moderately low inclusion ($PSI < 0.3$) do not have strong bias toward increased inclusion in cancer. But it do not exclude the possibility, that part of EP are not included in healthy adults ($PSI \sim 0$) and then any dysregulation for them should results in increase of their inclusion.

We thank the reviewer for this comment and attest that their understanding of Fig S2B is correct. We wanted to rule out the NULL expectation that any exon with low inclusion in normal tissue is more likely to have higher inclusion than lower inclusion in the corresponding tumor. But we agree that our control still leaves out the possibility that EPs with near zero inclusion in healthy tissue may only increase their inclusion in cancer. To address this, we removed the EP exons which had near zero inclusion in healthy tissue ($PSI < 0.05$ in $> 80\%$ of the samples) and assessed the enrichment of remaining EP exons among the cancer increased events. As shown below, the remaining set of EP events is still over-represented among the events which are increased in cancer (red bars, right side panel). We have now added this at Line 194 and Supplementary Fig. S2E

21. What do figure S2B show? What is the universe (all exons considered in analysis)? What are two factors compared by fisher test? What exactly was sampled randomly? What is the purpose of random sampling and may it be replaced by confidence intervals calculated on whole sample? What conclusion was drawn from this analysis (I186)?

NB: The figure in question is now S2D.

We apologize the lack of clarity. The Fisher table is defined as follows:

Let U be the universe of all alternatively spliced (AS) exons from which the N number of EP events were drawn. Let S be a set of N randomly chosen AS exons from U with low PSI (< 0.3) in GTEx.

	Increased in cancer	Not increased
S		
U minus S		

The plot in Fig S2D show the odds ratio distribution for 500 such random samplings of N exons.

The Fisher table for EN control is defined analogously where U is the universe of all alternatively spliced (AS) exons from which the N number of EN events were drawn, and S is a set of N randomly chosen AS exons from U with high PSI (> 0.7) in GTEx.

The motivation behind this analysis is as follows. In our analysis, EP exons tend to have low inclusion in healthy GTEx samples and therefore are naively expected to have increased inclusion in cancer. We wanted to assess whether the observed increase of PSI for EP events was above and beyond this baseline expectation, and hence the control for low PSI in GTEx. Scenario for EN exons is analogous. As the distributions in Fig S2D show the range of Odds ratio for controlled random exons never reach the observed odd ratio for EP and EN exons. The conclusions drawn from this analysis is that the observed enrichment of EP among the events increased in cancer is not simply because of their low baseline inclusion level.

Further, we have now added the confidence intervals to the Fig 2A.

22. I327 “In addition, a gain in CSF expression is significantly associated with worse patient survival” – for brain median for CSF is close to 1, that mean no effect, right? in this case rather higher expression of non-CSF seems to be associated with better prognosis. So the statement is not correct, there are no tests for difference from 1. Fig 4D just shows that increase in CSF expression is worse than increase in non-CSF expression.

Thanks for this comment, and we regret misleading representation. Please note that the boxplots show the overall distribution of effect sizes. As the reviewer would appreciate, the distribution for CSFs is skewed towards positive hazard ratios, hinting that there are far more CSFs which show poor prognosis of cancer patients as compared to any other splicing factors. To make the representation clearer, instead of boxplots, we have now made the barplots, showing the proportion of hazardous splicing factors (i.e. effect size > 1 and FDR < 0.3) in each of the category (i.e CSFs and non-CSFs). The plots show that for almost 30-50% of the critical positive regulators of EP events, their higher expression negatively impact the survival of cancer patients, which is a significant enrichment over splicing factors not-critical for EP splicing (Odds ratio > 2 in all cases and p-value < 0.001). We have now replaced the boxplots with these bar-plots to make the results clearer.

23. Fig 4A shows that model predicts well exon inclusion in cancer, but significant questions are: a) whether the model prediction is better than a trivial one (for instance mean (across all developmental stages) PSI) b) can model predict cancer-related changes, that is whether difference between prediction for cancer and healthy samples correlates with observed changes. If answer to a) is “no” then model is trivial if answer to b) is “no” than model makes no sense for studied processes (development and cancer

Thanks for these comments. We are bit unclear about “a”. We model the median inclusion level of EP events given the expression of splicing factors in each developmental timepoint and predict the median EP inclusion levels in TCGA in a sample-specific fashion given its sample-specific SF expression values. If we understand the reviewer, if we simply use the mean EP inclusion across developmental timepoints and use that as a proxy prediction in TCGA, it will be same for every sample and therefore will not be correlated with the actual sample-specific EP levels in TCGA.

Regarding “b”, the reviewer makes an interesting point, i.e., whether the model can predict the cancer-associated changes. We had already shown (Fig 4A) that the PLSR model trained on developmental gene expression data of the splicing factors could predict the median inclusion of EP events across cancer samples. We have now repeated this for GTEx and as shown below the model-predicted median EP inclusion levels in GTEx samples correlate well with the observed median EP levels.

Given that the model can predict sample-wise median EP inclusion levels in both GTEx and in tumors, one would expect that to the extent the observed EP inclusion levels can distinguish normal from samples, so can the predicted EP inclusion values. And indeed, it does. The receiver operator characteristic (ROC) curves below show that the predicted EP inclusion values can distinguish normal from cancer, extremely well in brain and fairly well in liver and kidney. Therefore, we believe that our model is not trivial and captures the cancer associated changes in splicing. We have added these addition analysis at line 356 and Supplementary Fig. S4A,B

24. L329 "we tested if the EP inclusion level is lower in tumor samples bearing nonsense (inactivating) mutations in CSFs. We first identified all SFs whose mutant samples have lower EP inclusion than the wildtype samples" – this looks like authors selected only cases that confirms their hypothesis... need to be clarified

We are sorry for the confusion. We tested **all** splicing factors which had non-sense mutations in the TCGA cohorts in at least one sample, and compared the PLSR regression coefficients between the splicing factors which increase or decrease the EP inclusion level, as showing in Fig 4E. We expect that SFs which decrease the inclusion of EP events to be detected by our models, i.e they should have positive regression coefficients. We have clarified this around line 381.

25. L332 "significantly higher (and positive) regression coefficients as compared to the other SFs (Fig. 4E)" – regression coefficients between what and what?

The regression coefficient is the coefficient for each splicing factor obtained from the PLSR regression where response variable is the median splicing of all the EP events and explanatory variables are the expression of splicing factors. We have edited the line 384 to avoid this confusion.

26. L359 "there is a remarkable association between the doubling time and the EP/EN inclusion levels across cell lines derived from the organ-specific cancer type (fig 5A)" – all p-value of fig 5A are above 0.05, then there are no significant association. BTW, kidney EP have positive slope but negative Pearson correlation coefficient, that is weird.

We appreciate this comment. Please note that in Fig 5A, we have plotted the median inclusion of all EP and EN events against the proliferation rates of the cell lines. Since EP/EN exons are involved in many other processes beyond proliferation, we have now taken an alternative approach to ascertain if the EP exons were preferentially utilized in the rapidly proliferating cell lines. Toward this, in each tissue, we

obtained all the alternatively spliced exons which were substantially positively and negatively correlated ($|PCC| > 0.5$) with the doubling time of CCLE cell lines and assessed the significance of their overlap with EP and EN exons using Fishers' test. As shown below, in Brain and Liver, we find that EP exons were significantly over-represented among the exons which were negatively correlated with the doubling time of CCLE cell lines at $FDR \leq 0.1$. This means that the EP and EN exons were more likely to be related to proliferation in the expected direction than null expectation.

However, in our manuscript, we acknowledged that these associations do not hold true for the case of kidney. While cell lines are standard choice to model several diseases, they do not capture the in vivo complexity. In our analysis though, we derived the EP and EN exons from the developing human embryos, and therefore it is remarkable that rapidly proliferating CCLE cancer cell lines indeed have higher usage of EP exons and lower usage of EN exons in brain, suggesting a conserved cell-intrinsic links between splicing and proliferation. We have moved Fig. 5A to the Supplementary Fig. S5A and use the above barplots as the main figure (Fig. 5A), and have explained these changed results at line 464 and line 556.

These results are further evident in the plot below showing the distribution of cross-cell line correlations between the proliferation rates and inclusions level for EP, EN and rest of the exons, suggesting the EP exons tend to be negatively correlated with the doubling time of CCLE cell lines (and vice versa for EN exons) in brain and liver.

Apart from that, the discrepancy between slope and the corresponding correlation coefficient results from using the Spearman correlation, whereas slope is based on linear modelling which is consistent

with Pearson correlation coefficient. We regret this mistake and have made the necessary corrections, and make it clear the type of correlation coefficient used in the legend of the Supplementary Fig. S5A.

27. What is the difference between 4A and 5B?

Figure 4A is for TCGA patients and 5B is for CCLL cell lines. We have clarified this in the legend of figure 5B at line 489.

28. L376 “the greater was the dependency of the cell line on that SF” – how the “dependence of the cell line” on SF was measured?

The dependence of a cell line on SFs (or more broadly, genes) were measured by the genome-wide loss of function screens followed by cell viability assay as performed by the DepMap consortium (<https://depmap.org/portal/depmap/>).

29. What data was used for fig 5c? Are they described in “Single cell validation.” section of methods? If so the data were obtained from multiple datasets and should be confounded by batch effects than makes comparison unreliable. Additionally, authors use very large symbols on 5c, so only few cells from the top layer are visible, rest is hidden behind them. Visualization should be changed in such way to make most of cells visible.

We appreciate this feedback and have decreased the size of symbols in Fig 5c. The datasets used for Fig 5c are indeed described in the methods section. Additionally, as we mention in response to the comment no.3, we controlled for batch effects by using batch ID as co-variate while scoring the activity of CSFs in tumors relative to the normal samples. We have clarified this at line 951.

Methods:

30. L593 “In 5-dimensional PC space (first 5 PCs explain ~65 % of variance), each timepoint occupies a unique coordinate based on the PC scores” what do authors mean under “timepoints” here?

Timepoints refer to developmental stages at which the transcriptomes were sampled.

31. L604 “For each exon, we selected the significantly positively or negatively correlated KEGG pathways based on the Benjamini-Hochberg FDR threshold of 0.05.” - how exactly BH correction was applied? Was it done for each exon or each pathway independently? Or was it performed for all exon*pathways pairs simultaneously? The former approaches could cause underestimation of FDR.

FDR correction was performed for a set of p-values, each representing the significance of correlation of a single exon’s inclusion with the activity of each of the KEGG pathway (N = 332). Since the objective of this experiment was to identify the pathways which correlated significantly with an exon, there were 332 tests performed for each exon, which were corrected for. We have clarified this at line 784 in the methods section.

32. L625 - “Therefore, we selected the events which were at least 2 standard deviations away from the mean of their distribution in the corresponding GTEx normal samples in a consistent direction (i.e. increased or decreased) in at least 15% of the cancer patients” – same exon can be both increased or decreased by this definition. It is better to use dPSI threshold, since changes below 5 (10%) are usually

considered as noise. Some estimate of statistical significance is necessary.

We thank the reviewer for this crucial comment. Note that we chose this metric to identify cancer associated changes based on a previous publication (Kahles et al., 2018). Such a consideration is required to better cope with inter-tumor heterogeneity. Analogous to mutations, a specific splicing events might not get mis-regulated in all samples. Hence the events which deviate significantly (i.e. 2 SD away from their normal mean) in a sizeable number of patients (i.e. 15 % in our case), should be robustly altered events and might be relevant for cancer progression. As pointed by the reviewer, we do not consider the exons which showed increased as well as decreased trend as per this definition. However, to address the reviewer's concern more directly, we recalculated the differential splicing events in cancer using Wilcoxon test between TCGA and GTEX ($|\Delta \text{PSI}| > 0.2$ & $\text{FDR} < 0.1$) and recomputed the enrichment of developmental splicing events. Out of 12 total comparisons, 10 were significant at an FDR threshold of 0.01 and 11 were significant at an FDR threshold of 0.25. This shows that our conclusion of recapitulation of the embryonic splicing in cancer are robust to the specific choice. We have now added this additional commentary at Line 200 and Supplementary Fig. S2G.

Reviewer #2, expert in RNA splicing and development (Remarks to the Author):

In their manuscript titled “Broad misappropriation of developmental splicing profile by cancer in multiple organs,” Singh et al. present a comprehensive, systems-level study that identifies developmentally-regulated splicing events in liver, kidney and brain and explores the misregulation of these exons in the corresponding, tissue-specific cancer. The study integrates information from a wide-variety of databases and public data collections to conclude that tumors show a progressive reversion to embryonic splicing patterns, identify potential splicing factors regulating these events and transcription factors controlling their expression, and suggest possible existing drugs that may target these factors and thus have putative therapeutic potential.

Major

As someone who studies developmental splicing, I was intrigued by the large number and wide variety of predictions that can be explored in future studies in terms of splicing regulators, transcriptional regulators, spliced exons and drugs that might influence these splicing outcomes and associated cellular processes. I think this is an interesting paper that puts forth multiple, important hypothesis that should be experimentally tested, and will be of interest to the oncogenesis community, but also developmental biologists and the splicing community. That said, the central tenet of the manuscript is not new, and the authors themselves cite multiple studies suggesting important roles for splicing and reversion to the embryonic state in various cancers throughout their manuscript. The manuscript also relies completely on systematics and correlation, which although it indeed provides strong motivation to pursue novel hypotheses, doesn't itself provide new data verifying those hypotheses. Below I provide several suggestions to improve the study and manuscript.

We thank the reviewer for their interest in this work and providing us many thoughtful suggestions which we have incorporated in the revised manuscript. Based on reviewers' suggestions, we have analyzed the mutation data of CSFs, show that CSFs are indeed mutational drivers in multiple cancer types, and extensively searched the existing literature to identify several examples to support the role EP/EN events encoding TRD, ND and WD40 domains in altering protein function and cancer progression. These new analyses are added in the revised manuscript. Below we provide a point-by-point response to the reviewers' comments, suggestions, and questions.

1. Please provide a full list of all abbreviations and define them when they are first introduced in the text. Please also check that abbreviations in figures are defined in the legend (ie LHC in Fig. S2 or vely in Fig. 1D).

We appreciate this feedback from the reviewer and have carefully expanded the used abbreviations. Additionally, at line 31, we have provided a list of commonly used abbreviations throughout the manuscript to help the readers.

2. The authors should somewhere address the relationship between “cause” and “effect” in terms of the embryonic splicing profile and misregulation of splicing factors in the cancers they examine. Based on Fig. S2 C & D, there is a progressive misregulation of splicing in early and late stage cancers. Does this reflect that splicing is therefore not a cause of the oncogenesis, but rather an effect of the changes that allow oncogenesis in the first place? For example, how many early stage cancers already contain inactivating mutations in key CSFs? Is there also a corresponding increase in CSF mutations in late stage cancers? In muscle disease, for example, mutations in splicing factors are causal for and precede the

disease, but it is still poorly understood how the wide variety of splicing changes and the reversion to embryonic splicing patterns on whole actually contributes to the disease. Can the authors provide such insight in the cancer context? Another aspect of this is how many CSF genes are actually activated in malignant cells? Is activation of 1 CSF sufficient, or do multiple CSFs have to be targeted to promote malignancy?

This is a wonderful comment regarding how one could interpret the splicing changes (or any molecular change for that matter) in terms of cause and effect. As the reviewer alludes to, at this point, we can only speculate if the splicing changes are a cause of carcinogenesis or whether these events are concomitant with a much broader reprogramming of the cell drifting towards malignancy. But considering that our findings implicate known master regulators (MYC, FOX family) in driving the splicing changes, it is likely that the embryonic reversal of cancer splicing co-occurs with much broader transcription and epigenetic reprogramming and is likely downstream of the other cancer driver events. We have added this additional commentary at line 692.

To directly assess the issue of causality, we analyzed the total number of mis-sense mutations among CSFs, in the early and late stage of cancers, as pointed by the reviewer. Since we are focusing only on the positive regulators of EP splicing, we expect those to be activated in cancer and hence should not have inactivating mutations. We observed that on an average, the set of CSFs exhibit mutations already in the early stages, and do not emerge only at a later stage.

We have added this new analysis at Supplementary Fig. S4H and discussed at line 666.

Although, in this analysis, we did not have access to longitudinal patient data, but the results hint at the causal role of CSFs in initiating the cancer. This is consistent with the previous reports where several splicing factors have been reported as major drivers in multiple cancers (for instance SF3B1 and U2AF2 in lung and pancreatic adenocarcinomas). We further obtained a previously reported list of SFs that drive human malignancies (Seiler et al., 2018), and found that CSFs are much more likely (OR > 2, p-value < 0.003) to be the cancer drivers as compared to the other splicing factors, further supporting the causal role of CSFs in initiating cancer.

This new analysis is added at Fig 4G and discussed at line 426.

As far as the question of number of CSFs that should be altered to promote malignancy is concerned, we believe that activation of a single CSF might not be sufficient to reprogram the transcriptome-wide landscape of alternative. The splicing regulatory networks are highly complex and often exhibit cooperativity and cross-regulation (Fu & Ares, 2014)

To get an estimate of the number of factors which are activated, for each cancer type, we computed the proportion of CSFs in each sample whose FC > 1.5 compared to their median expression across the corresponding GTEx normal tissue, and plot the density distribution of the proportions below:

Based on this, we observe that a median of 47%, 32%, 16% of the CSFs were activated in brain, liver and kidney cancer patients, respectively. This is consistent with the fact that brain had most significant enrichment of the EP followed by liver and kidney. Therefore, an up-regulation of multiple CSFs appears to happen in a coordinated fashion and is likely the result of upstream oncogenic events driven by Myc and Fox family of transcription factor regulators or contributes to the broader reprogramming involved in the malignant transformation of the cells. We have discussed this new analysis at line 434 and Supplementary Fig. S4G

Having said that, we would like to note that the analysis of somatic mutations in TCGA cohorts (Fig 4E) and shRNA knockouts in HepG2 cell line (Fig. 4F) suggest that the deletion of an individual critical splicing factor can still have a small (albeit significant) effect on the inclusion level of EP events, but it

might not be sufficient for global reprogramming of splicing, which is only possible *via* activation of multiple CSFs.

3. Figure 3 – The underlying finding that EP/EN exons contribute to alternative isoforms of proteins containing TRD, ND and WD40 domains is very interesting. Can the authors provide examples or do a transcript-level analysis to better characterize or to confirm that the exon switches do indeed cause protein-level coding differences? Have any of the identified exons already been reported to be regulated in cancer or development, that the authors can use as a “marker” to support that the ES/EN exons they identify actually contribute to isoform switches that are physiologically relevant? The correlation matrices in Fig 3C & D to me are somewhat redundant, because at least in my experience, we focus on either BP or MF terms because of the large overlap in genes in related terms between the two annotations. The enrichment is performed on the same list of terms, so I wouldn't expect anything different. I do agree that practically, such a plot (ie Fig3D) would help you focus on 3 processes instead of dealing with 18+ GO terms.

We thank the reviewer for this interesting comment. First, a general remark. We employ the broadly used exon inclusion quantification tool – SUPPA2, which, by design quantifies exon inclusion based on the transcripts that include the exon, and therefore, any observation we make at the exon level will be reflected at the transcript level.

To address this concern if the EP and EN events can indeed cause protein level coding differences, we have identified several examples from our list of the genes which host an EP or EN exons containing the TRD, ND and WD40 domains, where alternative splicing of these exons can have major impact on the function of protein and physiology of the cells. These examples are compiled in Supplementary Table S9. We highlight a few examples below.

1. APAF1 is a gene which encodes an apoptotic protein and hosts an EN exon encoding WD40 domain in the developing fetal brain. Interestingly, a previous report shows that APAF1 is alternatively spliced in prostate cancer cell lines, producing a shorter isoform called APAF1-ALT and lacking WD40 domain (Ogawa et al., 2003). Moreover, this shorter isoform impedes the induction of DNA-damage induced apoptosis in cells, thereby allowing cells to acquire DNA-damage induced resistance against treatment. Our work elucidates that such a splicing change is a general mechanism to control apoptotic function during normal embryonic development as well as cancer.

2. FLVCR1 is a gene which encodes a heme transporter and hosts an EN exon encoding TRD domain in brain. Previous work shows that various alternatively spliced isoforms of this gene lacking the transmembrane domains are expressed in the case of Diamond blackfan anemia (DBA). Importantly, the patients with DBA have an elevated risk of neoplastic growth (Vlachos et al., 2012). Therefore, our results imply that the regulation of iron metabolism by controlling its transport through alternatively spliced FLVCR1 gene is crucial for developing human brains (Lipiński et al., 2013) as well as cancers.

2. MCL-1 is a gene which encodes an anti-apoptotic protein and hosts an EN exon in brain (Supplementary Table S8, exon ENSE00000959746). However, a previous report shows that an alternatively spliced shorter isoform of MCL-1 protein (called MCL-1S) switches its function from anti-apoptotic to pro-apoptotic (Bae et al., 2000). Interestingly, the exon involved in this event is an EN exon in brain encoding a nitrosylation domain. This hints that both embryogenesis and cancer uses this splicing switch to control apoptosis.

We have discussed these examples at line 608.

We do agree that overlap between corresponding molecular and biological processes is expected. We perform this analysis precisely due to the reasons stated by the reviewer. We wanted to be sure that indeed the genes contributing to the enrichment of biological process (for instance N-glycosylation) were same as those contributing to the enrichment of the molecular function oligosaccharide transferase activity. Only through such correspondence analysis we could ascertain that any enriched molecular process is indeed related to the enriched biological processes. This correspondence helped us to draw our conclusions related to the role of TRD splicing in N-glycosylation and ND splicing in retrograde-transport.

4. Is it possible to provide experimental data for the OST complex, or GTPase activity or ARL1 to confirm that the correlations and predictions from the systematic analysis actually have coalesced on valid targets? The analysis looks compelling, but from experience, when you start working with individual genes from such an analysis you do not always get the expected result. The impact of the manuscript would be stronger if the authors could show new experimental data. More generally, are there possibly reported proteins or events that can confirm some of the systematic findings and can be used as a “case study” to support the conclusions, in lieu of new wet-lab experiments?

We thank the reviewer for this very important comment. As far as the experimental data is concerned, the broad role of alternatively spliced transmembrane domain in the regulation of N-glycosylation, and nitrosylation domain in the regulation of cellular transport during development and cancer is a quite novel proposition by us. We are in the process of establishing collaborations to experimentally validate our findings in cell line and animal models, specifically the role of EP/EN events in the regulation of OST and for RAS superfamily GTPases. This is a multi-year project and out of the scope of current work.

But as reviewer pointed to look for previously reported proteins, we are glad to have found examples which support our conclusions related to TRDs and NDs as discussed below:

1. Alternatively spliced NDs and GTPase activity in RHOA: Nitrosylation is crucial modification involved in the upregulation of guanine nucleotide exchange rate and its GTPase activity (Lin et al., 2018; Raines et al., 2007). Our analysis suggests that the loss of nitrosylation domain via alternative splicing would result in decreased GTPase activity. Intriguingly, we found a previous study which reported the loss of GTPase activity in one of the small GTPase encoded by RHOA gene due to alternative splicing (Miyamoto et al., 2018). We spotted that RHOA contains an EN exon encoding nitrosylation domain in liver and kidney (Supplementary Table S8), and is the same exon reported by Miyamoto et al., 2018. This example provides a strong case to support the broader role of alternatively spliced ND in the regulation of GTPases during development and cancer.

2. Alternatively spliced TRDs and change in Integrin functionality: As far as TRDs are concerned, the general idea that the lack of transmembrane region can change in protein function is expected to affect almost 40% of the human membrane bound proteins (Mittendorf et al., 2012; Xing et al., 2003). One such protein is an integrin protein encoded by ITGA2B gene. The ITGA2B gene encodes a transmembrane domain with exon no.29, which is an EN exon in brain. ITGA2B expresses a truncated isoform lacking transmembrane and cytoplasmic domain in multiple cancers including melanoma, prostate cancer, and leukemia (Jin et al., 2007; Trikha et al., 1998). Similar to our proposed role of the alternatively spliced TRD in the subunits of OST complex, one would expect that lack of TRD would result

the expulsion of ITGA2B from the plasma membrane. Indeed, past research has shown that the truncated isoform of ITGA2B is secreted into the ECM and breaks the adhesion of cells, promoting migration and metastasis (Jin et al., 2007). Our findings implicate that this mechanism of cellular migration is not only employed by cancers, but the migrating cells of developing embryos as well.

We have now mentioned these examples at line 303 and line 324.

Additionally, we have curated a table of several examples (Supplementary Table S9), where the alternatively spliced EP and EN exons have been shown to have major physiological effects relevant to diseases such as cancer.

5. Figure 5 – the authors state the association between EP/EN inclusion is “remarkable,” but especially in kidney the R is not remarkable, and only in brain is the regression/relationship statistically significant. The results in Fig. 5C are also not so clear-cut as the authors state in the text. I do not doubt that some ES events are linked to changes in proliferation, but the reality seems to be more nuanced and heterogenous. Can the authors define a narrower set of exons or even CSFs that are tightly linked with proliferation? This is also somewhat relevant to a point they raise in the discussion in lines 540-545. It is unclear that targeting a single EP exon would be effective, but is there a way to predict the smallest set of exons that would have to be targeted to produce an effect? How heterogenous are the different cancer cell lines in this respect?

We appreciate this feedback. We apologize for lack of clarity in describing these results. As far as kidney is concerned, we did acknowledge that the associations are true only for brain and liver cancer cell lines. However, we do agree that regression coefficients are not significant in all cases pertaining to liver and brain. As the reviewer hinted, only a subset of EP exons might be linked to the proliferation rate. An important point to note is that the results presented in Fig 5 were obtained using the median inclusion level of all the EP and EN events. Therefore, we asked a different question, namely, whether the EP exons were over-represented among the exons which were negatively correlated with the doubling time of CCLE cell lines (i.e positively correlated with the rate of proliferation). For this analysis, in each tissue, we obtained all the alternatively spliced exons which were substantially positively ($PCC > 0.5$) and negatively correlated ($PCC < -0.5$) with the doubling time of CCLE cell lines and assessed their overlap with EP and EN exons using a Fishers’ test. As shown below, in Brain and Liver, we find that EP exons were significantly over-represented ($FDR < 10\%$) among the exons which were negatively correlated with the doubling time of CCLE cell lines.

The EN exons, on the other hand, were significantly overrepresented among the exons which were positively correlated with the doubling time of CCLC cell lines. But in kidney, we do not observe the enrichment of EP/EN exons among the proliferation related exons, owing to the heterogeneity cited above. We have explained these results at line 464 and have replaced the Fig. 5A with these bar plots of enrichment.

The same result is evident when we plotted the overall distribution of the correlation between inclusion level and doubling time of CCLC cell lines, where EP exons tend to have negative correlations with doubling time in brain and liver.

6. Can the authors comment on EP/EN use in the CSFs they have identified? Splicing factors are reported to themselves be heavily spliced, in particular in both the developmental and oncogenic context. Are there any relationships in the data supporting that specific isoforms of CSFs are associated with oncogenesis?

We again thank the reviewer for this important point. We analyzed and observed that indeed, in general, splicing factors were overrepresented among the host genes of EP and EN events (Supplementary Fig. S4I). As many as 25% of CSFs in brain, 19% in liver, and 16% in kidney were the host genes of an EP event. This corresponded to > 2.5-fold enrichment Supplementary Fig. S4I in all cases as compared to random set of genes. Likewise, 27%, 20% and 11% of the CSFs in brain, liver and kidney were shared with the host genes of EN events.

Additionally, we observed that CSFs were more likely to host and EP/EN exon as compared to rest of the splicing factors (compare blue bars with red in each panel). Together, these observations indicate that splicing factors are dominant targets of embryonic splicing and CSFs having greater enrichment of embryonic splicing events as compared to the other splicing factors. We have added an additional Supplementary Fig. S4I and lines 673 to describe these results.

7. In the methods, the authors mention that the cancer-specific splicing events are limited to exon skipping events. Is this also true for the developmental events? Can the authors comment if their findings only are valid for exon skipping events, or for other types of AS events?

We regret the lack of clarity. We do not imply that cancer-specific splicing events are limited only to exon-skipping events. We chose to focus on exon-skip events as those are better annotated in transcriptional databases and are easier to interpret functionally. We have now mentioned this explicitly at line 811.

Minor points:

8. -Line 87: typo: transmembrane-region domain (ND) should be (TRD)
 We have made the desired change at line 89.

9. -Line 100: Is this MYC and FOXM1, or does the MYC, FOXM1 mean another TF is missing from the list?
 We apologize and have made the desired change at line 102. It should be MYC and FOXM1. We appreciate the reviewer for pointing this out.

10. -Lines 133-136: This sentence isn't logical. Enrichment of EP pathways in terms related to oncogenesis doesn't validate a role for EP pathways in embryogenesis and organ development. At best, it just reflects that both cancers and developing tissues are actively proliferating, and the approach the authors used accurately identifies these KEGG pathways.

We are sorry for the confusion. As rightly pointed by the reviewer, the main objective of using GO annotations was to test if our approach correctly identifies the preferentially embryonic pathways. We have made the desired change at line 139, 141 stating it more clearly.

11. -Lines 245-248: This is an interesting point, but is not clearly stated. If I understand correctly, the analysis identifies EP and EN exons as contributing to coding TRD, ND and WD40 domains across all of

the tissues, BUT the genes that are identified are tissue specific. I do not follow how this is then related to “multiple genes coordinate the splicing of these domains across tissues”, because identification of the CSF’s comes at a later point in the manuscript.

This is an interesting point and we do agree that the statement “multiple genes coordinate the splicing of these domains across tissues” can be confusing in its current form. The reviewer has correctly understood that although similar domains are enriched, the associated genes are tissue specific. What we meant was that multiple genes (which are specific to each tissue) coordinately splice in/out their TRDs, NDs, and WD40 domains. The words “multiple genes coordinate” refers to the genes which are involved in the splicing events (and not their regulators). To avoid any confusions, we have rewritten that line as follows: “multiple host genes of EP and EN exons coordinately spliced in and out these domains across tissues” (now line no. 267).

12. -Legend Fig. S1 – Plot C is missing in the legend, and D and E are mislabeled in the legend as C and D. We thank the reviewer for kindly pointing this out. We have made the desired rectifications in Fig S1.

13. -Legend Fig. S2 – Panel A is a dotplot, not a biplot. Is this a typo in the legend? Or is Panel A incorrect?

We agree that Fig S2 is not a biplot, as we are not showing the loadings. We have modified the legend of Fig S2 and changed the name ‘biplot’ to ‘scatterplot’.

Reviewer #3, expert in RNA splicing and cancer (Remarks to the Author):

The manuscript by Dr. Singh and colleagues is a very well written paper on the timely subject of cancer splicing providing valuable new data with novel and thought-provoking findings. The authors did a comprehensive bioinformatic survey of publicly available transcriptomic data to identify organ-specific alternative splicing profiles and compare them to cancer transcriptomic data of the same organs. They find that organ-specific developmental splicing is enriched in the respective cancer at a large-scale. They also identify PFAM domains affected by the alternative exons and splicing factors predicted to regulate these splicing events. This validation on a more global level is much needed in the field as there has only been anecdotal evidence of individual developmental splices being reactivated in cancer. In particular, their findings on splicing of EP transmembrane(TRD) and nitrosylation (ND) domains and their implications for secretory/Rab pathways and cell communication and signaling during neuronal development and cancer are quite novel and very insightful. Overall the paper is very strong in bioinformatics analysis and the findings are well presented and explained. Certain of the claims could be further strengthened by experimental validation. Nevertheless, these findings provide interesting hypothesis for experimental labs to further validate. I have just a few points/comments that need clarification:

We thank the reviewer for their interest in this work and providing us kind and instructive feedback. We have added experimental evidence to support the causal role of CSFs in the regulation of EP splicing, show that CSFs are indeed mutational drivers in multiple cancer types and extensively searched the literature to identify several examples of EP and EN events encoding TRD and ND which are previously reported to alter the protein function and enhance cancer progression. We have updated the revised manuscript with these changes. Below we provide a point-by-point response to the reviewers’ comments, suggestions, and questions.

Major

1. The authors mention they used SUPPA2 software for their splicing analysis. Did the authors try other commonly used splicing software like rMATS? If yes, where the results comparable? Some justification on their choice of software would be desirable.

We appreciate this comment from the reviewer. We acknowledge that there are numerous tools available for quantifying the alternative exon usage based on RNA-seq data, rMATS is preferred by some researchers. However, our choice of SUPPA2 was based on the careful consideration of several factors as explained below:

1. SUPPA2 can utilize precomputed TPM values of transcripts (Trincado et al., 2018). This enabled us to utilize uniformly processed TCGA and GTEx RNA-seq datasets from an elegant UCSC toil RNA-seq recompute compendium (Vivian et al., 2017), which were normalized for multiple computational batch effects.

2. SUPPA2 is faster as compared to rMATS as SUPPA2 can directly work with TPM values, whereas rMATS requires an external aligner (usually STAR 2 pass alignment) (Shen et al., 2014). This point was also illustrated in an independent benchmarking study (Muller et al., 2021). Aligning ~2500 samples that we analyzed in this study using STAR + rMATS requires several orders of magnitude greater computational time compared to using pre-processed datasets from the recompute2 project.

To address the reviewer's concern more directly, we used rMATS along with STAR 2 pass alignment for quantifying splicing in developing brain samples and assessed its concordance between SUPPA2. In each sample we compute the exome-wide correlation between the SUPPA2 and rMATS-estimates PSI. As shown in the boxplot below, median correlation between the two across all the brain samples is 0.84. This suggests that SUPPA2 is largely concordant with rMATS. The same conclusions were drawn in a previous study which compared SUPPA2, rMATS, and MISO and showed that except for intron retention events, SUPPA2 was highly concordant with other two tools (Muller et al., 2021)

We have mentioned this at line 740.

2. In the PFAM domain analysis in brain and kidney it looks like the TRD, ND and most of the other PFAM domains are enriched only in EN exons. Does this mean that the TRD- and ND-encoding exons are skipped in these cancers? If so, then this is a major point that deserves more detailed discussion.

This is a great comment and to break it down, the first part of comment is that most EN category of exons had a greater number of enriched domains as compared to EP category of exons. This is indeed surprising. This would imply that, overall, a relatively larger fraction of annotated domains covary with (or involved in) processes that are active postnatally, in contrast with prenatally active processes. A general bias in functional roles of alternatively spliced domains to be involved in development related processes has been noted previously (Liu & Altman, 2003). But the differential functional underpinning of this observation relative to EP and EN is currently unclear and will require further investigation. We have now noted this at line 602.

Second question is that whether TRDs and ND-encoding exons are skipped in cancers. We would like to highlight that although both EP and EN exons were enriched for TRD and NDs, as shown in Fig S3B, the host genes of EP and EN exons containing these enriched domains are different within and across organs (in other words, it is not the case that the same set of genes are contributing to the enrichment of these domains). Therefore, our interpretation is that these domains tend to be skipped as well as included during development and cancer, depending upon their host gene and its function as well as cancer type.

3. Did the authors investigate the potential effects of the EN and EP exons encoding for TRD and/or glycosylation domains on subcellular localization of the encoded proteins? For example, if a transmembrane domain (TRD) is skipped in cancer then that would be predicted to affect the subcellular localization and the solubility of the encoded proteins. Are there any examples of TRD EPs/ENs resulting in a transmembrane protein becoming secreted? Such EPs/ENs would be expected to play a major role in intercellular signaling for example.

This is a great comment and we too believe that inclusion/exclusion of TRDs should alter the localization of proteins on the membrane. In general, alternative splicing can affect almost every aspect of the function of a protein, such as enzymatic activity, folding, degradation, and localization. Generally speaking, as the reviewer alludes to, the removal of transmembrane domains can result in the novel soluble protein isomers and as much as 40% of the membrane proteins in humans are expected to have an isoform completely lacking the transmembrane domain (Xing et al., 2003). The change in protein solubility resulting from the removal of TRDs is well known for several proteins in humans including CD1 (Woolfson & Milstein, 1994), TMEFF2 (Quayle & Sadar, 2006), and FOLR1 (Hoier-Madsen et al., 2008).

Among the list of embryonic exons identified in this study, we could identify the ITGA2B gene, where an EN exon #29 encodes transmembrane domain and various human cancers including melanoma, prostate cancer, and leukemia were shown to express the truncated form of this protein. The truncated form lacks the transmembrane domain which results in the secretion of this otherwise transmembrane protein. The secretion of this protein breaks the adhesion of cells and promotes migration (Jin et al., 2007; Trikha et al., 1998). Moreover, the skipped exon has one site for N-glycosylation (<https://www.uniprot.org/uniprot/P08514>). Therefore, such exon skipping events involving TRDs events can have major impact on the solubility and hence function of the protein.

However, to the best of our knowledge, the role of alternatively spliced TRDs in the regulation of N-glycosylation (by possibly affecting the localization of the subunits of oligosaccharide transferase) have

not been reported so far. Additionally, we observed that many of the mitochondrial and plasma membrane proteins alternatively spliced their TRDs during embryonic development, which could have a profound effect over the energy metabolism, vesicle trafficking, and cell signaling.

We have now included these additional lines at explaining these results at 303-315.

4. Lines 318-324 on Critical Splicing factors: how did the authors assess that they were critical? CRISPR essentiality database? This should be stated in the text.

We regret lack of clarity. We use the term critical splicing factors based on the PLSR regression model for predicting the EP splicing. The factors which were significant predictors the EP level in this model were termed as critical splicing factors. The developmental role of critical splicing factors was further confirmed by their role in pre-natal human development (Fig. S4D). To avoid this confusion, we have added line: “We obtained the list of splicing factors which were significant positive predictors of the median EP during embryonic development based on their regression coefficients in PLSR model (Methods) and termed those as critical splicing factors (CSFs)” at line 361.

5. Lines 330-339, causal CSFs: the effect of some of these mutations might be more subtle than lead to total CSF inactivation. Another complementary way of validating this would be to look in DepMap or other essentiality databases for CRISPR or RNAi data for these CSFs and correlate with EP levels. Or even better, do the actual experiment in the lab for the top 10 factors: that is silence them by RNAi or CRISPR and assess if the EP inclusion levels are affected.

We agree that non-sense mutations can sometimes have more subtle effects. But since non-sense mutations truncate a protein, we believe that such mutations would often result in the inactivation of proteins. This idea is consistent with the notion that tumor suppressor genes, which are inactivated in cancer, are often enriched with non-sense mutations. To experimentally validate the role of CSFs in splicing, we need the datasets which have performed genome-wide RNA-seq experiments following the gene-deletions of CSFs.

Fortunately, we found one RNA-seq dataset in liver cancer HepG2 cell line following the shRNA knockdown of several RNA binding proteins. We downloaded the raw RNA-seq datasets for shRNA knockdown of ~231 RNA binding proteins in HepG2 (a liver cancer cell line), quantified gene expression and splicing using our pipeline, and trained a PLSR model for the regulation of embryonic splicing events specifically in HepG2 based on 50 control samples. We identified critical splicing factors based on their regression coefficients in PLSR model as described in the manuscript. Very encouragingly, we observed that upon shRNA knockdown a greater fraction of critical splicing factors of EP resulted in the decreased EP splicing as compared to non-critical splicing factors. Taken together, the analysis of EP events in the developmental transcriptomes, cancer transcriptomes, mutation analysis, and shRNA-seq provides very strong evidence for the role of critical positive regulators in the EP events. We have added this new analysis to Fig 4F and at line 388 and methods section at line 920

Further, based on our findings, we are setting up collaborations with an experimental group at NIH to generate the gene regulatory network underlying the splicing modulation in cancer. However, this is a long-term project and deemed to be published separately.

6. CSFs: There are no names of the CSFs mentioned. The authors should provide the names of at least their top ranking CSFs to enable comparisons and integration with previous literature. Do these include the already known CSFs that are frequently mutated in cancer? If this is the case the manuscript could be substantially strengthened.

A great comment and we sincerely thank the reviewer for this comment as it has helped strengthen our findings. From a previous publication (Seiler et al., 2018), we downloaded the list of 119 splicing factor genes (of which 75 were common with our analysis) which are putative drivers in one or more cancer types. We assessed whether the identified CSFs are more likely to be cancer drivers compared to other 'non-critical' splicing factors. We pooled the critical and non-critical splicing factors from all three tissues and indeed found that while only 11% of non-critical factors were putative drivers across multiple cancers, 23% of CSFs were putative cancer drivers (2.4 fold enrichment, p-value < 0.003).

Next, we extended this analysis in a cancer-specific manner. We selected all the splicing factors which had hotspot mutations in specific cancer types in Michael Seiler's work, and checked the overlap with CSFs and remaining splicing factors (nCSFs) in cancer specific way. Astonishingly, CSFs had hotspot mutations more frequently as compared to non-critical splicing factors, as shown below.

The names of critical splicing factors bearing driver mutations are as follows:

Brain: CDC5L, CDK12, EFTUD2, HNRNPK, PABPC1, PCBP2, SF1, SF3B3, SYNCRIP, THRAP3, ZC3H13

Liver: EFTUD2, PCBP1, SF3B1, SNRPN, U2AF2

Kidney: SF3B1

Interestingly, some of these CSFs are known drivers of various solid and hematological malignancies. For instance, SF3B1 and U2AF2 are frequent drivers of lung and pancreatic adenocarcinomas

Also, the PCBP2 which we found to be a driver CSF in brain tumors, has been reported to promote the growth of gliomas (Han et al., 2013). Another factor, CDC5L, is known to promote gliomas (Chen et al., 2016) and bladder cancers (Z. Zhang et al., 2020).

We think that our work provides a broader biological context to the events which are downstream of these driver events, i.e, the reactivation of patterns of embryonic splicing, which potentially makes the cell more conducive to rapid proliferation, migration, and angiogenesis.

We have discussed these important CSFs at line 396 and added the new analysis at line 426 and Fig. 4G, Supplementary Fig. S4F.

7. TFs regulating CSFs: a lab experiment would be appropriate to validate a subset of these predictions. Alternatively, the authors could mine the DepMap or other gene essentiality databases and/or publicly available data for further validation.

We appreciate this feedback from the reviewers. Indeed, gene regulation is best studied experimentally. But the DepMap portal currently does not have any datasets which have performed gene knockouts followed by RNA-seq. However, in our analysis presented in Fig 6, we have used knockTF database (6B, second col.), which is a compendium of gene expression measurement following TF knockouts using RNA-seq and microarray based platforms across multiple cell lines. As we have shown in Fig 6, the transcriptional targets of MYC and FOX family of regulators were significantly enriched for CSFs relative to the other splicing factors. We have initiated long-term collaborations with experimental groups to validate our findings. We now added line 696 to highlight the knockTF results.

8. Lines 424-425: The drug repositioning for targeting TFs with FDA drugs as an approach with therapeutic potential is rather speculative without any experimental data. Unless validation data are presented such claims are rather weak and should be removed.

We agree that our claims regarding use of FDA approved drugs are speculative. We have made this limitation more explicit at line 526 and moved the Drug repositioning pipeline to existing Supplementary table S7.

Minor points:

9. Line 247: 'the' is missing before 'observed enrichment...'

We have made the change at line no. 267.

10. Line 288: 'for' instead 'of' before 'alternatively spliced ND...'

We have made the change at line no. 320.

11. There is a few other sentences where some articles like 'the' are missing before the nouns (minor grammatical errors).

We have proofread the document multiple times and fixed any grammatical mistakes. We again thank the reviewer for their interest and patiently reading our work.

12. The Tsai et al. 2015 reference is missing.

We apologize and thank the reviewer for kindly pointing this out. We have the missing citation in the bibliography and apologize for the mistake.

- Bae, J., Leo, C. P., Sheau Yu Hsu, & Hsueh, A. J. W. (2000). MCL-1S, a splicing variant of the antiapoptotic BCL-2 family member MCL-1, encodes a proapoptotic protein possessing only the BH3 domain. *Journal of Biological Chemistry*, 275(33). <https://doi.org/10.1074/jbc.M909826199>
- Bland, C. S., Wang, E. T., Vu, A., David, M. P., Castle, J. C., Johnson, J. M., Burge, C. B., & Cooper, T. A. (2010). Global regulation of alternative splicing during myogenic differentiation. *Nucleic Acids Research*, 38(21). <https://doi.org/10.1093/nar/gkq614>
- Bray, N. L., Pimentel, H., Melsted, P., & Pachter, L. (2016). Near-optimal probabilistic RNA-seq quantification. *Nature Biotechnology*, 34(5). <https://doi.org/10.1038/nbt.3519>
- Chen, W., Zhang, L., Wang, Y., Sun, J., Wang, D., Fan, S., Ban, N., Zhu, J., Ji, B., & Wang, Y. (2016). Expression of CDC5L is associated with tumor progression in gliomas. *Tumor Biology*, 37(3). <https://doi.org/10.1007/s13277-015-4088-5>
- Fu, X. D., & Ares, M. (2014). Context-dependent control of alternative splicing by RNA-binding proteins. In *Nature Reviews Genetics* (Vol. 15, Issue 10). <https://doi.org/10.1038/nrg3778>
- Garrido-Martín, D., Borsari, B., Calvo, M., Reverter, F., & Guigó, R. (2021). Identification and analysis of splicing quantitative trait loci across multiple tissues in the human genome. *Nature Communications*, 12(1). <https://doi.org/10.1038/s41467-020-20578-2>
- Gaudet, P., & Dessimoz, C. (2017). Gene ontology: Pitfalls, biases, and remedies. In *Methods in Molecular Biology* (Vol. 1446). https://doi.org/10.1007/978-1-4939-3743-1_14
- Han, W., Xin, Z., Zhao, Z., Bao, W., Lin, X., Yin, B., Zhao, J., Yuan, J., Qiang, B., & Peng, X. (2013). RNA-binding protein PCBP2 modulates glioma growth by regulating FHL3. *Journal of Clinical Investigation*, 123(5). <https://doi.org/10.1172/JCI61820>
- Hoier-Madsen, M., Holm, J., & Hansen, S. I. (2008). α Isoforms of soluble and membrane-linked folate-binding protein in human blood. *Bioscience Reports*, 28(3). <https://doi.org/10.1042/BSR20070033>
- Jacobson, M., Sedeño-Cortés, A. E., & Pavlidis, P. (2018). Monitoring changes in the Gene Ontology and their impact on genomic data analysis. *GigaScience*, 7(8). <https://doi.org/10.1093/gigascience/giy103>
- Jin, R., Trikha, M., Cai, Y., Grignon, D., & Honn, K. V. (2007). A naturally occurring truncated $\beta 3$ integrin in tumor cells: Native anti-integrin involved in tumor cell motility. *Cancer Biology and Therapy*, 6(10). <https://doi.org/10.4161/cbt.6.10.4710>
- Kahles, A., Lehmann, K. Van, Toussaint, N. C., Hüser, M., Stark, S. G., Sachsenberg, T., Stegle, O., Kohlbacher, O., Sander, C., Caesar-Johnson, S. J., Demchok, J. A., Felau, I., Kasapi, M., Ferguson, M. L., Hutter, C. M., Sofia, H. J., Tarnuzzer, R., Wang, Z., Yang, L., ... Rättsch, G. (2018). Comprehensive Analysis of Alternative Splicing Across Tumors from 8,705 Patients. *Cancer Cell*, 34(2), 211-224.e6. <https://doi.org/10.1016/j.ccell.2018.07.001>
- Lin, L., Xu, C., Carraway, M. S., Piantadosi, C. A., Whorton, A. R., & Li, S. (2018). RhoA inactivation by S-nitrosylation regulates vascular smooth muscle contractive signaling. *Nitric Oxide - Biology and Chemistry*, 74. <https://doi.org/10.1016/j.niox.2018.01.007>

- Lipiński, P., Styś, A., & Starzyński, R. R. (2013). Molecular insights into the regulation of iron metabolism during the prenatal and early postnatal periods. In *Cellular and Molecular Life Sciences* (Vol. 70, Issue 1). <https://doi.org/10.1007/s00018-012-1018-1>
- Liu, S., & Altman, R. B. (2003). Large scale study of protein domain distribution in the context of alternative splicing. *Nucleic Acids Research*, *31*(16). <https://doi.org/10.1093/nar/gkg668>
- Ma, X. R., Prudencio, M., Koike, Y., Vatsavayai, S. C., Kim, G., Harbinski, F., Briner, A., Rodriguez, C. M., Guo, C., Akiyama, T., Schmidt, H. B., Cummings, B. B., Wyatt, D. W., Kurylo, K., Miller, G., Mekhoubad, S., Sallee, N., Mekonnen, G., Ganser, L., ... Gitler, A. D. (2022). TDP-43 represses cryptic exon inclusion in the FTD-ALS gene UNC13A. *Nature*, *603*(7899). <https://doi.org/10.1038/s41586-022-04424-7>
- Mittendorf, K. F., Deatherage, C. L., Ohi, M. D., & Sanders, C. R. (2012). Tailoring of membrane proteins by alternative splicing of Pre-mRNA. *Biochemistry*, *51*(28). <https://doi.org/10.1021/bi3007065>
- Miyamoto, S., Nagamura, Y., Nakabo, A., Okabe, A., Yanagihara, K., Fukami, K., Sakai, R., & Yamaguchi, H. (2018). Aberrant alternative splicing of RHOA is associated with loss of its expression and activity in diffuse-type gastric carcinoma cells. *Biochemical and Biophysical Research Communications*, *495*(2). <https://doi.org/10.1016/j.bbrc.2017.12.067>
- Moore, M. J., Wang, Q., Kennedy, C. J., & Silver, P. A. (2010). An alternative splicing network links cell-cycle control to apoptosis. *Cell*, *142*(4). <https://doi.org/10.1016/j.cell.2010.07.019>
- Muller, I. B., Meijers, S., Kampstra, P., van Dijk, S., van Elswijk, M., Lin, M., Wojtuszkiewicz, A. M., Jansen, G., de Jonge, R., & Cloos, J. (2021). Computational comparison of common event-based differential splicing tools: practical considerations for laboratory researchers. *BMC Bioinformatics*, *22*(1). <https://doi.org/10.1186/s12859-021-04263-9>
- Ntranos, V., Yi, L., Melsted, P., & Pachter, L. (2019). A discriminative learning approach to differential expression analysis for single-cell RNA-seq. *Nature Methods*, *16*(2). <https://doi.org/10.1038/s41592-018-0303-9>
- Ogawa, T., Shiga, K., Hashimoto, S., Kobayashi, T., Horii, A., & Furukawa, T. (2003). APAF-1-ALT, a novel alternative splicing form of APAF-1, potentially causes impeded ability of undergoing DNA damage-induced apoptosis in the LNCaP human prostate cancer cell line. *Biochemical and Biophysical Research Communications*, *306*(2). [https://doi.org/10.1016/S0006-291X\(03\)00995-1](https://doi.org/10.1016/S0006-291X(03)00995-1)
- Phillips, J. W., Pan, Y., Tsai, B. L., Xie, Z., Demirdjian, L., Xiao, W., Yang, H. T., Zhang, Y., Lin, C. H., Cheng, D., Hu, Q., Liu, S., Black, D. L., Witte, O. N., & Xing, Y. (2020). Pathway-guided analysis identifies Myc-dependent alternative pre-mRNA splicing in aggressive prostate cancers. *Proceedings of the National Academy of Sciences*, *117*(10), 5269–5279. <https://doi.org/10.1073/PNAS.1915975117>
- Quayle, S. N., & Sadar, M. D. (2006). A truncated isoform of TMEFF2 encodes a secreted protein in prostate cancer cells. *Genomics*, *87*(5). <https://doi.org/10.1016/j.ygeno.2005.12.004>
- Raines, K. W., Bonini, M. G., & Campbell, S. L. (2007). Nitric oxide cell signaling: S-nitrosation of Ras superfamily GTPases. In *Cardiovascular Research* (Vol. 75, Issue 2). <https://doi.org/10.1016/j.cardiores.2007.04.013>
- Seiler, M., Peng, S., Agrawal, A. A., Palacino, J., Teng, T., Zhu, P., Smith, P. G., Caesar-Johnson, S. J., Demchok, J. A., Felau, I., Kasapi, M., Ferguson, M. L., Hutter, C. M., Sofia, H. J., Tarnuzzer, R., Wang, Z., Yang, L., Zenklusen, J. C., Zhang, J. (Julia), ... Yu, L. (2018). Somatic Mutational Landscape of Splicing Factor Genes and Their Functional Consequences across 33 Cancer Types. *Cell Reports*, *23*(1). <https://doi.org/10.1016/j.celrep.2018.01.088>
- Shen, S., Park, J. W., Lu, Z. X., Lin, L., Henry, M. D., Wu, Y. N., Zhou, Q., & Xing, Y. (2014). rMATS: Robust and flexible detection of differential alternative splicing from replicate RNA-Seq data. *Proceedings of the National Academy of Sciences of the United States of America*, *111*(51). <https://doi.org/10.1073/pnas.1419161111>
- Slaff, B., Radens, C. M., Jewell, P., Jha, A., Lahens, N. F., Grant, G. R., Thomas-Tikhonenko, A., Lynch, K.

- W., & Barash, Y. (2021). MOCCASIN: a method for correcting for known and unknown confounders in RNA splicing analysis. *Nature Communications*, *12*(1). <https://doi.org/10.1038/s41467-021-23608-9>
- Trikha, M., Cai, Y., Grignon, D., & Honn, K. V. (1998). Identification of a novel truncated α IIb integrin. *Cancer Research*, *58*(21).
- Trincado, J. L., Entizne, J. C., Hysenaj, G., Singh, B., Skalic, M., Elliott, D. J., & Eyraes, E. (2018). SUPPA2: Fast, accurate, and uncertainty-aware differential splicing analysis across multiple conditions. *Genome Biology*, *19*(1). <https://doi.org/10.1186/s13059-018-1417-1>
- Van Nostrand, E. L., Freese, P., Pratt, G. A., Wang, X., Wei, X., Xiao, R., Blue, S. M., Chen, J. Y., Cody, N. A. L., Dominguez, D., Olson, S., Sundararaman, B., Zhan, L., Bazile, C., Bouvrette, L. P. B., Bergalet, J., Duff, M. O., Garcia, K. E., Gelboin-Burkhart, C., ... Yeo, G. W. (2020). A large-scale binding and functional map of human RNA-binding proteins. *Nature*, *583*(7818). <https://doi.org/10.1038/s41586-020-2077-3>
- Vivian, J., Rao, A. A., Nothhaft, F. A., Ketchum, C., Armstrong, J., Novak, A., Pfeil, J., Narkizian, J., Deran, A. D., Musselman-Brown, A., Schmidt, H., Amstutz, P., Craft, B., Goldman, M., Rosenbloom, K., Cline, M., O'Connor, B., Hanna, M., Birger, C., ... Paten, B. (2017). Toil enables reproducible, open source, big biomedical data analyses. In *Nature Biotechnology* (Vol. 35, Issue 4). <https://doi.org/10.1038/nbt.3772>
- Vlachos, A., Rosenberg, P. S., Atsidaftos, E., Alter, B. P., & Lipton, J. M. (2012). Incidence of neoplasia in Diamond Blackfan anemia: A report from the Diamond Blackfan anemia registry. *Blood*, *119*(16). <https://doi.org/10.1182/blood-2011-08-375972>
- Warzecha, C. C., Jiang, P., Amirikian, K., Dittmar, K. A., Lu, H., Shen, S., Guo, W., Xing, Y., & Carstens, R. P. (2010). An ESRP-regulated splicing programme is abrogated during the epithelial-mesenchymal transition. *EMBO Journal*, *29*(19). <https://doi.org/10.1038/emboj.2010.195>
- Woolfson, A., & Milstein, C. (1994). Alternative splicing generates secretory isoforms of human CD1. *Proceedings of the National Academy of Sciences of the United States of America*, *91*(14). <https://doi.org/10.1073/pnas.91.14.6683>
- Xie, C., Jauhari, S., & Mora, A. (2021). Popularity and performance of bioinformatics software: the case of gene set analysis. *BMC Bioinformatics*, *22*(1). <https://doi.org/10.1186/s12859-021-04124-5>
- Xing, Y., Xu, Q., & Lee, C. (2003). Widespread production of novel soluble protein isoforms by alternative splicing removal of transmembrane anchoring domains. *FEBS Letters*, *555*(3). [https://doi.org/10.1016/S0014-5793\(03\)01354-1](https://doi.org/10.1016/S0014-5793(03)01354-1)
- Zerbino, D. R., Frankish, A., & Flicek, P. (2020). Progress, challenges, and surprises in annotating the human genome. In *Annual Review of Genomics and Human Genetics* (Vol. 21). <https://doi.org/10.1146/annurev-genom-121119-083418>
- Zhang, C., Zhang, B., Lin, L. L., & Zhao, S. (2017). Evaluation and comparison of computational tools for RNA-seq isoform quantification. *BMC Genomics*, *18*(1). <https://doi.org/10.1186/s12864-017-4002-1>
- Zhang, Y., Yan, L., Zeng, J., Zhou, H., Liu, H., Yu, G., Yao, W., Chen, K., Ye, Z., & Xu, H. (2019). Pan-cancer analysis of clinical relevance of alternative splicing events in 31 human cancers. *Oncogene*, *38*(40), 6678–6695. <https://doi.org/10.1038/s41388-019-0910-7>
- Zhang, Z., Mao, W., Wang, L., Liu, M., Zhang, W., Wu, Y., Zhang, J., Mao, S., Geng, J., & Yao, X. (2020). Depletion of CDC5L inhibits bladder cancer tumorigenesis. *Journal of Cancer*, *11*(2). <https://doi.org/10.7150/jca.32850>

Reviewers' Comments:

Reviewer #1:

Remarks to the Author:

Authors significantly improved the manuscript and addressed some of my comments. Unfortunately, some of my concerns were not properly addressed. The main my concern is usage of unconventional and hardly interpretable methods to quantify AS and define EP. Instead of direct selection of developmentally regulated exons, authors first select developmentally regulated pathways and then identify exons by correlation with such pathways.

Authors almost refused my suggestions to use more convenient methods such as ones based on alignments and read counts. Pearson correlation coefficient around 0.84 that authors obtained by comparison with rMATs is not really high and suggest sufficient difference between two quantification pipelines. Do main conclusions of the paper holds if alternative (and more convenient) quantification method is used? Authors showed that it is not the case, when they use Wilcox test to define developmental AS (see author response). While Wilcox test is not the best chose in this case (count-based statistics are preferred) I would expect it to produce meaningful results. Wilcox test is simple and easily interpretable, the fact, that Wilcox-only group is so drastically different from Pathway-defined exons casts doubts on the later. Might functional enrichment of pathway-defined exons emerge as a byproduct of usage of similar (embryonic) pathways to define these exons? It is hard to say confidently, but from figure 1D it seems like EP are frequently show high absolute correlation with their host genes and it seems to be positive for majority of EP. This exon-host gene correlation could perfectly explain functional enrichment of pathway-only exons. The correlation might be a biological phenomena but might be as well consequence of a quantification method.

In their response authors say that "Fig 1E shows the median of all the EP events". But 1E is boxplot and it shows the distributions, not just median. Taking into account that some outliers are included into the plot I assume that actually they all are shown (at least the opposite is not stated). The ranges of both 1E plots are smaller that 0.5, so, there are NO exons with dPSI > 0.5, at least they are not included into figure 1E, that contradict to author reply "There are several events for which the delta PSI is greater than 0.5". So there is clear contradiction between figure 1E and the reply, please clarify. More broadly, I believe that AS events with dPSI around 0.1 could play biological role. But there are plenty of known brain developmental switch-like (dPSI > 0.5) events (see vastdb or Mazin et al for example), that seems to be not found by this work. In combination with author refusal to use more orthodox methods (alignment and differential splicing analysis) it makes me worry about validity of the results.

In reply to comment 7 authors say that CSFs ", as we define them, could be either positive regulators of EP events or equivalently, negative regulators of EN events". But according to the methods (I846) authors used only SF with positive correlation to EP, that is only positive regulators of EP. Please clarify what was actually done. If Methods are correct, then SFs like ptpb2 (very important repressor of EN in brain and muscle) were omitted from analysis. Additionally, negative regulators of EP whose expression increases with development might be as important as positive regulators with decreasing expression. But only later were considered as CSF while former were omitted.

Reviewer #2:

Remarks to the Author:

The authors have made significant modification to their original submission, both in adding additional experimental validations of the presented points and hypotheses, as well as adding additional analysis and validations of their methods. They have satisfactorily addressed my comments from my first review. In particular the validation with RNAi and knock-out data available from public sources as well as the detailed literature searches provide important supporting data for the systems-level insights the

authors reach. This manuscript identifies multiple starting points for developmental as well as cancer-related studies, as well as potential drug targets and a logic for selecting possible drug targets, that make it relevant to a broad audience. Conceptually, I also find the link between developmental splicing patterns and different cancers a useful and concrete demonstration of how missplicing can contribute to oncogenesis.

I also want to comment that I find the authors' pipeline of Kallisto + SUPPA2 justified. The additional analysis with rMATs now helps justify this choice, but pseudoalignment approaches also perform well in our experience, and are comparable with classic mapping approaches. I feel that characterization of novel alternative splice events in the cancer lines is a different question, although also interesting. Given the clear reversion to at least a subset of embryonic events shown in this manuscript already in early stage cancers, I wonder if the rates of novel splice events might be higher in later stage cancers, reflecting progressive misregulation of splicing. The question would be if the novel events drive cancer progression in the same way the ES events do, or if each "type" of missplicing reflects a different role or stage in oncogenesis. This is however a distinct question from what has been addressed to this point in this manuscript.

Minor:

In Figure 7, the A and B panel labels are missing in the figure, although they are mentioned in the legend.

Reviewer #3:

Remarks to the Author:

The authors have revised the manuscript and addressed all my comments and concerns. I recommend publication of this important work.

REVIEWER COMMENTS

Reviewer #1 (Remarks to the Author):

Authors significantly improved the manuscript and addressed some of my comments. Unfortunately, some of my concerns were not properly addressed. The main my concern is usage of unconventional and hardly interpretable methods to quantify AS and define EP. Instead of direct selection of developmentally regulated exons, authors first select developmentally regulated pathways and then identify exons by correlation with such pathways.

Authors almost refused my suggestions to use more convenient methods such as ones based on alignments and read counts. Pearson correlation coefficient around 0.84 that authors obtained by comparison with rMATs is not really high and suggest sufficient difference between two quantification pipelines. Do main conclusions of the paper holds if alternative (and more convenient) quantification method is used? Authors showed that it is not the case, when they use Wilcoxon test to define developmental AS (see author response). While Wilcoxon test is not the best chose in this case (count-based statistics are preferred) I would expect it to produce meaningful results. Wilcoxon test is simple and easily interpretable, the fact, that Wilcoxon-only group is so drastically different from Pathway-defined exons casts doubts on the later.

We are glad that the reviewer found the revised version of our manuscript significantly improved. In this comment the reviewer is referring to two different issues -- (1) our approach to quantify alternative splicing events, and (2) our approach to identify differential splicing events in pre- and post-natal stages of development. We will address these in turn.

With regards to our approach to quantify splicing, Kallisto + SUPPA2, the reviewer asks a relevant question: "do the main conclusions hold if we use an alternative method to quantify splicing events." We address this below, however, the reviewer brings up Wilcoxon test in this case, which has nothing to do with quantifying splicing but rather has to do with quantifying differential splicing, which is a separate downstream step after splicing quantification in each sample, and should be compared with PEGASAS; we will address that in turn further below.

Now we return to the first issue: whether our conclusion holds when using an alternative approach to quantify splicing events (i.e replacing SUPPA2)? In response to this concern, in the previous round, we had shown that across the developmental samples the splicing quantification using an alternative tool - rMATs is highly correlated (Pearson rho ~ 0.84) with SUPPA quantification. Given the complexity of these bioinformatics approaches, some discrepancy between any two tools is expected and we think that a global correlation of 0.84 should be considered high. We are unsure as to why the reviewer doesn't consider this level of correlation an adequate support of robustness.

Regardless, now additionally we have implemented yet another pipeline for both splicing quantification and differential splicing. Specifically, we replaced our original pipeline (i.e. kallisto + SUPPA2 + PEGASAS) with a new pipeline (STAR 2 pass + rMATs + Wilcoxon/Pearson's correlation). For this analysis, we downloaded the controlled access data from TCGA and GTEx for brain samples (i.e TCGA cancer and GTEx normal for brain) and quantified the splicing using STAR2 pass alignment (using the same pipeline

used by TCGA) following by rMATs. To re-emphasize, we completely eliminated the use of pseudo-alignments as well as SUPPA2 for splicing quantification with this pipeline. Further, we processed the RNA-seq data for human brain development using the same pipeline to quantify splicing. To assess if cancer splicing broadly reverts back to the pre-natal splicing, first we directly compared the Δ PSI between cancer (TCGA) and healthy (GTEx) brain samples against the Δ PSI between pre-natal and post-natal samples. As shown in the scatter plot below, we observed a highly significant and positive correlation between cancer associated changes (i.e TCGA - GTEx) and development associated changes (i.e Pre natal - post natal), providing a direct and additional validation of the main premise of our work with an entirely independent data processing pipeline.

Further, using this new splicing data, we defined a new set of embryonic splicing events (namely, EP for embryonic positive and EN for embryonic negative) by using Wilcoxon's rank sum test (absolute median inclusion difference > 0.2 and FDR < 0.1) between pre-natal and post-natal inclusion level of the exon skip events. We compared this set of the embryonic splicing events, using Fisher's exact test, with the cancer specific events (namely 'Increased' for events with increased inclusion and 'decreased' for events with decreased inclusion in TCGA cancer samples as compared to GTEx normals) which were derived using a similar approach. We observed a very strong and significant enrichment of the EP events among the events increased in cancer (and vice-versa for EN events), as shown in the bar-plots below:

These results entirely recapitulate our conclusions based on the alternative pipeline presented in the manuscript. We have added this additional analysis as Supplementary note 2 provide a measure of robustness to the readers.

Coming back to the second issue regarding the PEGASAS, there are few things that we would like to mention:

1. Firstly, in the previous revision, we clearly state that pathway-based approach provided a more direct interpretation of the functionality of the detected exons. i.e., an exon correlated with multiple embryonic pathways should be relevant to the embryonic development, whereas functional interpretation of the events identified based on differential splicing alone in a pathway-agnostic approach is not as straightforward.
2. The events selected based on pathway-based approach are more strongly correlated with each other, which is an important aspect for the functional relevance of alternative splicing as explained in the first revision and cited in our manuscript.
3. Thirdly, as previously suggested by the reviewer, we used an alternative and commonly used approach (based on Wilcoxon's rank sum test) to identify differential splicing events between pre- and post-natal stages of the development. Overall, the differently EP events identified based on this approach were significantly enriched in their respective cancers. This observation suggests that our results are not biased.
4. Additionally, relative to the events detected uniquely by the pathway approach, the events uniquely detected through the Wilcoxon based approach were poorly correlated amongst each other and were not as enriched in cancer-associated events.
5. Lastly, and importantly, as asked by the reviewers, we had also made the scatter plots of global splicing changes during development (pre-post) and cancer (TCGA-Gtex) and observed a significant and positive correlation between the two metrics in Brain and Liver. This is the most straightforward and unambiguous support of the hypothesis that splicing in cancer reverts to its embryonic counterparts.

Also, as shown above by our latest and independent analysis using STAR2 pass alignment, rMATs and Wilcoxon's test (and correspondingly, scatter plot of cancer associated changes with development associated changes), we observed a significant enrichment of the embryonic splicing events in cancer (and correspondingly significant positive correlation), thereby implicating that embryonic reversal of

cancer splicing is statistically strong such that it is immune to the method one chooses to quantify and detect splicing events.

Might functional enrichment of pathway-defined exons emerge as a byproduct of usage of similar (embryonic) pathways to define these exons? It is hard to say confidently, but from figure 1D it seems like EPs frequently show high absolute correlation with their host genes and it seems to be positive for the majority of EPs. This exon-host gene correlation could perfectly explain functional enrichment of pathway-only exons. The correlation might be a biological phenomenon but might be as well a consequence of a quantification method.

We appreciate this additional feedback from the reviewer. To test the possibility that the correlation between EP exons and their host genes may explain their functional enrichment, we performed the functional enrichment on a subset of EP exons' host genes, specifically, only those host genes which were uncorrelated with their EP exons (p -value of host gene \sim exon correlation > 0.05). Interestingly, even in that case, we observed a significant enrichment of the several neuron and brain developmental functional terms, as shown in the dotplot below. Therefore, the functional enrichment is not driven by the exon \sim host-gene correlation.

In their response authors say that “Fig 1E shows the median of all the EP events”. But 1E is boxplot and it shows the distributions, not just median. Taking into account that some outliers are included into the plot I assume that actually they all are shown (at least the opposite is not stated). The ranges of both 1E plots are smaller than 0.5, so, there are NO exons with dPSI > 0.5, at least they are not included into figure 1E, that contradict to author reply “There are several events for which the delta PSI is greater than 0.5”. So there is clear contradiction between figure 1E and the reply, please clarify. More broadly, I believe that AS events with dPSI around 0.1 could play biological role. But there are plenty of known brain developmental switch-like (dPSI > 0.5) events (see vastdb or Mazin et al for example), that seems to be not found by this work. In combination with author refusal to use more orthodox methods (alignment and differential splicing analysis) it makes me worry about validity of the results.

We regret the lack of clarity. In figure 1E, each datapoint within a boxplot is the median value across the individual EP event, and the boxplot represents the median EP event splicing levels across the samples (pre- or post-natal). Since high delta-PSI (> 0.5) are expected to be achieved by a minority of EP events, the median occludes such events. We hope that this clarifies the apparent contradiction. To be sure, when we look at the delta-PSI of individual EP events (which is equivalent to ‘down’ category defined in Mazin et al.), we do find many that are > 0.5. In particular, we find 75 exon skip events in Liver, 122 in Kidney, and 86 in Brain having DPSI > 0.5 (pre – post natal). To compare these numbers with the suggested resources, we looked into Mazin et al. data where for ‘down’ category (equivalent to EP), just 14 cassette exons in Liver, 128 in Kidney, and 181 in brain had dPSI value of >0.5. The reviewer’s anticipation is correct and indeed we find several EP events with high DPSI. Moreover, as we have discussed above, even using the ‘orthodox method’ to quantify splicing robustly supports our conclusions.

In reply to **comment 7** authors say that CSFs “, as we define them, could be either positive regulators of EP events or equivalently, negative regulators of EN events”. But according to the methods (l846) authors used only SF with positive correlation to EP, that is only positive regulators of EP. Please clarify what was actually done. If Methods are correct, then SFs like ptbp2 (very important repressor of EN in brain and muscle) were omitted from analysis. Additionally, negative regulators of EP whose expression increases with development might be as important as positive regulators with decreasing expression. But only later were considered as CSF while former were omitted.

We again apologize for the lack of clarity. To answer this question, we need to get into some background context. As the reviewer would surely appreciate, in diseases such as cancer, one would typically want to target gain of function mutations (~or genes) which might drive the cancer progression. Loss of function mutations (~ genes) are already inactivated in cancer, and therefore do not make good therapeutic targets, without relying on indirect approaches such as synthetic lethality.

For splicing factors, we focused on the positive regulators of EP splicing. So, our description in the methods section is correct. In the previous response document, we just made the point that we can define the critical splicing factors either as positive or negative regulators, depending on the direction of regression coefficients. In our case, we chose to follow up on the positive regulators of EP events because we wanted to focus on gain of function, which is therapeutically targetable as explained above.

Also, we did not omit the factor PTPB2, and as per our PLSR model in Brain, it is one of the critical positive regulators of EP splicing (Supplementary Table S5). Analogously if we choose the response variable to be EN events in our PLSR model, then PTPB2 is a critical negative regulator of EN splicing). To summarize, as the expression of PTPB2 goes up, our model predicts that EP would also go up (and EN would go down).

Reviewer #2 (Remarks to the Author):

The authors have made significant modification to their original submission, both in adding additional experimental validations of the presented points and hypotheses, as well as adding additional analysis and validations of their methods. They have satisfactorily addressed my **comment**s from my first review. In particular the validation with RNAi and knock-out data available from public sources as well as the detailed literature searches provide important supporting data for the systems-level insights the authors reach. This manuscript identifies multiple starting points for developmental as well as cancer-related studies, as well as potential drug targets and a logic for selecting possible drug targets, that make it relevant to a broad audience. Conceptually, I also find the link between developmental splicing patterns and different cancers a useful and concrete demonstration of how missplicing can contribute to oncogenesis.

I also want to **comment** that I find the authors' pipeline of Kallisto + SUPPA2 justified. The additional analysis with rMATs now helps justify this choice, but pseudoalignment approaches also perform well in our experience, and are comparable with classic mapping approaches. I feel that characterization of novel alternative splice events in the cancer lines is a different question, although also interesting. Given the clear reversion to at least a subset of embryonic events shown in this manuscript already in early stage cancers, I wonder if the rates of novel splice events might be higher in later stage cancers, reflecting progressive misregulation of splicing. The question would be if the novel events drive cancer progression in the same way the ES events do, or if each "type" of missplicing reflects a different role or stage in oncogenesis. This is however a distinct question from what has been addressed to this point in this manuscript.

We thank the reviewer for the positive and encouraging comments. Indeed, we do think that our work could serve as a valuable resource for those interested in alternative splicing in development and in cancer, and in fact we have already initiated such follow up works in collaboration with experimentalists. Reviewer makes a very intriguing point about the functional class and significance of early versus late occurring alternative splicing events which we will plan on pursuing. **We have now discussed** the following in the manuscript **at line 547**.

In our current work, we do not focus on identifying novel splicing events. However, this is a potentially interesting investigation, and the implication is that alternative splicing can derive cancer progression through one of the two different routes; either through the reactivation of the multiple aspects of embryonic physiology as shown in the current work, or other through the neo-functionalization as has been proposed to occur during evolution via alternative splicing (Bush et al., 2017; Liu et al., 2017)

and/or gene duplication (Teshima & Innan, 2008) or a combination thereof. Worth mentioning here is an interesting trend that was highlighted more than a decade ago that the number of duplicates and number of alternatively spliced isoforms of genes are inversely correlated during evolution (<https://journals.plos.org/ploscompbiol/article?id=10.1371/journal.pcbi.0030033>), implying that new functions and proteome diversity arise through these complementary pathways. To us, this means that alternative splicing might provide additional complexity to the proteome of the cancer cells, and such complexity is most likely driven through novel splicing events.

Therefore, regarding this point, we speculate that it is not just the stage of cancer progression which could influence the rate of novel splicing events, but cancers in distinct patients might be driven either by the reactivation of embryonic splicing patterns, or the expression of the neo-junctions (Kahles et al., 2018) making functional improvisations as evidenced through the lens of evolution (Bush et al., 2017).

Additionally, as rightly reinforced by the reviewer, we are also happy to share that after additional concerns from the first reviewer, we repeated our splicing analysis by replacing the Kallisto + SUPPA2 pipeline with STAR2 + rMATs pipeline. Even with this entirely independent data processing, we observed a strong enrichment of the embryonic splicing events in cancer which provide us confidence regarding the technical ingredients in our results.

Minor:

In Figure 7, the A and B panel labels are missing in the figure, although they are mentioned in the legend.

We appreciate this correction from the reviewer and have made the desired change in Figure 7.

Reviewer #3 (Remarks to the Author):

The authors have revised the manuscript and addressed all my **comments** and concerns. I **recomm**end publication of this important work.

We are grateful to the reviewer for constructive comments and positive assessment of the revised manuscript.

Reviewers' Comments:

Reviewer #1:

Remarks to the Author:

I thank authors for additional analysis performed. It is now clear that similarity between embryonic stage and cancer can be found even using conventional methods of alternative splicing analysis. However I have other questions.

1. I still cannot understand what is shown on figure 1E even with author explanation from their replay. Figure legend provide almost no information, so it will be completely understandable for reader. What are data points on this plot?
2. What background gene set was used for GO-enrichment analysis?
3. Go enrichment analysis shown in author replay differs drastically from fig 1C, so, contrary to their replay it is strongly affected by exon ~ host gene correlation.
4. I also want to return back to one of my comments from fist review (minor comment 23). Authors claim that their model predict changes but show results of prediction only for absolute values, not for dPSI. They provide no details about their AUC analysis in the replay, so it is hard to understand whether it confirms their statement or not, anyway this analysis was not included into the manuscript. I also want to clarify what I meant by trivial model. It is just dataset-wise mean (or median) PSI of each event.
5. Single cell analysis (integration, umap constraction, cell type annotation, etc) is not described in the methods

Reviewer's #1 comments

I thank authors for additional analysis performed. It is now clear that similarity between embryonic stage and cancer can be found even using conventional methods of alternative splicing analysis. However I have other questions.

We are glad that reviewer is convinced that reactivation of developmental splicing is an observation which is robust with respect to the choice of the computational and statistical methods.

1. I still cannot understand what is shown on figure 1E even with author explanation from their replay. Figure legend provide almost no information, so it will be completely understandable for reader. What are data points on this plot?

We thank the reviewer for pointing our attention towards the legend of figure 1E. To clarify it again here, each data point in the boxplot is the median inclusion level of all the EP events for each developmental time point, the Boxplot itself shows the distribution of these values across the prenatal (correspondingly, postnatal) timepoints. We have edited the legend of figure 1E to alleviate any potential confusion for the readers.

2. What background gene set was used for GO-enrichment analysis?

Background for GO-enrichment analysis was the default whole genome, i.e., all the genes which are annotated in GO database.

3. Go enrichment analysis shown in author replay differs drastically from fig 1C, so, contrary to their replay it is strongly affected by exon ~ host gene correlation.

So to provide a quick background, in figure 1C, we had shown the functional enrichment of the genes which constitutes the set of KEGG pathways that are preferentially active during pre-natal development. We also had included supplementary figure 1E (and Supp. Table S4) which showed that the identified embryonic events are significantly related to tissue and developmental functional terms for brain and liver.

The reviewer essentially objected that the enrichment shown in Supp. Fig 1E (and the pathway-only exon in Supp Note 1) might be driven by correlations between exon inclusion and host gene expression. To quote the reviewer "*The correlation might be a biological phenomenon but might be as well consequence of a quantification method*". In order to address this concern, we repeated the functional enrichment analysis by excluding the exons whose inclusion was correlated with their host genes (i.e. p-value of correlation ≤ 0.05). Even in that case, we observed a significant enrichment of the similar functional terms (as in Supp figure 1E) as shown below:

New figure after excluding correlated exons

Old Supp fig 1E

Therefore, the results from two functional enrichment analyses are highly concordant, and several terms are specifically related to brain development and physiology such as pallium development, dendrite development, synapse organization, regulation of neurotransmitters, neuron death, glutamate secretion, and developmental cell growth.

However, in the most recent review, reviewer #1 incorrectly compares the new figure with Fig 1C, instead of fig 1E (or pathway-only GO enrichment figure in Supp note 1), and objects that the new enriched terms are drastically different from before. This is clearly a simple oversight on the reviewer’s part.

4. I also want to return back to one of my comments from fist review (**minor** comment 23). Authors claim that their model predict changes but show results of prediction only for absolute values, not for dPSI. They provide no details about their AUC analysis in the replay, so it is hard to understand whether it confirms their statement or not, anyway this analysis was not included into the manuscript. I also want to clarify what I meant by trivial model. It is just dataset-wise mean (or median) PSI of each event.

We appreciate this concern, but we would like to again clarify the objectives of the PLSR model. The goal of our PLSR model was to predict the 'median' inclusion level of all the EP events, which is a single numeric value for each sample. More specifically, we are not predicting the inclusion level of individual splicing event (as they are expected to be correlated). Furthermore, estimation of dPSI in the cancer context requires paired tumor-normal data, which is currently inadequate for our purposes.

5. Single cell analysis (integration, umap construction, cell type annotation, etc) is not described in the methods

Thanks for this comment and we would like to clarify that cell type annotations and other steps involved in the analysis of single cell RNA-seq data were performed by the respective authors of the publications from which data were downloaded. Specifically, not only the processing single cell data, but also the cell annotation (into malignant and non-malignant), as well as the UMAP coordinates were downloaded directly. We have now make it even more explicit in our methods section.